# Selenium-based metabolic oligosaccharide engineering strategy for quantitative glycan detection

Xiao Tian [1], Lingna Zheng [2], Changjiang Wang [1], Yida Han [1], Yujie Li [1], Tongxiao Cui [1], Jialin Liu [3], Chuanming Liu [4], Guogeng Jia [3], Lujie Yang [5], Yi Hsu [1], Chen Zeng [5], Lijun Ding [4], Chu Wang [3], Bo Cheng [6], Meng Wang [2] ✉ & Ran Xie [1] ✉

Metabolic oligosaccharide engineering (MOE) is a classical chemical approach to perturb, profile and perceive glycans in physiological systems, but probes upon bioorthogonal reaction require accessibility and the background signal readout makes it challenging to achieve glycan quantification. Here we develop SeMOE, a selenium-based metabolic oligosaccharide engineering strategy that concisely combines elemental analysis and MOE, enabling the mass spectrometric imaging of glycome. We also demonstrate that the new-to-nature SeMOE probes allow for detection, quantitative measurement and visualization of glycans in diverse biological contexts. We also show that chemical reporters on conventional MOE can be integrated into a bifunctional SeMOE probe to provide multimodality signal readouts. SeMOE thus provides a convenient and simplified method to explore the glyco-world.

Metabolic oligosaccharide engineering (MOE) is an enticing chemical glycobiological approach to probe, perturb and perceive glycans[1,2]. By hijacking the underlying glycan biosynthetic machinery, monosaccharides carrying chemical modifications are supplied to cells or living organisms. These metabolic precursors/analogs are metabolically activated and incorporated into the glycomes. In a second step, modification with bioorthogonal functionalities (e.g., azide, alkyne, also called the chemical reporter or chemical handle), can be probed by bioorthogonal reactions, to allow for the dynamic and versatile glycan visualization, as well as glycoconjugate isolation and -omic analysis (Fig. 1a)[3–5]. So far, this powerful technique enabled the probing of cell-surface sialoglycoconjugates, mucin-type O-linked glycosylation (i.e., O-GalNAc type), O-GlcNAcylation and fucosylation in vertebrate cells and living animals[6], as well as the probing of bacterial glycoconjugate constituents such as bacillosamine (Bac)[7],

pseudodaminic acid (Pse)[8], 3-dexoy-D-manno-oct-2-ulosonic acid (Kdo)[9] and N-acetylmuramic acid (MurNAc)[10], to investigate and manipulate glycans in a myriad of biophysiological contexts. Of note, the MOE probes, both in their free form and in the hydrophobically caged form that improves metabolic efficiency, have been realized. However, one major limitation of MOE has been the lack of quantitative glycan measurement.

We envisaged that the extension of MOE with a more "quantitative" readout both in vitro and in vivo would greatly augment the utility of this classical strategy. For instance, intercellular exchange of cell-surface molecules (e.g., glycoconjugates) is a general yet pivotal process in the immune system, such as the interaction between natural killer (NK) cells and their target cells[11]. The advent of such detection methodology would allow researchers to probe the amount of glycans transferred in a designated cell or organ. The Varki group has

[1]State Key Laboratory of Coordination Chemistry, School of Chemistry and Chemical Engineering, Chemistry and Biomedicine Innovation Center (ChemBIC), Nanjing University, Nanjing, China. [2]CAS Key Laboratory for Biomedical Effects of Nanomaterials and Nanosafety, Institute of High Energy Physics, Chinese Academy of Sciences, Beijing, China. [3]College of Chemistry and Molecular Engineering, Peking University, Beijing, China. [4]Center for Reproductive Medicine and Obstetrics and Gynecology, Nanjing Drum Tower Hospital, Nanjing University Medical School, Nanjing, China. [5]Department of Pharmacology, School of Medicine, Southern University of Science and Technology, Shenzhen, China. [6]School of Pharmaceutical Sciences, Peking University, Beijing, China. ✉e-mail: wangmeng@ihep.ac.cn; ranxie@nju.edu.cn

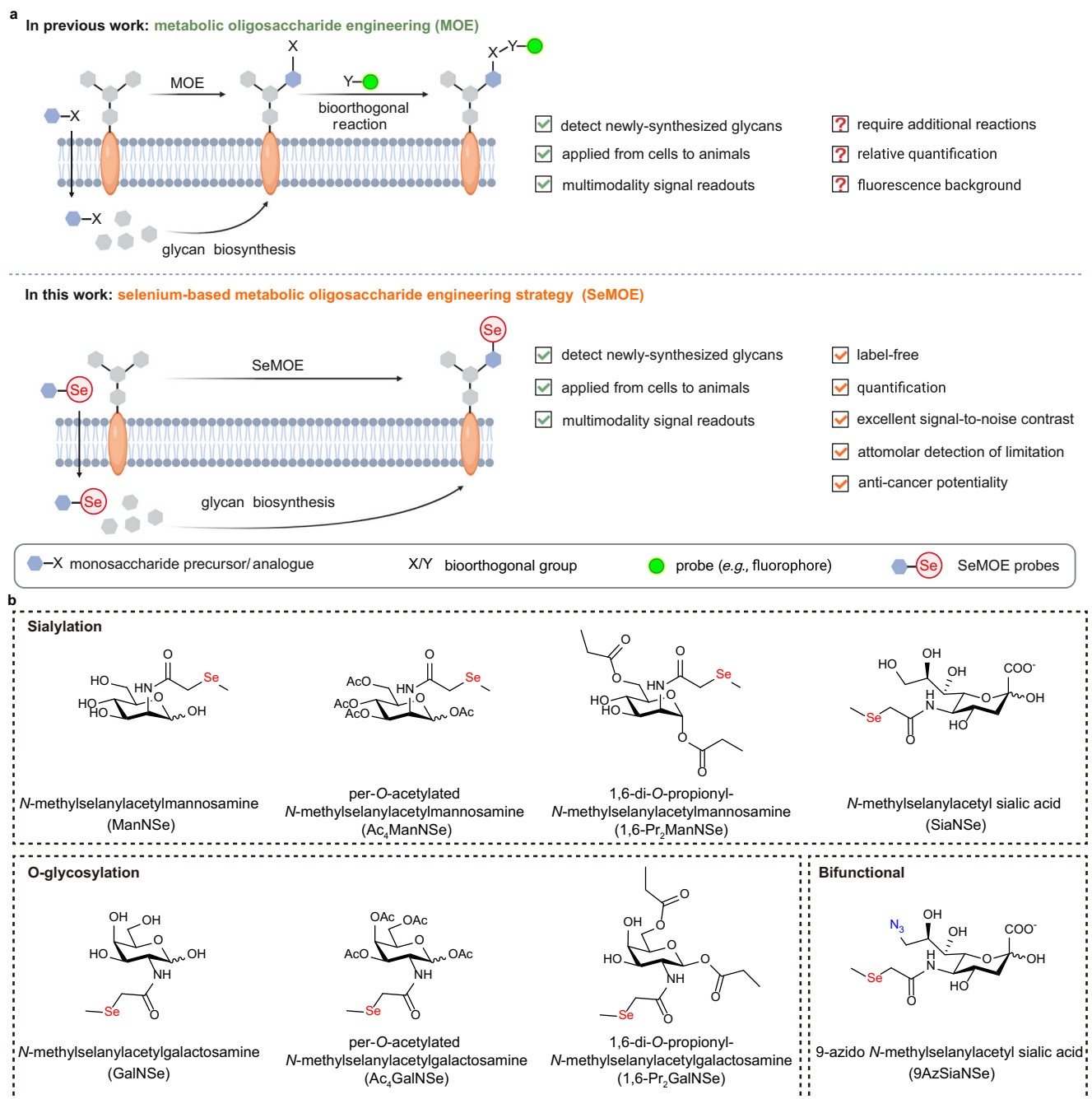

**Fig. 1 | Schematic of SeMOE methodology. a** Comparison of SeMOE and classical MOE. Instead of using monosaccharide reporter bearing bioorthogonal functionalities (e.g., azide) in standard MOE procedure, selenium-containing probes were metabolically incorporated for glycan detection based on elemental analysis. **b** SeMOE probes used in this study. Figure created with BioRender.com.

demonstrated this concept via radiolabeling approaches, but suffers from cost-ineffectiveness and radioactive safety risks[12]. Additionally, maneuvering MOE and installing the incorporated bioorthogonal functional groups with alternative signal output modalities, such as 1,2-diamino-4,5-methylenedioxybenzene (DMB)[13], Europium (Eu) isotopic mass tag[14] and surface-enhanced Raman spectroscopy (SERS) probe, have allowed for the detection and quantification of monosaccharide derivates[15]. Nevertheless, the stoichiometric attachment of complementary probes to glycoconjugates is inevitably hampered by their accessibility, and the non-specific interactions between added probes and cellular components (e.g., cell surface) additionally increased the difficulty of quantitative readouts. Therefore, developing a simple, safe

and sensitive quantification methodology for glycan detection is necessary.

To expand the applicability of MOE, herein we report a selenium-based metabolic oligosaccharide engineering strategy (SeMOE) that combines elemental analysis and MOE to enable the quantity assessment and mass spectrometric imaging of glycome in a concise procedure (Fig. 1a). We demonstrate that these SeMOE probes are readily adaptable toward the detection, quantification and visualization of glycans in diverse biological contexts. In addition, chemical reporters on conventional MOE can be integrated into a bifunctional SeMOE probe, to provide multiple sensing modes. Finally, a preliminary investigation of the potential anti-cancer function of SeMOE probes is also shown.

## Results

### Development of SeMOE probes

Selenium is an essential trace nutrient for cellular functions in the physiological environment, mostly in the form of unusual amino acids selenocysteine and selenomethionine, to form selenoproteins[16]. Selenium can be quantitatively analyzed and imaged by elemental analysis methods, including atomic absorption spectroscopy (AAS)[17], thermal ionization mass spectrometry (TIMS)[18], glow discharge mass spectrometry (GDMS)[19], and x-ray fluorescence (XRF)[20]. Noteworthy, the inductively coupled plasma mass spectrometry (ICP-MS) is a robust quantitative technique for multi-element analysis and isotope ratio measurement at the trace and ultra-trace level in life sciences[21]. In this technique, samples are firstly ionized using an inductively coupled plasma (ICP), generating atomic and small polyatomic ion beams. After ion beam concentration via the ion optics, the ionized isotopes were detected via the mass spectrometer (MS). ICP-MS has excellent detection limits for most elements in the periodic table (sub-femtogram levels), a wide linear dynamic range (8 orders of magnitude), high analytical throughput, and minimal matrix effects[22]. ICP-MS has now evolved into a versatile platform technology with hyphenated techniques, and exhibits great compatibility with varied sample forms such as solution, tissue slices and even a single cell[23,24].

We chose to introduce the selenium element in the form of $N$-methylselanylacetyl side chain for sialic acid, mannosamine and galactosamine, because pertinent literature solidified that glycan biosynthetic pathway can tolerate the subtle chemical modification(s) on these analogs, and effectively situate them into cellular sialoglycans and $O$-glycans in the same way as their natural counterparts[25,26]. We therefore synthesized a total of 8 selenium-containing unnatural sugar precursors/analogs, which we termed "SeMOE probes" thereafter (Fig. 1b, Supplementary Note 1). To emphasize, per-$O$-acetylated, partially $O$-propionylated, as well as unprotected versions of SeMOE probes were all used in this work, to ensure the successful SeMOE incorporation. Once installed into the glycoconjugates, we expect a label-free selenium spike upon minimal-to-none background elemental signal, with guaranteed detection of limitation and sensitivity (Fig. 1a). The unique isotopic pattern feature of selenium was also demonstrated for the sialome identification in a glycoproteomic analysis.

### SeMOE enables label-free glycan detection and quantification

We started with a feasibility evaluation of the SeMOE strategy. Selenosugars (1β-methylseleno-galactosamine, 1β-methylseleno-$N$-acetylgalactosamine and 1β-methylseleno-$N$-acetylglucosamine) were previously reported as the key metabolite for selenium detoxication[27,28]. We therefore asked whether the SeMOE probes, in their current forms, are toxic. In general, millimolar (mM) concentrations of unprotected sialic acid analogs (e.g., ManNAz, SiaNAz), or micromolar (μM) concentrations of per-$O$-acetylated/partially $O$-propionylated $N$-acetylhexosamine analogs (e.g., Ac$_4$ManNAz, 1,6-Pr$_2$GalNAz) are widely used in a variety of cell lines and accepted as non-toxic. Selenium-containing monosaccharides exhibited cytotoxicities comparable with their corresponding native sugar or azidosugar counterparts, as measured by a commercialized CCK-8 assay (Supplementary Fig. 1). In addition, lectin-based flow cytometric analysis confirmed that treatment of SeMOE probes at safe dosages did not inhibit or alter the intrinsic sialic acid synthesis in HeLa and MCF-7 cells (Supplementary Fig. 2). We also do not observe significant changes in major selenoprotein expression level, as evidenced by RT-qPCR analysis (Supplementary Fig. 3). To verify the metabolic incorporation of the SeMOE probes, HeLa cells were incubated with SeMOE probes within a safe dosage range (varied from 50 μM to 2 mM), lysed for protein extraction, and analyzed using ICP-MS. We observed a concentration- and time-dependent increase in the total Se level, with minimal background noise, as normalized by the Se standard curve (Fig. 2a, Supplementary Fig. 4, Supplementary Fig. 5). Besides, Se levels

showed probe type-dependent increase with Ac$_4$ManNSe > 1,6-Pr$_2$ManNSe > SiaNSe > 9AzSiaNSe > ManNSe. We further performed the titration experiment on HeLa cells with 1,6-Pr$_2$ManNSe and SiaNSe, and confirmed that selenium detection by ICP-MS provided nanomole (nM) detection sensitivity (Fig. 2b). To exemplify, a signal-to-noise ratio (S/N) of $6.2 \pm 0.2$ was achieved at protein fractions in cells treated with 1 nM 1,6-Pr$_2$ManNSe (Fig. 2b). In comparison, fluorochromes via bioorthogonal reaction with equal S/N required treatment of 10 μM 1,6-Pr$_2$ManNAz in identical HeLa cell line[29].

ICP-MS hyphenated techniques like laser ablation (LA-ICP-MS)[30], or ICP-MS-based techniques such as cytometry by time of flight (CyTOF)[31], add spatial resolution to data acquisition. We wonder whether SeMOE probes are also characterizable with these techniques. To ascertain, whole-cell proteins treated with a panel of 1,6-Pr$_2$ManNSe, were resolved on SDS-PAGE, transferred to PVDF membrane, and subjected to LA-ICP-MS linear scan analysis (Fig. 2c). Robust $^{78}$Se signals were observed in the protein bands for the 1,6-Pr$_2$ManNSe-treated group, whereas only basal level of selenium signal was detected in the control group (Fig. 2c). We also validated that CyTOF can effectively and quantitatively measure the newly-incorporated selenium elements in HeLa and MCF-7 cell lines (Fig. 2d, Supplementary Fig. 6). Remarkably, all sialo-SeMOE probes demonstrated linear correlation between cell number and total Se level (Fig. 2e, Supplementary Fig. 7).

Next, we sought to verify that the selenium signal obtained from ICP-MS was indeed associated with glycans in vitro. To begin with, HeLa cells were co-incubated with new-to-nature sialo-SeMOE probes. Treatment with *Streptococcus pneumoniae* neuraminidase A (sialidase) on the cell-surface milieu or in cell lysates, both demonstrated significant decreases in Se signal[32] (Fig. 2f), indicating that most Se signal is localized in cell-surface sialylated glycans, and the signal is probe type-dependent. Interestingly, Ac$_4$ManNSe exhibited the most salient neuraminidase resistance, in accordance with the protein S-glyco-modificaiton (Supplementary Fig. 8), a side reaction when per-$O$-acetylated $N$-acetylhexosamines were used in MOE[33]. Instead, partially $O$-propionylated $N$-acetylhexosamine analogs at the C-1 and C-6 positions effectively circumvent this side reaction[29]. Alternatively, treatment of cell-surface glycoproteins with peptide-N-glycosidase F (PNGase F), which cleaves the asparagine side chain amide bond between glycoproteins and $N$-glycans[34], revealed a similar Se signal trend (Supplementary Fig. 9a). Besides, treatment with P-3F$_{AX}$Neu5Ac[35], a cell-permeable metabolic inhibitor of sialic acid biosynthesis also reduced the Se signal in a dose-dependent manner (Supplementary Fig. 9b). In addition, we confirmed the antagonization between sialo-SeMOE probes and their native or azido-functionalized counterparts (Fig. 2g, Supplementary Fig. 10). Eventually, we quantitatively measured the metabolic conversion rate of sialo-SeMOE probes in glycoproteins using DMB derivatization (Fig. 2h). The SiaNSe peaks from cellular samples shared same elution time with synthetic standard substance, demonstrating that sialo-SeMOE probes, although varied in their initial forms, were metabolically incorporated into glycoproteins. The incorporation rate of Ac$_4$ManNSe, 1,6-Pr$_2$ManNSe, SiaNSe and ManNSe were 33.51%, 16.53%, 5.23% and 1.74%, respectively. We also noticed a dose-dependent increase in sialo-SeMOE probe uptake and incorporation using DMB derivatization (Supplementary Fig. 11). Previous research based on Europium mass tag and bioorthogonal reaction gave the quantitative density of cell-surface newly-synthesized fucosylation at $5\text{-}100 \times 10^{-18}$ mol/cell[14]. Based on ICP-MS, we measured the Se signal at $100\text{-}2000 \times 10^{-18}$ mol/cell in various cancer cell lines, which was comparable in the orders of magnitude. Although the methodology and the object of study are different, these numbers will perceptually provide the ICP-MS readouts. These data, collectively, support the central tenant that the seleno-tagged sialic acid probes majorly deposit selenium, in the form of sialic acid derivative, into sialoglycans, via the glycan biosynthetic pathway. We also demonstrated that the

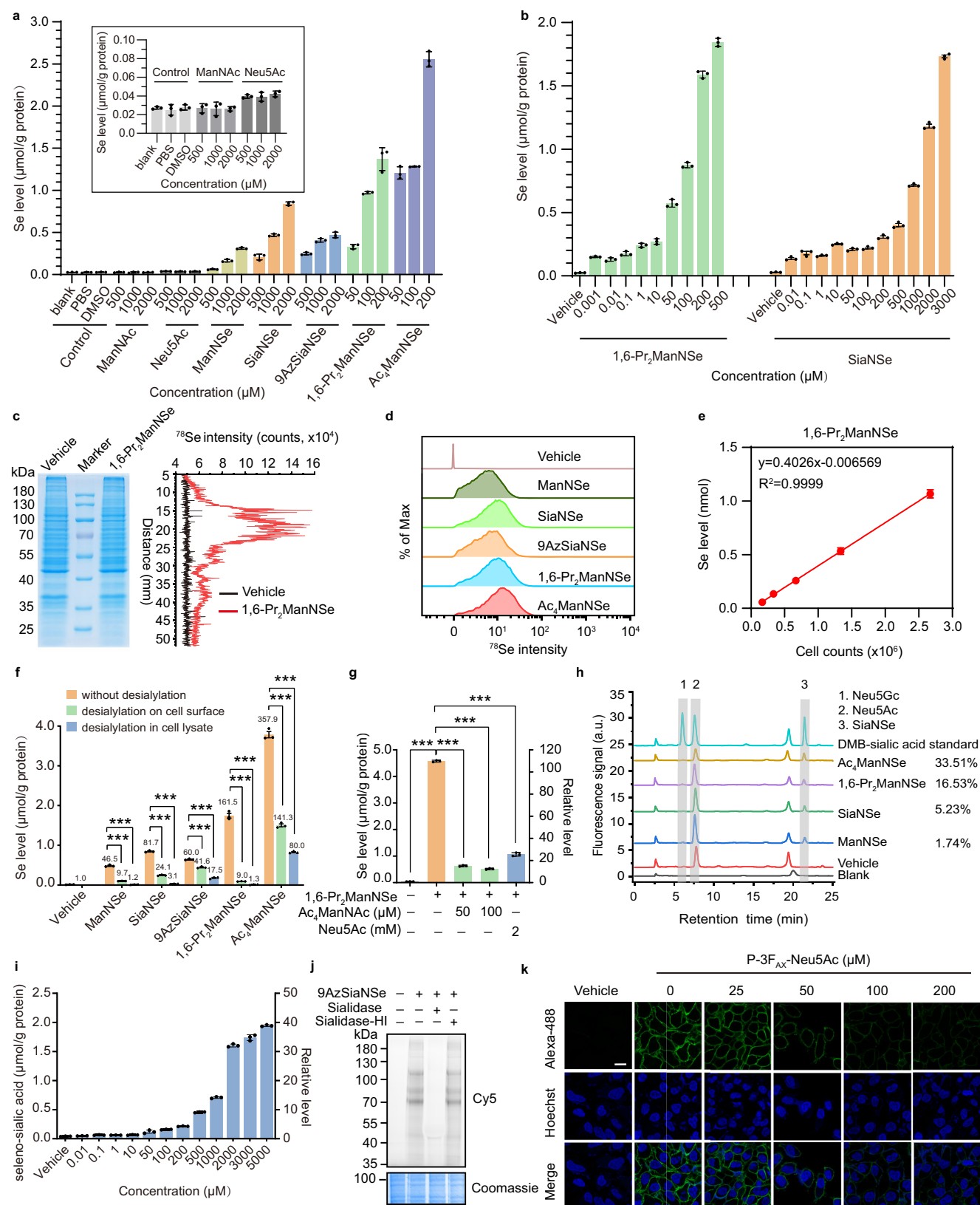

1,6-Pr$_2$GalNSe probe can be metabolically incorporated into cells via the GalNAc salvage pathway (Supplementary Fig. 12)[36,37].

**Dual-modality glycan detection with bifunctional SeMOE probes**
Integration of multimodality within one MOE probe is the next research goal we want to pursue. We envisioned that the facile introduction of

both bioorthogonal functional group (i.e., azide) and ICP-MS responsive element (i.e., selenium) would provide orthogonal signal readouts (fluorochrome vs. elemental analysis) for the study of a more complex biological scenario[38]. To synthesize, we installed azide at the C-9 position of *N*-methylselanylacetyl sialic acid (SiaNSe), to generate bifunctional 9-azido *N*-methylselanylacetyl sialic acid (9AzSiaNSe) (Fig. 1b).

**Fig. 2 | Glycan amount assessment and dual-modality glycan detection by SeMOE. a** Se levels of glycoproteins from HeLa cells treated with respective monosaccharides for 48 h. **b** Se levels of glycoproteins from HeLa cells treated with 1,6-Pr$_2$ManNSe or SiaNSe at varied concentrations for 48 h. **c** $^{78}$Se intensity analysis of HeLa cell proteins. HeLa cells were treated with 200 μM 1,6-Pr$_2$ManNSe or vehicle (PBS) for 48 h. Whole-cell proteins were resolved and transferred on PVDF membrane, and detected by LA-ICP-MS. **d** CyTOF analysis of HeLa cells treated with vehicle (PBS), 2 mM ManNSe, 2 mM SiaNSe, 2 mM 9AzSiaNSe, 200 μM 1,6-Pr$_2$ManNSe and 100 μM Ac$_4$ManNSe for 48 h, respectively. At least 30,000 events were gated and analyzed according to the CyTOF gating strategy (Supplementary Note 2). **e** Linear correlation between the cell count and Se level of 200 μM 1,6-Pr$_2$ManNSe-treated HeLa cells by ICP-MS. **f** Se levels of glycoproteins from respective SeMOE probe-labeled HeLa cells treated with or without sialidase. The number above each column represents the relative Se level compared to the vehicle group. **g** Se levels of glycoproteins from HeLa cells treated with 200 μM 1,6-Pr$_2$ManNSe and metabolic competitors (Ac$_4$ManNAc or Neu5Ac) for 48 h. **h** DMB-derived sialic acid analysis of glycoproteins from HeLa cells treated with vehicle, 2 mM ManNSe, 2 mM SiaNSe, 200 μM 1,6-Pr$_2$ManNSe and 100 μM Ac$_4$ManNAc for 48 h, respectively. Peak 1, 2 and 3 represent Neu5Gc, Neu5Ac and SiaNSe, respectively. The metabolic incorporation rate is calculated as the peak area ratio between peak 3 and the sum of peak 1, 2, and 3. **i** Seleno-sialic acid levels of HeLa cells treated with vehicle (PBS) or 9AzSiaNSe at varied concentrations. **j** In-gel fluorescence scanning of 9AzSiaNSe-labeled HeLa cell proteins treated with sialidase or heat-inactivated sialidase (sialidase-HI). **k** Confocal fluorescence imaging of HeLa cells treated with 2 mM 9AzSiaNSe and the indicated concentrations of P-3F$_{AX}$-Neu5Ac for 48 h. Scale bar:10 μm. Error bars represent mean ± SD of $n$ = 3 independent biological replicates. ***$P$ < 0.0001, one-way ANOVA, post hoc Dunnett's test (**f**, **g**). Source data including exact $P$ values are provided as a Source Data file.

Characterization of azide expression based on flow cytometry, confocal microscopy and in-gel fluorescence all demonstrated that 9AzSiaNSe can tolerate the sialic acid biosynthetic pathway, to be incorporated onto cell-surface sialome, in a time- and dose-dependent manner (Supplementary Fig. 13a–e). Simultaneously, complementary detection of 9AzSiaNSe using the ICP-MS modality not only confirmed the concentration-dependent trend in its metabolism, but also provided meticulous and quantitative information at nanomolar to sub-micromolar concentration range (Fig. 2i). Supportive argumentation such as the treatment with sialidase, with sialic acid biosynthetic inhibitor, or with the native sialic acid as the metabolic antagonizer, all strengthened the fact that 9AzSiaNSe indeed serve as a dual-modal sialic acid analog in SeMOE strategy (Fig. 2j, k, Supplementary Fig. 13f, g).

## SeMOE probe recapitulated the fate of sialic acid in cellular metabolism

In the Roseman-Warren-Bertozzi biosynthetic pathway, exogenous supplied sialic acid precursors (e.g., ManNAc) or sialic acid (e.g., Neu5Ac) were taken up by cells into the cytosol, converted into cytidine-5′-monophospho-$N$-acetylneuraminic acid (CMP-Sia) in the nucleus, trafficked into the Golgi apparatus, where they were utilized by sialyltransferases (Fig. 3a)[39]. Subsequent translocations of the sialoglycoconjugates (e.g., sialoprotein, sialolipid) were then accomplished via the secretory pathway, and ultimately displayed on the cell membrane. We reasoned that the unique feature of SeMOE would open up an avenue to delineate the biodistribution as well as the cellular fate of newly-synthesized sialoglycans. Therefore, we treated HeLa cells with SeMOE probes. The membrane, cytosolic, and nucleus fractions were isolated from whole-cell lysates using a commercialized kit, and both the protein precipitates (Fig. 3b) and the whole-cell lysates (Fig. 3c) in each fraction, were analyzed using ICP-MS. The majority of newly-synthesized sialylated proteins were localized in the cell membrane (51%–81%), showcasing the intrinsic function of sialoproteins as extracellular scaffolds (Fig. 3b). Se signals resided in the nucleus (6%–23%) and cytoplasmic matrix (13%–28%) implying protein-bound SeMOE probes may also de presented in these cellular regions (Fig. 3b), in accordance with the previous observation[13]. Strikingly, for the whole-cell lysate fractionation, a large proportion of seleno-glycan signal was found in the cytosolic fraction (50%–72%) rather than on the plasma membrane, presumably due to the inclusion of SeMOE probes in free or nucleotide-activated form (Fig. 3c). In parallel, we evaluated the cellular distribution of 1,6-Pr$_2$GalNSe-treated cells, and observed the existence of Se signals from nucleus (16%), cytoplasm (27%), and membrane (57%), implying that the 1,6-Pr$_2$GalNSe experienced Gal-NAc salvage pathway and theoretically converted into $O$-GalNSe (on membrane) and $O$-GlcNSe (in nucleocytoplasmic regions), respectively (Supplementary Fig. 14). Alternatively, we individually extract glycoproteins, glycolipids, and glycoRNAs from aliquot of cells treated with 200 μM 1,6-Pr$_2$ManNSe, and measured their relative abundance in ICP-MS after normalization. We monitored a gradual signal shift from the metabolic reservoir to sialoglycoconjugates (Fig. 3d), with varied incorporation rates of selenium-containing sugar probes (slope in Fig. 3e). Substantial Se signals were found in the metabolic flux pool (61%–83%, most likely in the form of selenium-containing metabolites) and were thus not the limiting step in sialoglycoconjugates incorporation (Fig. 3d). Hence, measurement of selenium-containing entities using SeMOE reflected the averaged sialylation turnover rate, and showcased the seleno-glycan expression on major sialome classes.

By exploiting the isotopic signatures of certain elements, innovational techniques such as stable isotope labeling by/with amino acids in cell culture (SILAC)[40] and isotope-targeted glycoproteomics (IsoTaG)[41] have brought quantitative chemical glycoproteomics into the realm of possibility. Traditionally, bioorthogonal conjugation of isotope-containing affinity tags is a must to realize glycopeptide enrichment and isotopic mass pattern prediction. The isotopic envelope pattern of selenium is guaranteed by its six stable isotopes with distinctive distribution ($^{74}$Se, 0.89%; $^{76}$Se, 9.37%; $^{77}$Se, 7.63%; $^{78}$Se, 23.77%; $^{80}$Se, 49.61%; $^{82}$Se, 8.73%)[16]. We postulated that the effective incorporation of SeMOE probes into glycoproteins would achieve direct isotopic pattern prediction for glycopeptides, without the need for secondary tagging (IsoTAG) or costly and time-consuming (SILAC). Consequently, we performed comparative studies using a previously reported computational algorithm, namely selenium-encoded isotopic signature targeted profiling (SESTAR++)[42,43], and a glycan-first glyco-peptide search engine, pGlyco3[44,45], to comprehensively analyze the intact $N$-glycopeptides in cells treated with SeMOE probes (Fig. 3f, Supplementary Data 1–3). The substitution rate (SR) of SeMOE probes (SR = spectrum numbers of $N$-sialoglycopeptides containing selenium / spectrum numbers of total $N$- sialoglycopeptides × 100%), at a range of 0.53%–23.96%, was observed, in concert with the DMB-labeled sialic acid analysis (Fig. 3g, Fig. 2h). Moreover, $N$-sialoglycopeptides containing selenium exhibited typical Se isotopic signature in MS2 (Fig. 3h). We also randomly picked ten glycopeptides across all data sets (acquired via pGlyco3), individually compared their selenium incorporation rate, and found that the incorporation rate is glycopeptide-dependent (Supplementary Fig. 15), which presumably reflects the variations in glycoprotein half-life. Unfortunately, the pattern recognition ability for SESTAR++ is greatly influenced by the microheterogeneity of $N$-glycan, the variation in glycan substitution rate, and the intricate convolution of selenium isotopic distribution. Therefore, identified $N$-sialoglycopeptides containing selenium <3 kDa exhibited expected isotopic patterns in SESTAR++, while $N$-sialogly-copeptides containing selenium >3 kDa showed a better match hit in pGlyco3 (Supplementary Fig. 16). Nevertheless, the combined utilization of SESTAR++ and pGlyco3 mutually improved the identification and annotation of selenium-containing glycopeptides from complex samples. *In toto*, SeMOE grants a pan-scale paradigm to monitor sialylation processes and the sialic acid metabolism.

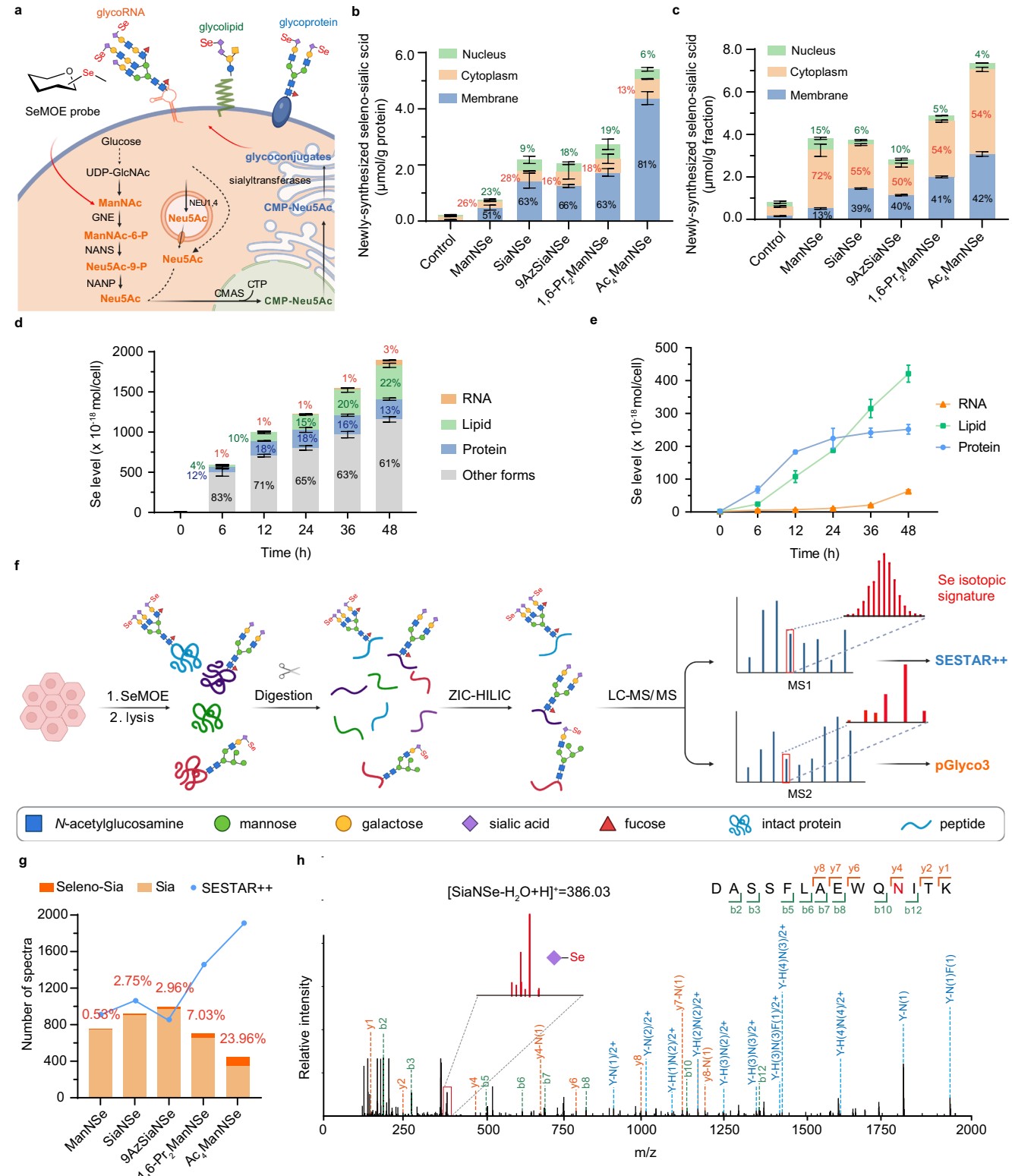

## Tracking glycan transfer in cell-cell interaction and communication using SeMOE

Intercellular crosstalk and interaction orchestrate various biological processes such as organismal development and immune interaction, and are essential for cell viability[11]. Both contact-dependent communication (e.g., via gap junction or cell-surface ligand-receptor interaction) and contact-independent communication (e.g., via secreted molecules signaling) require diverse biomolecules to serve as the information carrier[46,47]. However, the oligosaccharide moiety of

glycoconjugates, or secreted sugar chains that participate in the cell-cell interaction, are unconsciously overlooked due to challenges in glycan characterization and quantification. To demonstrate the application of SeMOE as the evaluation platform for glycans upon cellular interaction, we selected two in vitro models as examples: (a) contact-dependent trogocytosis assay[48], and (b) contact-independent cancer cell-macrophage communication[49].

For the trogocytosis assay, we screened and selected 200 μM 1,6-Pr$_2$ManNSe, or 50 μM Ac$_4$ManNSe as the ideal probe for T lymphocyte

**Fig. 3 | Interpretation of the dynamic cellular fate of newly-synthesized sialic acids by SeMOE. a** Schematic of the sialic acid biosynthesis pathway in cellular metabolism. Figure created with BioRender.com. **b, c**, Newly-synthesized seleno-sialic acid levels in different subcellular proteins (**b**) or subcellular lysates (**c**). **d** Newly-synthesized seleno-sialic acids in different glycoconjugates from MCF-7 cells treated with 200 μM 1,6-Pr₂ManNSe for indicated time. **e** Dynamic assessment of newly-synthesized seleno-sialic acids in glycoconjugates of HeLa cells treated with 200 μM 1,6-Pr₂ManNSe for the indicated time. **f** Workflow of intact *N*-glyco-proteomic and SESTAR++ analysis by SeMOE. *N*-glycopeptides from SeMOE probe-treated HeLa cells were enriched by ZIC-HILIC column and analyzed by LC–MS/MS.

Raw data were analyzed by pGlyco3 or SESTAR++. Figure created with BioRender.com. **g** Numbers of spectra (MS2) for glycopeptides containing natural sialic acid (Sia), or seleno-sialic acid (Seleno-Sia) identified by pGlyco3, and the numbers of spectra (MS1) for peptides matched in SESTAR++ searches. The percentage in orange represents the ratio between seleno-sialic acid-containing peptides and total sialopeptides, calculated according to spectrum numbers. **h** Example of a MS2 for the intact *N*-glycopeptide DASSFLAEWQNITK, showing a seleno-sialic acid with the Se isotopic signature. Hexose (H), HexNAc (N). Error bars represent mean ± SD of *n* = 3 independent biological replicates. Error bars represent mean ± SD of *n* = 3 independent biological replicates. Source data are provided as a Source Data file.

Jurkat cells (i.e., effector cell) (Supplementary Fig. 17). SeMOE probe-treated Jurkat cells were firstly stained by DiI-PE (PE channel in flow cytometry), a cell plasma membrane tracker, and then co-incubated with the target K562 cells at 37 °C for 2 h, with different effector-to-target (E: T) ratios. K562 cells (APC channel in flow cytometry) were next sorted, and the transferred selenium from Jurkat cells was then quantified using ICP-MS (Fig. 4a, Supplementary Note 3). As expected, sialidase treatment on Jurkat cells prior to and during K562 co-incubation significantly reduced the amount of selenium in Jurkat cells, (Supplementary Fig. 16a) further indicating that the majority of Se signals transferred to K562 cells during trogocytosis is originated from covalently-bound seleno-sialoglycans on Jurkat cell surface, although undesired protein S-glyco-modification (Ac₄ManNSe group), or selenium-containing metabolites may also contributed the swapping processes. Impressively, ~1–3 billion (1.6–4.6 femtomole) of seleno-glycans per Jurkat cell, of which 0.6–3% (33.3–59.2 attomole) is scavenged to K562 cells during trogocytosis. In comparison, the classical cluster of differentiation (CD) markers on immune cell only have an antigen density ranging from 10,000 to a million[50–52]. By contrast, selenoproteins have a density of ~2 million copies per cell (Supplementary Note 4)[53], the seleno-sialoglycoconjugates thus have three orders of magnitude over the "background" from endogenous signal. Additionally, the seleno-sialoglycan transfer percentage varied at a range from 0.3% to 10.5%, as E: T ratio and SeMOE probe type varied (Fig. 4b–d, Supplementary Fig. 18b, c). We also demonstrated a similar glycan exchange of incorporated 9AzSiaNSe between Jurkat and K562 cells (Supplementary Fig. 18d–f).

For the tumor cell-macrophage communication studies, we adopted a transwell assay[54] to investigate the glycosylated mediators between cancerous cells and non-activated macrophage (M0) RAW 264.7 cells, using a pulse-chase experiment setting (Fig. 4e, Supplementary Fig. 19). A continuous Se signal downfall was observed from RM1 cell (a carcinoma cell line) within 24 h, which was gradually channeled into seleno-sialoglycan signals observed in RAW 264.7 cells (Fig. 4f, g). We attribute the inflection point after 6 h to RAW 264.7 cellular division (Fig. 4g). Similar sialoglycan transfer trend was observed when 4T1 cell was used instead (Supplementary Fig. 20). We also observed seleno-glycan transfer from 1,6-Pr₂GalNSe-treated cancer cells to RAW 264.7 cells, which exhibited a similar fluctuation pattern with seleno-sialoglycan. (Supplementary Fig. 21). To examine whether seleno-glycans would alter the macrophage state in response to indirect signaling, we characterized the polarization state of RAW 264.7 cells with or without SeMOE probe treatment on RM1 (Fig. 4h, Supplementary Fig. 22). As a result, soluble signaling enacted by gly-cosylated, secreted biomolecules triggered the functional differentia-tion of macrophages from M0 to classically activated macrophages (M1)[55], as evidenced by flow cytometric M1 biomarker staining and the reactive oxidation species (ROS) assay (Fig. 4h, Supplementary Fig. 22a, b). Reciprocally, we explored the incorporation of RM1-released, selenium-containing signal molecules in pre-induced mac-rophage phenotypes M0, M1, and alternatively activated macrophage (M2), which correspond to the normal, pro-inflammatory, and tumor-associated state of macrophages, respectively[56]. M1-type macrophage illustrated better signal molecule incorporation when compared to

M2-type counterparts after proliferation normalization (Fig. 4i, Sup-plementary Fig. 22c), suggesting that seleno-saccharides were readily poised as the signal molecule trackers in distinguishing macrophage phenotypes.

Furthermore, we applied this SeMOE methodology in monitoring the sialic acid transfer between mouse primary granulosa cells (GCs) and oocytes. In mammals, oocyte maturation is closely related to its abundant *N*-glycosylation and its interaction with GCs[57]. Notably, nutrients such as glucose, pyruvate, nucleotides, and amino acids are transferred from GCs to oocytes through gap junctions[58]. Inhibition of gap junction or glycosylation is detrimental to oocyte maturation and GC development[59]. GCs were isolated from ovaries of 3-4 week ICR female mice, treated with 9AzSiaNSe, washed, co-incubated with oocytes for 14 h, and visualized or quantified using confocal micro-scopy or ICP-MS analysis (Fig. 4j). 9AzSiaNSe achieved dual-module characterization of GCs at both protein and cellular levels (Fig. 4k–m, Supplementary Fig. 23a). As expected, glycan transfer between GCs and oocytes was successfully visualized by azide labeling (Fig. 4n, Supplementary Fig. 23b), manifesting the SeMOE applicability in pri-mary cells. Unfortunately, we failed to measure the amount of glycans transferred into oocytes via ICP-MS mode, due to limitation in sample size. This is the first evidence that cell-surface sialoglycans participate in the substance exchange event in granulosa-oocyte interaction. Moreover, the addition of gap junction inhibitor carbenoxolone (CBX) during co-incubation processes resulted in obvious sialic acid transfer inhibition (Fig. 4n)[60]. The above-mentioned data conjointly proved the versatility and universality of the SeMOE strategy in interrogating cellular glycan exchange.

## SeMOE warrants in situ visualization of sialoglycans in biological tissues

We envisioned that coupling SeMOE technology with LA-ICP-MS would fulfill in situ visualization of newly-synthesized glycan biodistribution in biological samples, with both high sensitivity and molecular speci-ficity. Wild-type male BALB/c mice were intraperitoneally injected with Ac₄ManNSe or 1,6-Pr₂ManNSe at various dosages once daily for five days. On the sixth day, mice were euthanized, the major organs were either weighed and homogenized for ICP-MS element analysis or sec-tioned for LA-ICP-MS imaging (Fig. 5a). Incorporation of seleno-sialic acid in mice organs exhibited robust Se signal, with the highest Se signal at 105.32 μg/g in the liver (Fig. 5b). Due to the weight loss of mice upon administration with high dosage SeMOE probes (320 mg/kg) (Supplementary Fig. 24), we attributed the signal intensity decrease in this group to animal toxicity. Since selenium-containing compounds are typically metabolized into seleno-amino acids in vivo, we asked whether administration of the SeMOE probe would perturb the seleno-amino acid levels, i.e., transformed into other known selenium-containing biological entities rather than in the form of selenium-containing sugar/glycan. So we performed selenium speciation on selenocysteine (SeCys₂), Se-methyl-L-selenocysteine (MeSeCys), sele-nomethionine (SeMet), selenate [SeO₄²⁻], and selenite [SeO₃²⁻], five compounds associated with selenoprotein conversion. Major organs administered with or without SeMOE probes displayed near-identical selenium speciation, suggesting that injection of seleno-sialoglycans

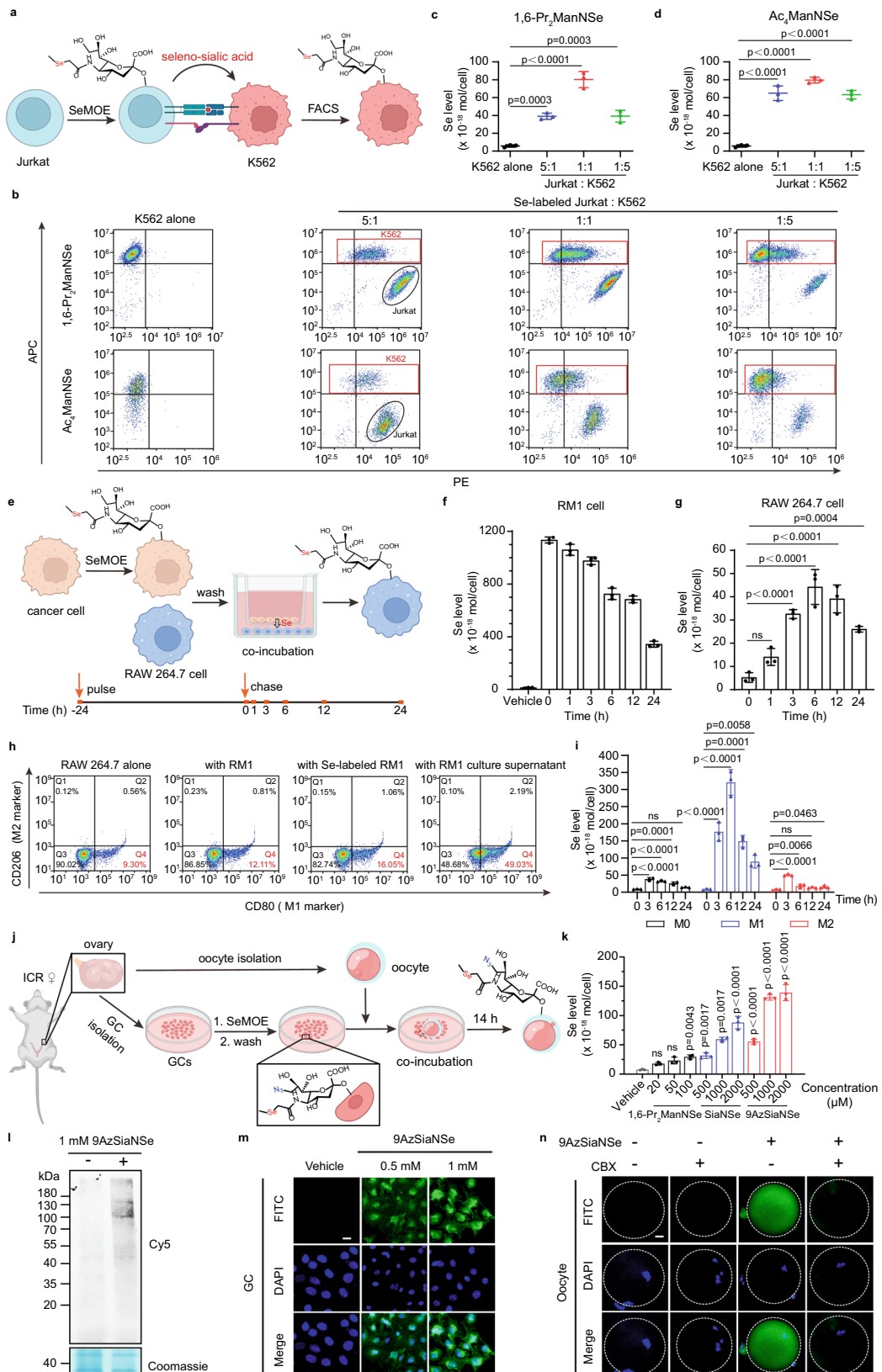

will not significantly affect the selenoprotein constituents in mice (Supplementary Fig. 25 a–d). The only exception was observed in the liver, with a signal increase for SeCys$_2$, MeSeCys and SeO$_3^{2-}$, probably because the liver is mostly involved in the digestive product metabolism (Supplementary Fig. 25b, f). Interestingly, the brain also achieved distinguishable seleno-sialoglycan S/N contrast, an organ that

previously prevented the entrance of per-$O$-acetylated ManNAz due to blood-brain-barrier (BBB)[61,62]. It would be captivating to investigate how the per-$O$-acetylated selenium-containing counterpart allows for the movement across BBB. Thin tissue sections on major mice organs were investigated with respect to the elemental distribution of $^{78}$Se, $^{80}$Se, $^{32}$S, $^{56}$Fe, $^{63}$Cu and $^{66}$Zn (Fig. 5c, Supplementary Fig. 26a, b).

**Fig. 4 | Glycan transfer tracking in various cell-cell interactions. a** Schematic of glycan transfer during Jurkat-K562 trogocytosis. SeMOE probe-treated Jurkat cells were stained by the cell membrane staining tracker Dil-PE, washed and then co-incubated with Did-APC-stained K562 cells at 37°C for 2 h, followed by sorting and ICP-MS analysis of K562 cells. **b** Flow cytometry analysis of Jurkat-K562 trogocytosis. **c, d** Seleno-sialic acid transfer from 1,6-Pr$_2$ManNSe (**c**) or Ac$_4$ManNSe-treated (**d**) Jurkat cells to K562 cells. Jurkat cells and K562 cells were gated and sorted according to the gating strategy (Supplementary Note 3). **e** Schematic of cancer cell-RAW 264.7 communication. SeMOE probe-treated cancer cells were washed, and co-incubated with RAW 264.7 cells in a 0.4 μm-sized transwell culture system for varied time, followed by cell counting and ICP-MS analysis of cancer cells and RAW 264.7 cells, respectively. **f** Se levels of Se-labeled RM1 cells during co-incubation with RAW 264.7 cells. **g** Seleno-sialic acid transfer from RM1 to RAW 264.7 cells. **h** M1 polarization of RAW 264.7 cell induced by RM1 cell or RM1 cell

culture supernatant. **i** Seleno-sialic acid transfer from RM1 to RAW 264.7 cells in different polarization states. **j** Schematic of GC labeling and co-incubation with oocytes. SeMOE probe-treated freshly isolated GCs were washed, and co-incubated with oocytes for 14 h, followed by confocal imaging or ICP-MS analysis. **k** Se levels of GCs treated with respective SeMOE probes at indicated concentrations for 48 h. **l** In-gel fluorescence scanning of GCs treated with or without 1 mM 9AzSiaNSe for 48 h, followed by reaction with alkyne-Cy5. **m** Confocal fluorescence imaging of 9AzSiaNSe-labeled GCs. Scale bar: 20 μm. **n** Confocal fluorescence imaging of oocytes after co-incubation with 9AzSiaNSe-labeled GCs with or without 100 μM CBX. Scale bar: 20 μm. Error bars represent mean ± SD of *n* = 3 independent biological replicates. ns, not significant; all significant *P* values are indicated (one-way ANOVA followed by post hoc Dunnett's test). Source data including exact *P* values are provided as a Source Data file. **a**, **e**, **j** created with BioRender.com.

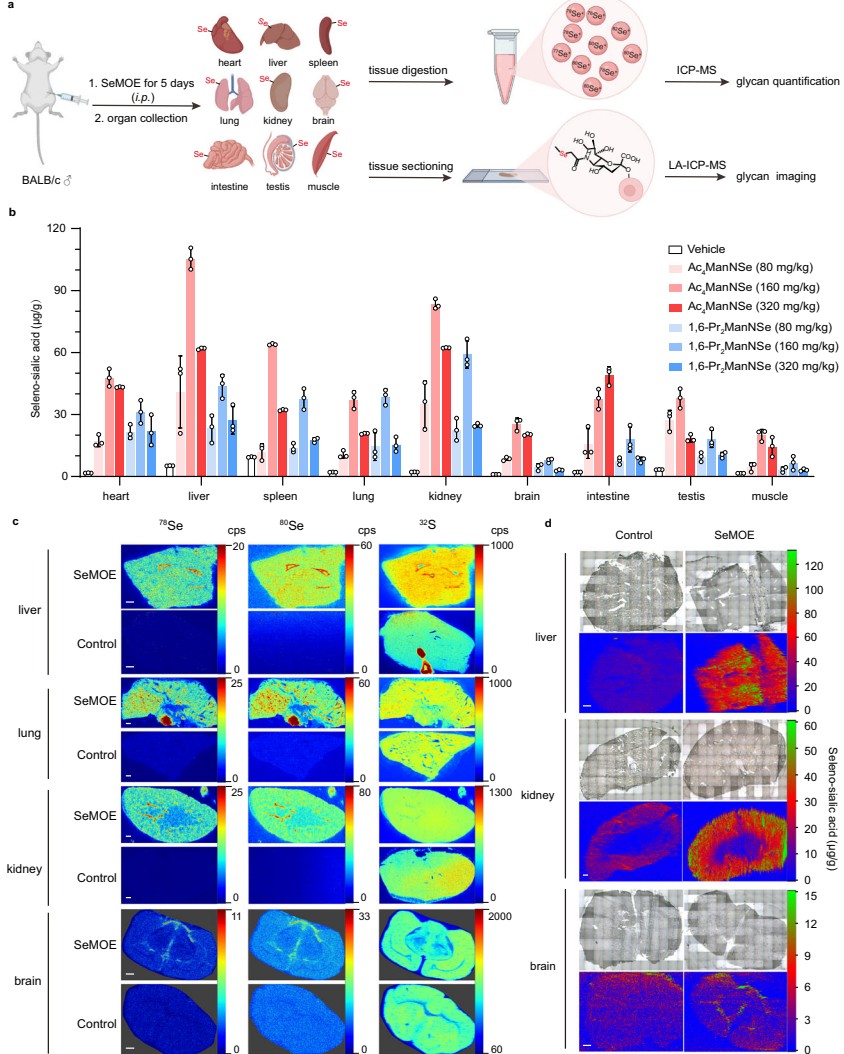

**Fig. 5 | In situ evaluation and visualization of mouse tissue sialoglycans. a** Schematic of SeMOE in vivo and in situ tissue glycan imaging and measurement. Male BALB/c mice were once daily, intraperitoneally injected with SeMOE probes at varied dosages, while control mice received 70% DMSO (vehicle) alone. On day 5, the mice were euthanized and perfused. The organs were digested for ICP-MS analysis, or sectioned for LA-ICP-MS imaging. Figure created with BioRender.com. **b** Seleno-sialic acid levels of tissues from the mice treated with Ac$_4$ManNSe or 1,6-

Pr$_2$ManNSe at the indicated concentration. Each dot represents an individual mouse. *n* = 3 mice per group. **c** In situ sialoglycan imaging of mouse liver, lung, kidney, and brain by LA-ICP-MS. Scale bar: 400 μm. **d** In situ sialoglycan quantification of mouse liver, kidney and brain by LA-ICP-MS. Scale bar: 400 μm. Error bars represent mean ± SD of *n* = 3 independent biological replicates. Source data are provided as a Source Data file.

Strikingly, newly-synthesized sialoglycans showed anatomical structure-specific biodistribution in major organs. For example, intensified selenium signals were found in the liver and lung lobes, the tracheal junction, surrounding tissues of the kidney, the contour of the thalamus, hippocampus and corpus callosum in the brain (Fig. 5c). What's more, in situ spatial imaging and quantification of seleno-glycan was realized and calibrated by using the Se standard substance (Fig. 5d, Supplementary Fig. 26c). To sum up, the label-free SeMOE, in tandem with LA-ICP-MS, effectively circumvent the secondary labeling used in classical MOE, and can be portrayed as an innovative methodology to visualize and evaluate the newly-synthesized sialoglycans in animals with element standards.

## Discussion

Revolutionary MOE has emerged as an organic chemistry-based, two-step methodology in the perturbation, profiling and quantification of newly-synthesized glycans in living organisms. The utility of mono-saccharide precursors/analogs tagged with selenoethers provides an elemental handle rather than a chemical one, to allow for convenient monitoring by elemental mass spectrometry. The SeMOE strategy thus is poised as a simplified, label-free, sensitive yet versatile method for the quantification of newly-synthesized non-natural seleno-glycans. The ICP-MS source effectively eliminates the varied response factors between different glycans and glycoconjugate classes. Besides, the unique isotopic signature of selenium allows for convenient glyco-proteomic applications. Of note, SeMOE is applicable from the molecular level to the tissue level. For LA-ICP-MS imaging, the rapid scanning, wide dynamic and mass range are ideally suited to measuring multiple unknown concentrations and isotope ratios in samples with minimal preparation.

We have preliminarily investigated the potential biological effects of SeMOE probes on tumor cells, in the aspect of ROS generation, cell migration and cell apoptosis. Intriguingly, SeMOE probes facilitated the rapid ROS generation in a variety of cancer cell lines rather than in normal cells (Supplementary Fig. 27, Supplementary Fig. 28a–d). Control experiment with vehicle, or equimolar $Na_2SeO_3$, and corresponding monosaccharide ensured that the increase in ROS production is an intrinsic property from SeMOE probes, presumably related to their redox properties (Supplementary Fig. 25e). Selenium-containing entities such as 1,4-anhydro-4-seleno-D-talitol (SeTal)[63], selenadiazole derivatives (SeDs)[64], and anti-inflammatory drug Ebselen[65] all provided therapeutic opportunities. Inhibition on cell migration (Supplementary Fig. 29), as well as promotion of cell early apoptosis (Supplementary Fig. 30) were demonstrated. Overall, the precise function of SeMOE probes remains equivocal, and will be benefited from its mechanistic elucidation.

While elemental detection via ICP-MS is robust and quantitative, the experimental setting is not as common as fluorescence-based instruments in a general laboratory. However, in comparison with nuclear analytical techniques such as gamma spectroscopy, ICP-MS instrument stands out as a cost-effective and user-friendly alternative. Structural information about the detected biomolecules needs additional means. The LA-ICP-MS also suffers from image resolution (μm range in LA-ICP-MS vs. nm resolution in super-resolution fluorescent microscopy) and sample destruction, making it difficult to monitor dynamic processes in living systems. For the application of SeMOE methodology, the following limitations should be taken into consideration: (1) incorporation of these probes into other biochemical pathways, for instance, early reports on the mammalian GalNAc salvage pathway may convert the activated form of GalNSe to its *N*-methylselanylacetylglucosamine (GlcNSe) counterpart (i.e., UDP-GlcNSe). Because they are general nucleotide sugar donors, and thus can be readily incorporated into *N*-linked-, mucin-type *O*-linked-, and *O*-GlcNAcylated-glycoproteins[66]. ManNSe is also amenable to mono-saccharide interconversion toward UDP-GalNSe or UDP-GlcNSe, as a substrate for epimerases[67]. (2) Other selenium-containing entities,

rather than selenium-containing glycans (e.g., selenoproteins, protein S-glyco-modification) may, at least in part, play a role in these measurements. Even though the same limitation exists in classical MOE, some drawbacks can be effectively circumvented by second-step labeling. For example, reactive fluorochromes only "light up" the cell-surface azides due to accessibility, the cytosolic azide (in all possible speciations) therefore exists yet remains "invisible". Therefore, cautions, or even meticulous and elaborate calibration/normalization are needed in such data interpretation, to transfer the quantitative feature of ICP-MS into quantification toward glycans. Nevertheless, the inter-disciplinary nature of classic MOE is further extended by SeMOE.

The bifunctional seleno-sialic acid enables simultaneous visualization and sialoglycans measurement using different modalities. Information lost in static mass spectrometric analysis can be well compensated by the dynamic feature of fluorescence imaging or gly-coprotein enrichment. However, enrichment of labeled proteins in this work still requires the bioorthogonal azide group. Integration of algorithms that can distinguish selenium-isotope signals without the need for enrichment is promising. Furthermore, SeMOE strategy is readily expanded to other metabolic chemical reporters, including but not limited to glycans, lipids, proteins, and other post-translational modifications (e.g., acetylation). Methodological adaption from che-moenzymatic glycan labeling[68] or Se-click reaction[69] will integrate the enrichment and quantification of glycoconjugates, which doubtlessly provide room for methodological improvement. In addition, if the SeMOE analogs were synthesized as Se-isotopically enriched derivatives, these probes are readily distinguished by ICP-MS and may find broader applications in temporal studies[70]. These concepts are being pursued in our laboratory. To sum up, we are optimistic that the introduction of SeMOE, an innovative and simplified method, into chemical glycobiology, will augment a more accurate interpretation of glycans in biophysiological and therapeutic events.

## Methods

The work performed at Nanjing University complies with all relevant ethical regulations for animal testing and research. All animal experimental protocols were approved by the Nanjing University Institutional Animal Care and Use Committee.

### Cell lines and mice

Cell lines HeLa (CCL-2), A549 (CRM-CCL-185), 293 T (CRL-3216), Hep G2 (HB-8065), SW620 (CCL-227), 4T1 (CRL-2539), CT26 (CRL-2638), B16-F10 (CRL-6475), RAW 264.7 (TIB-71), MCF-7 (HTB-22), MCF 10 A (CRL-10317), Jurkat E6-1 (TIB-152), K562 (CCL-243), RM1 (CRL-3310), T24 (HTB-4), HMC3 (CRL-3304), MRC-5 (CCL-171), SV-HUV-1 (CRL-9520) and HK-2 (CRL-2190) were all purchased from the American Type Culture Collection (ATCC). HCCC-9810 cells (TCHu 17) were purchased from the Institute of Biochemistry and Cell Biology, Shanghai Institutes for Biological Sciences, Chinese Academy of Sciences, Shanghai, China. HeLa, A549, 293 T, Hep G2, SW620, 4T1, CT26, B16-F10 and RAW 264.7 cells were cultured in DMEM (Gibco) supplemented with 10% (v/v) fetal bovine serum (FBS) (Gibco), 100 U ml$^{-1}$ penicillin and 100 μg ml$^{-1}$ streptomycin (Gibco). MCF-7, MCF 10 A, HCCC-9810, Jurkat E6-1, K562, RM1 and T24 cells were cultured in RPMI 1640 (Gibco) supplemented with 10% (v/v) FBS (Gibco), 100 U ml$^{-1}$ penicillin and 100 μg ml$^{-1}$ streptomycin (Gibco). HMC3 and MRC-5 cells were cultured in α-MEM (Gibco) supplemented with 10% (v/v) FBS (Gibco), 100 U ml$^{-1}$ penicillin and 100 μg ml$^{-1}$ streptomycin (Gibco). SV-HUV-1 and HK-2 were cultured in DMEM/F-12 (Gibco) supplemented with 10% (v/v) FBS (Gibco), 100 U ml$^{-1}$ penicillin and 100 μg ml$^{-1}$ streptomycin (Gibco). Cultures were grown in T25 or T75 flasks (Thermo Fisher) and maintained at 37 °C with 5% $CO_2$. All cell lines were verified to be free of mycoplasma. Wild-type (WT) BALB/c and ICR mice were purchased from Gem-Pharmatech Co. Ltd., Nanjing, China, and kept under specific pathogen-free (SPF) conditions with free access to standard food and water. All

animal experiments were approved by the Nanjing University Institutional Animal Care and Use Committee.

## Antibodies and reagents

Antibodies and dye included streptavidin-Alexa Fluor 488 (Thermo, S32354, 1:2,000), streptavidin-Alexa Fluor 555 (Thermo, S32355, 1:2,000), Hoechst 33342 (Thermo, S32354, 1:1,000), NucBlue™ Fixed Cell ReadyProbes™ Reagent (DAPI) (Thermo, R37606), Did-APC (Beyotime, C1039, 1 mM stock solution, 1:400), Dil-PE (Beyotime, C1991, 1:400), anti-mouse CD80-PE (Biolegend, 104707, 1:200), anti-mouse CD206-APC (Biolegend, 141707, 1:200), biotinylated Sambucus Nigra Lectin (SNA)(VectorLabs, B-1305-2, 2 mg/mL stock solution, 1:400), biotinylated Maackia Amurensis Lectin II (MAL II)(VectorLabs, B-1265-1, 1 mg/mL stock solution, 1:200). Alkyne-biotin (catalog no. 1137), alkyne AZDye 488 (catalog no. 1277), alkyne-Cy5 (catalog no. TA116), alkyne AZDye 647 (catalog no. 1301), and 2-(4-((bis((1-tert-butyl-1H-1,2,3-triazol-4-yl)methyl)amino)methyl)-1H-1,2,3-triazol-1-yl) acetic acid (BTTAA) (catalog no. 1236) were purchased from Click Chemistry Tools. Iodoacetamide and 4,5-methylenedioxy-1,2-phenylenediamine dihydrochloride (DMB) were purchased from Sigma Aldrich. P-3F$_{AX}$-Neu5Ac (catalog no.5760) was purchased from Tocris Bioscience. LPS and IL-4 were purchased from Proteintech. Neuraminidase A (sialidase) from *Streptococcus pneumoniae* (NanA) was expressed and purified as previously described[32]. PNGase F was purchased from New England Biolabs (NEB). All organic agents were of at least analytic grade, obtained from commercial suppliers and used without further purification. *N*-acetylmannosamine (ManNAc), *N*-acetylneuraminic acid (Neu5Ac), *N*-glycolylneuraminic acid (Neu5Gc) and per-*O*-acetylated *N*-acetylmannosamine (Ac$_4$ManNAc) were purchased from Carbosynth. *N*-azidoacetylmannosamine (ManNAz), *N*-azidoacetylneuraminic acid (SiaNAz), per-*O*-acetylated *N*-azidoacetylmannosamine (Ac$_4$ManNAz), and 1,6-di-*O*-propionyl-*N*-azidoacetylmannosamine (1,6-Pr$_2$ManNAz) were synthesized as reported[29,71,72]. Reagents for CyTOF analysis were purchased from Fluidigm. Dithiothreitol (DTT) used in LC−MS/MS analysis was purchased from J&K Scientific. Sequencing-grade modified trypsin was purchased from Promega. Se standard solution (100 ppm) was obtained from the National Sharing Platform for Reference Materials, China.

## SeMOE in vitro and in vivo

Ac$_4$ManNSe and Ac$_4$GalNSe were made to 100 mM stock solution in sterile dimethyl sulfoxide (DMSO). 1,6-Pr$_2$ManNSe and 1,6-Pr$_2$GalNSe were made to 200 mM stock solution in sterile PBS. ManNSe, SiaNSe and 9AzSiaNSe were made to 500 mM stock solution in sterile PBS. All stock solutions were stored at −20°C. For SeMOE treatment in vitro, cells were incubated with complete medium supplemented with indicated SeMOE probes at varied concentrations at 37 °C for 48 h when they reached approximately 30% confluence. After that, cells were trypsinized, collected, washed three times with PBS, counted by the automated cell counter (Invitrogen), and then used for further fraction isolation or ICP-MS analysis.

For SeMOE in vivo, Ac$_4$ManNSe or 1,6-Pr$_2$ManNSe was diluted to 20 mg/mL in 70% DMSO (v/v) and in 0.9% NaCl, respectively. Mice were randomly selected before further treatment. Male BALB/c mice (8–10 weeks old) were once daily, intraperitoneally injected with Ac$_4$ManNSe (160 mg SeMOE probes/kg/day), while control mice received the corresponding vehicle alone. On day 5, the mice were euthanized. The mice were perfused with PBS, and major organs were collected and washed three times with PBS, frozen in liquid nitrogen-isopentane, and stored at −80 °C until used for nitric acid digestion or tissue section preparation.

## Se speciation analysis

The mouse tissues (15 mg) acquired from mice with or without SeMOE probe treatment were immersed in 1 mL Millipore water supplemented with proteinase K (4 mg/mL) and trypsin (4 mg/mL) were homogenized at 37 °C for 24 h. The water bath was sonicated for 5 min every 2 h. After enzymatic hydrolysis, the supernatant liquid was collected by centrifugation (12,000 *g*, 10 min, 4 °C), filtered by a 0.22 μm filter and detected by HPLC-ICP-MS with PRP-X-100 anion exchange HPLC column (4.1 × 250 mm, 10 μm). The mobile phase was 0.01 M citric acid solution (pH 4.5) with a flow rate of 0.6 mL/min. The standards for Se-related metabolic products including SeO$_3^{2-}$, selenocystine (SeCys$_2$), methylseleno-cysteine (MeSeCys), and selenomethionine (SeMet) were used for seleno-amino acids quantification.

## Isolation and validation of subcellular fractionations

For analysis of total seleno-glycans in subcellular fractions, the Membrane, Cytosol and Nuclear Protein Extraction Kit (KeyGEN BioTECH, KGBSP002, KGBSP002) was used to isolate subcellular fractions according to the manufacturer's instructions. Briefly, 1 × 10$^7$ of SeMOE probe-treated MCF-7 cells were scraped off and collected. Cell pellets were suspended in 1 mL Extraction Buffer A (supplemented with protease inhibitor) and lysed in a glass homogenizer, followed by incubation for 20 min at 4 °C and centrifugation (15,000 *g*, 10 min, 4 °C). The supernatant was collected as cytoplasmic fraction. After washed twice with Extraction Buffer A, the remaining pellets were resuspended in 500 μL Extraction Buffer B (supplemented with protease inhibitor), incubated for 10 min at 4 °C and centrifuged at 15,000 *g* for 10 min at 4 °C. This supernatant was collected as nuclear fraction. Then 500 μL Extraction Buffer C (supplemented with protease inhibitor) was subsequently added to the remaining pellets, followed by incubation for 10 min at 4 °C and centrifugation (15,000 *g*, 10 min, 4 °C). The supernatant was collected as membrane fraction. Protein concentrations of isolated subcellular fractions above were determined by BCA protein assay, lyophilized, and analyzed by ICP-MS. Se concentration was normalized by sample weight.

For analysis of seleno-glycans covalently bound to proteins, proteins in isolated subcellular fractions above were precipitated by three volumes of alcohol at −80°C overnight. The precipitated proteins were collected by centrifugation (12,000 *g*, 4 °C, 10 min) and washed three times with 75% ethanol (v/v), and digested in nitric acid and hydrogen peroxide (2:1, v/v) at room temperature overnight, and analyzed by ICP-MS. Se concentration was normalized by protein amount.

## Extraction of protein, lipid and RNA

MCF-7 cells were treated with or without 200 μM 1,6-Pr$_2$ManNSe for varied time (6 h, 12 h, 24 h, 36 h, 48 h). After that, 1.5 × 10$^7$ cells were trypsinized, washed three times with PBS, counted by the automated cell counter (Invitrogen), and divided into three equal parts for protein, lipid and RNA extraction, respectively. All experiments were done with three replicates.

For protein isolation, 5 × 10$^6$ cells were suspended in 300 μL cold RIPA buffer (1% Nonidet P-40 (v/v), 1% sodium deoxycholate (w/v), 150 mM NaCl, 0.1% SDS (w/v), 50 mM triethanolamine and EDTA-free Roche protease inhibitor, pH 7.4), lysed by sonication, and centrifuged at 12,000 *g* for 10 min at 4 °C for removal of debris. The protein concentration was determined by BCA protein assay. Then proteins were precipitated by adding into a MeOH/CHCl$_3$ mixture (aqueous phase/MeOH/CHCl$_3$ = 4:4:1, v/v/v), the isolated proteins were collected by centrifugation (20,000 *g*, 10 min, 4 °C), and washed three times with cold methanol. To extract cell lipids, 5 × 10$^6$ cells were suspended in 150 μL cold water,lysed by sonication and mixed with 400 μL methanol and 200 μL chloroform, then the mixture was stirred at room temperature for 30 min. After stirring, phase separation was initiated by adding 200 μL chloroform, followed by centrifugation (2800 *g*, 10 min, room temperature). The subnatant fraction (containing lipids) was carefully transferred into a fresh tube. As for RNA extraction, RNA samples were prepared as previously described[73], and the RNA concentrations were determined by Nanodrop (Thermo Scientific). All samples of protein,

lipid and RNA isolated above were lyophilized overnight, and stored at −80 °C until digestion by nitric acid for ICP-MS analysis.

## Enzymatic treatment of live cells and protein samples

Neuraminidase A (sialidase) from *Streptococcus pneumoniae* (NanA), was expressed and purified as previously described[32]. Expressed sialidase was concentrated to 7.0 mg/mL. In the live cell experiment, $5 \times 10^6$ HeLa cells were collected by 10 mM ethylenediaminetetracetic acid disodium (EDTA Na2) in PBS, washed three times with PBS, and resuspended in sialidase buffer (500 µL HBSS buffer, 10 µL sialidase and 2.5 µL of 1 M MgCl$_2$) and incubated at 37 °C for 30 min. After that, cells were washed three times with PBS and stored at -80 °C for further digestion and ICP-MS analysis.

In the protein lysate experiment, whole-cell proteins were isolated as described above, and adjusted to 2 mg/mL using BCA assay. Protein samples were digested with sialidase and PNGase F, respectively. For sialidase-treated protein samples, 200 µg of glycoprotein (100 µL) from HeLa cells were mixed with 0.5 µL of 1 M MgCl$_2$ and 10 µL sialidase, and reacted for 2 h at 37 °C. For PNGase F treatment, 100 µg of glycoprotein (50 µL) from HeLa cells was mixed with 6 µL 10 × denaturing buffer and 4 µL deionized water, and denatured for 10 min at 100 °C. Then, 10 µL GlycoBuffer 2 (10 ×) (NEB), 10 µL 10% NP-40 (NEB), 10 µL PNGase F (NEB) and 10 µL water were added into the reaction system. PNGase F cleavage occurred for 2 h at 37 °C. After enzymatic treatment, cell proteins were precipitated by adding into a MeOH/CHCl$_3$ mixture (aqueous phase/MeOH/CHCl$_3$ = 4:4:1, v/v/v). The precipitated proteins were collected by centrifugation (20,000 $g$, 10 min, 4 °C), and washed three times with cold methanol. Above enzyme-treated live cell and protein samples were digested in nitric acid and hydrogen peroxide (2:1, v/v) at room temperature overnight, and analyzed by ICP-MS.

## Inhibition and metabolic competition assays

For inhibition of sialyltransferases, HeLa cells were treated with 0, 25, 50, 100 or 200 µM P-3F$_{AX}$-Neu5Ac (Tocris), along with indicated concentration of SeMOE probes for 48 h. In the metabolic competition experiment, the natural monosaccharides, as well as their azidocounterparts, were used along with SeMOE probes for metabolic competition. HeLa cells were treated with these carbohydrate precursors/analogs at varied ratios as indicated for 48 h. The cells were subjected to cell fluorescence imaging or ICP-MS analysis afterward.

## Acid digestion and ICP-MS quantitative analysis

After SeMOE treatment, the cells were collected, washed, and counted as described above. For sample analysis of major biomacromolecules, protein, lipid and RNA extracts were collected as described above. To prepare acid digestion samples, $1 \times 10^5$-$5 \times 10^6$ cells, or 0.01-1.0 mg biomacromolecule samples were mixed with 44 µL nitric acid and 22 µL 30% hydrogen peroxide, and digested at room temperature overnight. After digestion, Millipore water was added dropwise until a final nitric acid concentration of 2%, with 1.5 mL as the final volume.

For mouse tissue samples, mouse organs were perfused, washed, frozen and stored as described above. Mouse organs were cut into pieces, lyophilized overnight and accurately weighed before mixed with 2 mL nitric acid and 1 mL 30% hydrogen peroxide. Mouse tissues (~20 mg) were digested at 150 °C for ~1 h until they became clear solutions. After that, digested solutions were heated at 100 °C to eliminate excess nitric acid, and 2% nitric acid was then added for samples to redissolve in 5.0 mL as the final volume. The sample solutions were stored at 4 °C until ICP-MS analysis. Se standard solutions with different concentrations (0, 1, 10, 50, 100, 333, and 500 ppb), as well as sample solutions, were analyzed by solution nebulization ICP-MS (PerkinElmer, NexION 300D, USA). The parameters of ICP-MS solution analysis are shown in Supplementary Table 1. The isotopes [78]Se were adopted for analysis, and the calibration curves of Se of

ICP-MS solution analysis are shown in Supplementary Fig. 3. Se concentrations were quantified according to the standard curve. Se or seleno-glycan levels were normalized according to the cell number or sample weight.

## Selenium analysis of proteins on the PVDF membrane by LA-ICP-MS

HeLa cells were treated with PBS or 200 µM 1,6-Pr$_2$ManNSe for 48 h and lysed. Proteins were separated by SDS-PAGE and transferred to a PVDF membrane. The PVDF membrane was washed three times with deionized H$_2$O, dried at room temperature, and subjected to LA-ICP-MS for linear scan analysis. The parameters and conditions of the laser ablation system and ICP-MS are shown in Supplementary Table 2.

## CyTOF analysis

Cells were seeded into 6-well plates ($5 \times 10^5$ cells/well), treated with vehicle or indicated SeMOE probes at 37 °C for 48 h. After that, $2 \times 10^6$ cells were collected, washed twice with 1 mL PBS (without Ca$^{2+}$, Mg$^{2+}$), and incubated with 1 mL 0.5 µM cisplatin (Fluidigm) in PBS (without Ca$^{2+}$, Mg$^{2+}$) at room temperature for 2 min, followed by centrifugation at 400 $g$ for 5 min at room temperature. Cell pellets were washed twice with 2 mL CyFACS (Fluidigm), incubated with 125 nM Intercellular-Ir2 (Fluidigm) in FIX and PERM Buffer (Fluidigm) at 4 °C overnight, and then centrifuged at 800 $g$ for 5 min and discard the supernatant. After that, cells were washed with 1 mL CyFACS and 1 mL Ultra-Water (Fluidigm), resuspended in Cell Acquisition Solution (Fluidigm) containing 10% EQ Four Calibration Beads (Fluidigm). Sample concentration was adjusted to acquire at a rate of 200–300 events/s using a wide-bore (WB) injector on a CyTOF-Helios instrument (Fluidigm). The CyTOF data were exported as *.fcs files. Pre-gating was performed in FlowJo software (BD Biosciences) to filter the data to consist only of live, intact, single cells. At least 30,000 events were gated and analyzed according to the CyTOF gating strategy (Supplementary Note 2).

## DMB assay for sialic acid detection

HeLa cells were seeded into 6-well plates ($5 \times 10^5$ cells/well), and incubated with indicated SeMOE probes at 37 °C for 48 h. After that, about $2 \times 10^6$ cells were harvested, resuspended in 100 µL RIPA buffer (1% Nonidet P-40 (v/v), 1% sodium deoxycholate (w/v), 150 mM NaCl, 0.1% SDS (w/v), 50 mM triethanolamine and EDTA-free Roche protease inhibitor, pH 7.4) and incubated at 4 °C for 30 min, followed by centrifugation (12,000 $g$, 10 min, 4 °C) to remove cell debris. Three volumes of cold alcohol were added to precipitate the cellular proteins, and the mixture was kept at −80°C overnight. Protein precipitates were collected by centrifugation (12,000 $g$ for 10 min at 4 °C) and washed four times with 75% alcohol (v/v). Protein samples (~400 µg) or sialic acid standard were then dispersed in 70 µL 2 M acetic acid and incubated at 80 °C for 3 h, cooled on ice, and filtered through 10 kDa MWCO filters (Millipore) by centrifugation (15,000 $g$, 15 min, 4 °C). For sialic acid DMB derivatization, deionized H$_2$O, DMB, β-mercaptoethanol, and Na$_2$S$_2$O$_4$ were added to the filtrate above to make 100 µL as the final volume and adjust the final concentration of acetic acid, DMB, β-mercaptoethanol, Na$_2$S$_2$O$_4$ at 1.4 mM, 7 mM, 0.75 M, and 18 mM, respectively. Derivatization was performed in the dark at 50 °C for 2 h, cooled on ice for 10 min, and neutralized with NaOH solution (0.2 M, 25 µL). Samples were diluted 500× with Millipore water and analyzed by RP-HPLC (Aglient 1260, XDB-C18 column, 5 µm, 4.6 × 250 mm) with a fluorescence detector ($\lambda_{ex} = 373$ nm, $\lambda_{em} = 448$ nm). The flow rate was 0.8 mL/min and the elution gradient was: T (0 min) 84% H$_2$O + 9% CH$_3$CN + 7% CH$_3$OH; T (14 min) 84% H$_2$O + 9% CH$_3$CN + 7% CH$_3$OH; T (22 min) 64% H$_2$O + 18% CH$_3$CN + 18% CH$_3$OH; T (28 min) 64% H$_2$O + 18% CH$_3$CN + 18% CH$_3$OH; T (29 min) 84% H$_2$O + 9% CH$_3$CN + 7% CH$_3$OH; T (30 min) 84% H$_2$O + 9% CH$_3$CN + 7% CH$_3$OH. The incorporation efficiency of seleno-sialic acid into glycoproteins was quantified by the integration of peak areas.

## S-glyco-modification

$3 \times 10^6$ of HeLa cells were washed three times with PBS, lysed in cold PBS by sonication, and the debris was removed by centrifugation (20,000 $g$, 10 min, 4 °C). 50 µL cell lysates (2 mg/mL) were incubated with indicated monosaccharides at varied concentrations at 37 °C for 2 h. After precipitation by adding 150 µL methanol, 37.5 µL chloroform and 100 µL Millipore water, the aqueous phase was removed by centrifugation (20,000 $g$, 5 min, 4 °C). The precipitated proteins were washed twice with cold methanol, stored at −80 °C until nitric acid digestion and ICP-MS analysis, or resuspended in 50 µL 0.4% SDS (w/v) in PBS for click reaction and in-gel fluorescence imaging.

## In-gel fluorescence scanning

In the click reaction setting of cell lysates, 100 µM alkyne-Cy5 or alkyne-Cy3, premixed BTTAA-CuSO4 complex solution (50 µM CuSO4, BTTAA/CuSO4 2:1) and 2.5 mM freshly prepared sodium ascorbate were added to protein lysates (2 mg/mL in RIPA buffer), and vortexed at room temperature for 2 h. Cy5- or Cy3-labeled protein samples were resolved by SDS-PAGE, imaged by Typhoon FLA 9500 (GE) and analyzed by image Lab 3.0. The gels were stained by Coomassie Brilliant Blue (CBB) as the loading control. Images of Coomassie Brilliant Blue-stained gels were collected on a ChemiDoc XRS+ (Bio-Rad).

## Cell confocal fluorescence microscopy imaging

Cells were seeded into Lab-Tek Chambered Coverglassd ($1 \times 10^4$ cells/well), treated with vehicle or indicated metabolic precursors at 37 °C for 48 h, and fixed with 4% formaldehyde (w/v) in PBS for 15 min. For click reaction, cells were incubated with 50 µM alkyne AZDye 488, premixed BTTAA-CuSO4 complex (50 µM CuSO4, BTTAA/CuSO4 6:1) and 2.5 mM freshly prepared sodium ascorbate in PBS at room temperature for 10 min. For nucleus staining, cells were incubated with 5 µg/mL Hoechst 33342 or DAPI at room temperature for 20 min. Cells were washed three times with PBS after each step. Finally, cells were imaged by a Zeiss LSM 700 laser scanning confocal system equipped with a ×40 or ×63 oil immersion objective lens.

## Flow cytometry analysis

Cells were seeded into 6-well plates ($5 \times 10^5$ cells/well), treated with vehicle or indicated SeMOE probes at 37 °C for 48 h. SeMOE-treated cells were trypsinized, transferred into 96-well tissue culture plates, centrifuged (400 $g$, 5 min, 4 °C) and washed three times with 1% FBS (v/v) in cold PBS. For click reaction experiment, cells ($5 \times 10^5$) were resuspended in 100 µL PBS containing 0.5% FBS (v/v), 50 µM alkyne-biotin, premixed BTTAA-CuSO4 complex (50 µM CuSO4, BTTAA/CuSO4 6:1) and 2.5 mM freshly prepared sodium ascorbate, followed by reaction at 4 °C for 10 min. After three washes with 200 µL 1% FBS (v/v) in cold PBS, cells were incubated with 1 µg/mL Streptavidin-Alexa Fluor 488 conjugate at 4 °C for 30 min, followed by three washes with 1% FBS (v/v) in cold PBS and flow cytometry analysis. For sialic acid level analysis, SeMOE-treated cells ($5 \times 10^5$) were incubated with 5 µg/mL biotinylated SNA or MAL II in 100 µL of 1% FBS in PBS at 4 °C for 1 h. After three washes with 1% FBS (v/v) in cold PBS, cells were incubated with 100 µL Streptavidin-Alexa Fluor 555 conjugate (1 µg/mL) at 4 °C for 30 min. The cells were washed three times with 1% FBS (v/v) in cold PBS, and applied to BD LSRFortessa Flow Cytometer system or ACEA NovoCyte benchtop flow cytometer (Acea Biosciences, USA). Flow cytometry data were analyzed using FlowJo.

## β-elimination treatment of cellular glycoproteins

500 µg of glycoprotein sample from MCF-7 cells was incubated with 200 µL of 1 M NaBH4/0.05 M NaOH for 24 h at 45 °C, cooled on ice, and mixed with 200 µL acetic acid. After β-elimination treatment, cell proteins were precipitated by adding into a MeOH/CHCl3 mixture (aqueous phase/MeOH/CHCl3 = 4:4:1, v/v/v). The precipitated proteins were collected by centrifugation (20,000 $g$, 10 min, 4 °C), washed three times with cold methanol, dried at room temperature, digested by nitric acid, and analyzed via ICP-MS.

## Detection of UDP-GalNSe

Vehicle- or 1,6-Pr2GalNSe-treated MCF-7 cells ($1 \times 10^7$) were washed three times with cold PBS, incubated with 2 mL of 80% methanol in water at −80 °C for 1 h, scraped and centrifuged centrifuged at 12,000 $g$ for 20 min at 4 °C. The supernatants were diluted to 10-fold of volume with water. 1 µL sample was then subjected to UPLC-TSQ-MS/MS (Thermo Fisher Scientific) with a triple quadrupole system for UDP-GalNSe level analysis. The standard compound UDP-GalNSe was synthesized by our laboratory.

## N-glycoproteomics and SESTAR++ analysis

For protein sample preparation, HeLa cells were treated with vehicle or indicated SeMOE probes (2 mM ManNSe, 2 mM SiaNSe, 2 mM 9AzSiaNSe, 200 µM 1,6-Pr2ManNSe or 100 µM Ac4ManNSe), collected, washed, resuspended in 4% SDS (m/v) in PBS containing protease inhibitor, and lysed by sonication, followed by heating at 95 °C for 10 min and centrifugation (20,000 $g$, 10 min, 18 °C) to remove debris. Protein concentrations were determined by a BCA protein assay kit (Thermo Fisher Scientific). Then, cell proteins were precipitated by eight volumes of cold methanol at −80 °C overnight. Precipitated proteins (1 mg) were denatured in 8 M urea, reduced in 10 mM dithiothreitol (DTT) at 37 °C for 1 h, alkylated in the dark by 20 mM iodoacetamide (IAA) at room temperature for 30 min, and diluted in 50 mM NH4HCO3 to a final urea concentration of 0.8 M. The resulting solution was incubated with trypsin (Promega) (substrate: enzyme ratio at 50:1) at 37 °C for 18–20 h, desalted using the C18 column (Waters), dried by vacuum centrifugation. The dried peptides were resolved and enriched with an in-house ZIC-HILIC (Merck Millipore) micro-column as previously described[44]. Enriched glycopeptides were dried by vacuum centrifugation and subjected to an Orbitrap Fusion Lumos Tribrid Mass Spectrometer with an EASY-Spray ionization source (Thermo Fisher Scientific) for LC–MS/MS analysis.

For LC–MS/MS analysis, all above enriched glycopeptide samples were resuspended in 0.1% FA in water, separated over a 50 cm Easy-Spray reversed-phase LC column (75 µm inner diameter packed with 2 µm, 100 Å, PepMap C18 particles, Thermo Fisher Scientific). The solvents (A: water with 0.1% formic acid and B: 80% acetonitrile with 0.1% formic acid) were driven and controlled by a Dionex Ultimate 3000 RPLC nano system (Thermo Fisher Scientific). The gradient was 6 h in total:1–7% solvent B in 10 min, followed by an increase from 7–35% solvent B from 11 to 311 min, an increase from 35–44% solvent B from 311 to 353 min, and an increase from 44% to 99% solvent B from 353 to 356 min. The MS parameters for glycopeptide analysis were: (1) MS: resolution = 120,000; AGC target = 500,000; maximum injection time = 50 ms; included charge state = 2–6; dynamic exclusion duration = 15 s; each selected precursor was subject to HCD-MS/MS; (2) scout HCD-MS/MS: isolation window = 2; resolution = 15,000; AGC target = 500,000; maximum injection time = 250 ms; collision energy = 30%; stepped collision mode on, energy difference of ± 10%.

For data analysis, RAW data of MS1 and MS2 were then searched by SESTAR++ and pGlyco3, respectively, as previously described[42,44]. Peptide spectrum matches in the pGlyco3 searches, MS1 matches in the SESTAR++ searches, and intact N-glycopeptides identified are shown in Supplementary Data 1–3. Briefly, the raw data were searched against SwissPort Homo sapiens proteome database downloaded from Uniprot (https://www.uniprot.org) and glycan database built in pGlyco3 or uploaded (https://github.com/pFindStudio/pGlyco3/tree/main/gdb_files). The SiaNSe-H2O (m/z = 385.02755, defined as NS) and the 9AzSiaNSe-H2O (m/z = 410.03404, defined as SZ) were set as the "variable glycan modification" in the pGlyco3 software, which substituted Sia (A)-H2O, respectively. The N-glycans were searched against pGlyco-N-Human. Check glycan legality was enabled. The parameter

precursor tolerance was set to ± 10 ppm and fragment tolerance ± 20 ppm. Other parameters were at default values.

## Trogocytosis analysis of Jurkat cell and K562 cell

Jurkat cells were firstly incubated with 50 µM Ac$_4$ManNSe, 200 µM 1,6-Pr$_2$ManNSe, or 1 mM 9AzSiaNSe at 37 °C for 48 h, collected by centrifugation (400 $g$, 5 min), washed three times with 2% FBS (v/v) in cold PBS and treated with or without 0.2 mg/mL sialidase at 37 °C for 30 min. For Ac$_4$ManNSe- or 1,6-Pr$_2$ManNSe- treated cells, Jurkat cells were stained with DiI-PE staining buffer (5 µM DiI-PE in complete culture medium) in the dark for 10 min at room temperature, and washed three times with complete culture medium. For 9AzSiaNSe-treated cells, Jurkat cells were biotinylated via click reaction. Briefly, the 9AzSiaNSe-treated Jurkat cells were resuspended at $5 \times 10^6$ cells/mL in a click reaction buffer containing PBS, 0.5% FBS (v/v), 50 µM alkyne-biotin, premixed BTTAA-CuSO$_4$ complex (50 µM CuSO$_4$, BTTAA/CuSO$_4$ 6:1) and 2.5 mM freshly prepared sodium ascorbate. The cells were reacted at 4 °C for 10 min, and washed three times with 2% FBS (v/v) in cold PBS.

K562 cells were stained with Did-APC staining buffer (5 µM Did-APC in complete culture medium) in the dark for 10 min at room temperature, and washed three times with complete culture medium.

For trogocytosis experiment, DiI-PE and biotinylated labeled Jurkat cells, were co-cultured with Did-APC labeled K562 cells ($2 \times 10^6$ cells/mL) in complete 1640 medium, respectively. Co-incubations were set up in 6-well plates at varied cell number ratios (Jurkat:K562 at 5:1, 1:1, 1:5), with total cells at a cell number of $8 \times 10^6$/well. Cell mixtures were centrifuged at 150 $g$ for 30 s to favor cell contact, and co-incubated with or without 0.2 mg/mL sialidase for 2 h at 37 °C. After co-incubation, cells were centrifuged (400 $g$, 4 min, 4 °C), washed three times with 2% FBS (v/v) in cold PBS. Then, the co-incubation of DiI-PE labeled Jurkat cell and DiI-APC labeled K562 cells were directly used for flow cytometry analysis and sorting, while the co-culture of biotinylated Jurkat cells and DiI-APC labeled K562 cells were additionally incubated with 1 µg/mL streptavidin-Alexa Fluor 555 (PE) in the dark at 4 °C for 30 min. After three washes with 2% FBS (v/v) in cold PBS, cells were applied to for flow cytometry analysis and sorting. Flow cytometry analysis was conducted on a ACEA NovoCyte benchtop flow cytometer (Acea Biosciences, USA), and flow cytometry sorting was conducted on a BD FACSAria™ III Sorter. K562 cells were sorted in APC positive gate according to the flow cytometry gating strategy (Supplementary Note 3), accurately counted, digested with nitric acid, and analyzed by ICP-MS.

## Transwell co-culture of mouse cancer cells and RAW 264.7 cells

All transwell co-culture experiments were performed in 0.4 µm-sized transwell inserts (Coring, USA). Before co-incubation, mouse cancer cells were treated with 200 µM 1,6-Pr$_2$ManNSe or 1,6-Pr$_2$GalNSe for 24 h in 6-well plates, while RAW 264.7 cells were incubated with or without 100 ng/mL LPS or 20 ng/mL IL-4 for 36 h in 6-well plates. SeMOE-treated cancer cells were collected, seeded into inner transwell inserts ($1 \times 10^5$ cells/insert), and incubated with 200 µM 1,6-Pr$_2$ManNSe or 1,6-Pr$_2$GalNSe overnight to guarantee SeMOE efficacy. In the meantime, RAW 264.7 cells were collected, seeded into 12-well plates ($5 \times 10^5$ cells/well), and incubated with or without 100 ng/mL LPS or 20 ng/mL IL-4 overnight to maintain the desired macrophage polarization status. After cell adherence, cancer cells and RAW 264.7 cells were washed three times with PBS, and co-cultured in a transwell setting at 37 °C for 24 h. Both cancer cells and RAW 264.7 cells were collected at various time points (1 h, 3 h, 6 h, 12 h, 24 h), washed three times with cold PBS, counted by the automated cell counter (Invitrogen), and subjected to macrophage polarization detection, ROS measurement or ICP-MS analysis.

For the effect of cancer cell-culture medium on macrophage polarization status, RM1 cell supernatant was collected, added to the culture medium of RAW 264.7 cells at a ratio of 50% or 100% (v/v), and incubated for 48 h. After that, RAW 264.7 cells were collected, washed three times with cold PBS, and used for macrophage polarization detection. RAW 264.7 cells were collected, washed three times with cold PBS containing 2% FBS (v/v), resuspended at $5 \times 10^6$ cells/ mL in cold PBS containing 2% FBS (v/v), anti-mouse CD80-PE (Biolegend, 104707, 1:200) and anti-mouse CD206-APC (Biolegend, 141707, 1:200). After incubation in the dark at 4 °C for 30 min, cells were washed three times with 2% FBS (v/v) in cold PBS, and analyzed by NovoCyte benchtop flow cytometer in PE and APC channel.

## Isolation and co-culture of mouse GCs and oocytes

GCs and oocytes were isolated from ovaries of 3–4 week-old ICR female mice. Briefly, mouse ovaries were collected, removed excess tissue with tweezers. For GC isolation, these ovaries were punctured with an 18-gauge needle to release GCs. The GCs were filtered through a 40 µm filter, collected into 15 mL centrifuge tubes (Corning, USA) and centrifuged at 400 $g$ for 5 min. GC pellets were resuspended in complete DMEM/F-12 medium supplemented with 10% FBS (v/v), 100 U/mL of penicillin, 0.1 mg/mL of streptomycin, 1 mM pyruvate, and 2 mM glutamine at a density of $5 \times 10^5$ cells/mL, seeded in 6 cm Petri dishes, and cultured at 37 °C, 5% CO$_2$ with saturated humidity. The medium was replaced every 48 h of incubation until 80–90% confluency. To isolate oocytes, mouse ovaries were chopped with a blade to release oocytes. Denuded oocytes were obtained by a series of mouth-controlled micropipettes with successively smaller diameters, and cultured in droplets of M2 medium (Sigma) at 37 °C in an incubation chamber (Billups Rothenberg, Del Mar, CA) infused with 5% O$_2$ and 5% CO$_2$.

For SeMOE of GCs, freshly isolated GCs were seeded into a 48-well plate with cell culture slides at a density of $5 \times 10^5$ cells/mL and treated with vehicle or the indicated SeMOE probe when they reached approximately 30% confluence. After incubation for 36–48 h, GCs were trypsinized, collected, fixed with 4% formaldehyde (w/v) in PBS for 20 min at room temperature, and washed three times with 0.5% BSA (m/v) in PBS. Then, cells were incubated with 50 µM alkyne AZDye 488 or alkyne AZDye 647, BTTAA-CuSO$_4$ complex (50 µM CuSO$_4$, BTTAA/CuSO$_4$ 6:1) and 2.5 mM freshly prepared sodium ascorbate in PBS for 10 min at room temperature, followed by incubation with DAPI for 10 min at room temperature, and three washes with 0.5% (m/v) BSA in PBS. For co-culture of GCs and oocytes, GCs were seeded into 24-well plates ($5 \times 10^5$ cells/well), treated with 1 mM 9AzSiaNSe for 36–48 h, and washed three times with PBS. After that, freshly isolated mouse oocytes were added to wells, and co-cultured with 9AzSiaNSe-labeled GCs in M2 medium with or without 100 µM CBX for 14 h. Oocytes were collected by mouth-controlled micropipettes for further glycan confocal fluorescence microscopy imaging. Briefly, oocytes were fixed with 4% formaldehyde (w/v) in PBS for 20 min, and permeabilized with 0.5% Triton X-100 (v/v) in PBS for 10 min. After three washes with 0.5% BSA (m/v) in PBS, oocytes were incubated with 50 µM alkyne AZDye 488, BTTAA-CuSO$_4$ complex (50 µM CuSO$_4$, BTTAA/CuSO$_4$ 6:1) and 2.5 mM freshly prepared sodium ascorbate in PBS for 10 min at room temperature. After click reaction, oocytes were incubated with DAPI for 10 min at room temperature, followed by three washes with 0.5% (m/v) BSA in PBS. Fluorescence imaging was performed on a Zeiss LSM 700 laser scanning confocal system equipped with a ×40 or ×20 objective lens.

## In situ visualization and quantification of tissue sialoglycans by LA-ICP-MS

The organs of SeMOE-treated mice were collected, frozen in liquid nitrogen-isopentane, and sectioned to 10 µm slides for LA-ICP-MS imaging. For multi-element imaging, an Iridia laser ablation system (Teledyne Photon Machines, Bozeman, USA) with a 193 nm ArF excimer laser and a low-dispersion ablation cell was used. The LA system

was coupled to icpTOF 2 R ICP-TOFMS (TOFWERK AG, Thun, Switzerland) by the aerosol rapid introduction system (ARIS). High-purity helium was used as transport gas to carry ablated sample aerosols from the ablation chamber to ICP-TOFMS. For LA-ICP-TOFMS experiments, daily tuning was performed by using a NIST 612 glass standard RM (National Institute of Standards and Technology, Gaithersburg, USA) to obtain high sensitivity for $^{59}$Co, $^{115}$In, and $^{238}$U signal and a low oxide rate (e.g., $UO^+/U^+$ <3%). In order to obtain the shortest single pulse response, the flow of the inner cell and the outer cell was optimized. The size of laser spot was set as 20 µm, the distance between lines was set as 20 µm, the dosage was set as 1, the laser frequency was set as 250 Hz. The parameters and conditions of laser ablation system and ICP-TOFMS are showed in Supplementary Table 3. HDIP software (Teledyne Photon Machines, Bozeman, USA), TOFware, and laser image viewer software (TOFWERK AG, Thun, Switzerland) were used for imaging.

For quantitative imaging, the Iridia laser system was coupled to NexION 300D ICP-MS (PerkinElmer, Norwalk, USA) by the ARIS. Helium was used as transport gas. For LA-ICP-MS experiments, ICP-MS was tuned by using a NIST SRM612 glass for a maximum $^{115}$In and a low oxide rate (e.g., $UO^+/U^+$). The isotopes $^{78}$Se was measured throughout all experiment. To ensure that tissue samples could be fully ablated, the size of laser spot was set as 20 µm, the distance between scanning lines was set as 40 µm. The optimized single pulse response was about 10 ms, the laser frequency was set as 100 Hz. The gelatin calibration standards containing 0, 1, 10, 50, 100 ppm selenium and tissue sialoglycans were analyzed together. The parameters and conditions of laser ablation system and ICP-MS are shown in Supplementary Table 4. Microsoft Office Excel 2020 and Origin Lab 2020 were used as data treatment software, and images integration was performed by the Iolite Software (v3.6) on Igor Pro 7 (WaveMetrics, USA).

### Measurement of intracellular ROS levels

The production of ROS induced by SeMOE probes was detected by a fluorescent ROS indicator, 2′,7′-dichlorofluorescein diacetate (DCFH-DA, ROS assay kit, Beyotime, S0033). Briefly, cells were seeded into 12-well plates ($2.5 \times 10^5$ cells/well), treated with or without indicated SeMOE probes at 37 °C for 48 h. After SeMOE treatment, cells were washed three times with PBS and incubated with 5 µM DCFH-DA in the dark for 30 min at 37 °C. Then, cells were trypsinized, followed by three washes with PBS. The level of intracellular ROS was determined by the ACEA NovoCyte benchtop flow cytometer (Acea Biosciences, USA) in the FITC channel.

### Cell proliferation assay

Cells were seeded in a 96-well plate (2000 cells/well), treated with vehicle or SeMOE probes at varied concentrations for 48 h. The cells were washed three times with PBS and incubated with 100 µL medium containing 10 µL 2-(2-methoxy-4-nitrophenyl)-3-(4-nitrophenyl)-5-(2,4-disulfophenyl)-2H-tetrazolium monosodium salt solution (WST-8, Cell Counting Kit-8, Fdbio Science, FD3788) at 37 °C for 3 h. The absorbance at 450 nm was measured by a Synergy H4 Hybrid Reader (Bio-Tek) and normalized to the vehicle-treated cells.

### Quantitative real-time PCR (qRT-PCR)

MCF-7 cells were seeded into 6-well plates ($5 \times 10^5$ cells/well), treated with or without indicated SeMOE probes at 37 °C for 48 h. Total RNA was isolated from SeMOE probe-treated MCF-7 cells by RNA isolater total RNA extraction reagent (Vazyme, R401), and then converted to cDNA using the HiScript III RT SuperMix for qPCR kit (Vazyme, R323-01), followed by PCR using ChamQ universal SYBR qPCR master mix reagent (Vazyme, Q711-02) with gene-specific primers. RT-qPCR was conducted on a StepOnePlus™ real-time PCR System (Thermo Fisher Scientific). All of the reactions were run in quadruplicate. The expression levels of mRNAs were normalized to *GADPH* mRNA using

the $2^{-\Delta\Delta C_T}$ method. The primers used for qRT-PCR were provided in Supplementary Table 5.

### Cell migration assay

MCF-7 cells were seeded in 6-well plates ($7 \times 10^5$ cells/well) overnight. Then cells were wounded by pipette tips, washed three times with PBS, and incubated with or without 1 mM ManNSe, 1 mM SiaNSe or 1 mM 9AzSiaNSe for 36 h. The migrated cells were imaged by EVOS™ XL Core Imaging System (Thermo Fisher Scientific) at varied time. The inhibition of cell migration was analyzed by image J and normalized to the vehicle-treated cells.

### Cell apoptosis assay

The effect of SeMOE probes on cell apoptosis was detected by APC-Annexin V/PI Detection Kit (UElandy, A6030). In brief, cells were seeded into 6-well plates ($5 \times 10^5$ cells/well), treated with or without indicated SeMOE probes at 37 °C for 48 h. MCF-7 cells were collected and washed three times with 2% BSA (m/v) in PBS. 10 µL of binding buffer, 5 µL APC-Annexin V and 5 µL PI were sequentially added to $5 \times 10^5$ cells, mixed and incubated in the dark for 15 min at room temperature, followed by the addition of 400 µL binding buffer. Stained cells were analyzed immediately by ACEA NovoCyte benchtop flow cytometer (Acea Biosciences, USA) in FITC and APC channels.

### Statistics and reproducibility

Experiments in this study were performed at least in triplicate. Each dot represents an individual sample. Statistical analysis was performed using GraphPad Prism and Origin. Error bars in all statistical data represent mean ± standard deviation (SD). Comparison between the two groups was analyzed by Student's two-tailed t-test when not otherwise specified. Differences among multiple groups were analyzed by One-Way ANOVA test followed by post hoc Dunnett's test.

### Reporting summary

Further information on research design is available in the Nature Portfolio Reporting Summary linked to this article.

## Data availability

The mass spectrometry data, as well as all spectra for identified glycopeptides from different samples, have been deposited to the ProteomeXchange Consortium (http://proteomecentral.proteomexchange.org) via the PRIDE partner repository43 with the dataset identifier PXD042137 (N-glycoproteomics and SESTAR++ analysis based on SeMOE). Confocal microscopy imaging and LA-ICP-MS imaging data of this study can be accessed from Figshare [https://doi.org/10.6084/m9.figshare.c.6790350.v2]. The remaining data are available within the Article, Supplementary Information or Source Data. Source data are provided as a Source Data file.

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

## Acknowledgements

This study was supported by the National Natural Science Foundation of China (2207070006) (R.X.), the Natural Science Foundation of Jiangsu Province (BK20200299, BK20202004)(R.X.), the Chinese Academy of Sciences (YJKYYQ20210025)(M.W.), the Beijing Natural Science Foundation (Z230008) (M.W.), the Programs for High-level Entrepreneurial and Innovative Talents Introduction of Jiangsu Province (Individual and Group Program)(R.X.), the Fundamental Research Funds for the Central Universities (021414380508)(R.X.), the STI2030-Major Projects (2022ZD0211804)(R.X.), the National Key R&D Program of China (2022YFA1505600)(R.X.), and the China National Natural Science Foundation of Youth Fund Project (22107005)(B.C). We thank Prof. Xing Chen (Peking University) for help with the glycan labeling and glycoproteomics.

## Author contributions

R.X. and M.W. supervised the project. R.X. and X.T. conceived the study, designed the experiments, and analyzed the data. X.T. conducted most of the experiments unless specified otherwise. L.Z. performed the ICP-MS and LA-ICP-MS experiments under the supervision of M.W. X.T., B.C., T.C., and Y.Hs. contributed to the chemical synthesis. X.T. and Cha.W. performed the SeMOE in vivo. X.T., Y.Ha., and Y.L. performed the cytotoxicity assay, cell subcellular fraction isolation, and RNA (or lipid) sample preparation. X.T. and J.L. conducted the intact N-glycopeptide enrichment and pGlyco3 analysis. C.L., X.T., and Cha.W. conducted the co-incubation of GC-oocyte under the supervision of L.D. X.T. and G.J. performed the SESTAR++ searches under the supervision of Chu.W. X.T., L.Y., and C.Z. contributed to the CyTOF analysis. R.X. and X.T. wrote the manuscript with contributions from all the authors.

## Competing interests
