## [Peer Review File · Nature Communications]

Selenium-based metabolic oligosaccharide engineering strategy for quantitative glycan detectionEditorial Note: Parts of this Peer Review File have been redacted as indicated to remove third-party material where no permission to publish could be obtained.

REVIEWER COMMENTS

Reviewer #1 (Remarks to the Author):

The authors have done stellar work to develop SeMOE, the incorporation of selenium into metabolic labelling strategies to complement existing methods in chemical biology. The work is very comprehensive, from meticulous characterisation of the strategy, metabolic incorporation, proteomics, "imaging", and application in measuring cell-to-cell transfer for glycans between cells.

In my view, the manuscript is very suitable to a broad readership as it will be very useful. I only have a few suggestions based on understandability and potential drawbacks.

1) P. 2 the list of bioorthogonal sugars is longer if bacterial glycans are included.

2) ICP-MS should be introduced more carefully, including to discuss the workflow, advantages and drawbacks. It is very likely that readers (including myself) do not know the details of the technique.

3) The emphasis here is not on the GalNSe monosaccharides – however, it needs to be stated that for these sugars, it is not clear whether they are taken up by the mammalian GalNAc salvage pathway. Multiple papers have addressed metabolic compatibility in the past, and I would urge the authors not to overlook this.

4) Fig. 2b, being able to detect signal with low nanomolar monosaccharide concentrations is impressive but it is not clear what this is being compared to when stating that it is 5 orders of magnitude better compared to bioorthogonal sugars (p. 5). Furthermore, how does this fit to the authors using 50-200 μ M concentrations for the rest of the manuscript which is the same concentration range as for conventional MOE probes?

5) It should be stated what the current drawbacks are. ICP-MS is common but not common enough to compete with current technology in imaging and MS. For MS efforts, I would expect some kind of statement to realistically state the drawbacks, i.e. dynamic range, signal, number of glycopeptides... it is really not a problem if SeMOE does not have the same sensitivity/capabilities yet as other approaches as it is a new technique, but I would appreciate if the authors stated where they see room for improvement.

6) In Fig. 4, I don't follow what was APC labelled and what was PE labelled. The figure legends are not really informative, e.g. "Seleno-sialic acid transfer from RMI to RAW264.7 cells" and similar vague legends need to be expanded to show what was done. In the ICP-MS experiments, the authors need to bring the y axis into context, i.e. what are common values observed here, what is expected?

7) At some point, the authors switch to using the free monosaccharides which they need to use much higher concentrations of. This is especially true in the ROS experiments. I am not sure why this was done as an increase in concentration would also lead to an increase in ROS production. Did the authors also try with per-acetylated compounds?

8) Some phrases or sentences are off e.g. "nibbled" or "The interdisciplinary thought collision is vivified".

In the SI, please take care to include volumes of buffers, numbers of cells, confluency, types of plates etc. used in all experiments so anyone can repeat the experiment. For instance, the statement "MCF-7 cells were seeded in 6-well plates and incubated in 1640 medium overnight. The cells were collected, treated with vehicle or selenosugars as indicated for 36 h." needs much more information to be reproducible.

9) In my view, the potential anti-cancer application is a bit too preliminary to be included in the title of

the paper.

Reviewer #2 (Remarks to the Author):

Please refer to the attached document for suggestions/revisions.

In this manuscript “SeMOE allows for quantitative glycan perception and exhibits anti-cancer potentiality” the authors describe the synthesis and *in vitro/vivo* evaluation of selenium-containing sialic acid precursors, the presence of the selenium atom permitting quantitation by ICP-MS. The most useful innovation in this research, albeit one that is not well elaborated in the manuscript, is that the ICP-MS permits absolute glycan quantitation, the ICP MS source effectively eliminating the different response factors between different glycans and even different classes of glycoconjugates. However, there are some serious problems with the manuscript in its current form, both scientific and stylistic. These issues must be addressed before publication in *Nat. Comm.*

1. Aside from the ease of absolute quantitation (with out external instrument calibration) there is nothing that the SeMOE approach brings to the glycoscience community that cannot be achieved by other, currently existing tools. A major (if not the major) impetus behind the entire biorthogonal-labelling field was that the method permitted the enrichment/isolation of tagged glycoconjugates for subsequent analysis and/or tracking in live cells. The only way that SeMOE permits these classic MOE experiments is by still retaining the azide functional group. Thus, the sole innovation of the SeMOE is quantitative. In other words, the authors do almost nothing to indicate why a researcher should adopt this tool: no urgent research need is presented in the introduction, and the conclusion is highly exaggerated. [“We believe that the SeMOE strategy can serve as the Occam’s razor [...]”. Occam’s razor is a philosophical concept used by scientists to choose between (or make) hypotheses; it is not a chemical biology research tool. If the authors believe their tool is simpler than current MOE strategies (hence their reference to Occam) they are mistaken! There is nothing simple about this approach, either biochemically or analytically: the SeMOE approach requires sophisticated and expensive analytical tools that are virtually unavailable to nearly all life science researchers in most institutions. Please, remove reference to Occam!]. With respect to both absolute quantitation and low detection limit, radiolabeling approaches exist [<https://doi.org/10.1002/0471142727.mb1704s87>] which should be acknowledged. Additionally, the radiolabeling approach does not encounter issues about the tolerability of the unnatural Se-containing group appended to the mannosamine (see below). In sum, the manuscript in its current form gives no real reason for investing in SeMOE, other than that it can be done.

2. Se is a large atom. We know that subtle changes to the N-acetimido group of mannosamine influence the rate of incorporation of the corresponding sialic acid [<https://www.nature.com/articles/s41598-022-26521-3>]. The authors should compare by in gel fluorescence (like Fig 2j) 9AzSiaNSe and 9AzSia to verify. Likewise, the unnatural CMP-Neu5Ac analogues may, inadvertently, be inhibitors of sialyltransferases and/or the enzymes in the CMP-Neu5Ac pathway: a lectin blot or flow cytometry experiment should be performed to demonstrate that these SeMOE tools do not perturb the formation of sialosides. The utility of SeMOE, like MOE, is lost if these reagents influence the normal biosynthesis of glycoconjugates: I believe they do and it is erroneous to assume otherwise in the absence of evidence.

3. The flow cytometry data (Fig 2d and Sup. Fig. S4) should be displayed as histograms as is conventional (additionally, the number of events in these figures should be clearly indicated). Presenting the data as scatter plots obscures the fact that in 1,6-Pr2ManSe-treated cells (Fig 2d) the vast majority of events display a Se signal essentially undistinguishable from the control. Have the authors an explanation for this? This phenomenon is also inconsistent with the linear relationship presented in Fig 2e (and Sup. Fig S5): some form of gating must have happened here, correct?

4. ManNAc (and analogues) is converted to GlcNAc by most cells [<https://doi.org/10.1074/jbc.M212127200>]; GlcNAc-6-phosphate, meanwhile, is a substrate for a de-N-acylase while UDP-GlcNAc (and analogues) is epimerized to UDP-GalNAc. Thus, there is ample opportunity for the Se to end up in various glycoconjugates. Could this be why almost one third of the signal from Ac4ManSe is retained after neuraminidase treatment (Fig 2f)?

5. Statistics: Some statement of how stats were performed should be included in the methods section. Secondly, throughout the manuscript two-way ANOVAs are performed, eg. Fig 2f comparing untreated to neuraminidase cell surface and neuraminidase cell lysates. This should be recalculated with a one-way ANOVA followed by an appropriate *post hoc* test (like Tukey). A two-way test could be used to compare the different SeMOE reagents and neuraminidase treatment. Alternatively, if multiple comparisons are

made to a single control (as is frequently the case in this manuscript) then a Dunnett's test should be performed. In any case, the two-way test is inappropriate in all cases I could find in this paper.

6. Figure 3d attempts to present a mass balance for Se by using a pie graph, a format that does not permit an indication of the standard deviation of each measurement. More importantly, though, the experimental section indicates that each class of glycoconjugate appears to have been acquired separately, that is a protein fraction was collected from a separate set of samples as the lipid and RNA (and no details are provided about how the "other forms" of Se were deduced. If I understand the protocol correctly, then the parts-of-a-whole pie graph are misleading, because no attempt at making a mass balance was performed. To properly do this, separate fractions from the same group of cells should be measured and then normalized.

7. Comparison of the SeMOE tool to SILAC is made with some support provided by the glycoproteomics work. [Note: no mention of the mass spectrometer used for this was provided]. Substitution rates of up to 23% are reported (Fig 3f). As I understand this figure, the number of peptide spectra containing normal Neu5Ac are compared with those containing Se-Neu5Ac. But this isn't how SILAC is done at all! To get a real substitution rate, the same glyco-peptides need to be compared head-to-head....it does not appear that this was even attempted. In the present form, these data tell me more about the non-detection rate of the HPLC-MS that they do about the incorporation rate. Please pick several glycopeptides in common across all data sets and compare normal and Se-Neu5Ac quantities in each. You will very likely find that the incorporation rate is glyco-peptide-dependent, which will be a reflection of protein half-life. In the present form, these data are not convincing and incompletely analyzed.

Fig. 3h demonstrates the presence of a Se isotope window of 8 m/z, yet the experimental protocols suggest that the HCD-MS/MS isolation window was 2 m/z. How is this possible? If precursor ions were selected across a 2 m/z range, then at most only two Se atoms will be present

8. There are many over-stated or unsupported sentences in the manuscript. Here are several:

"And the unique property of selenium, such as redox response, can be exploited instead to interrogate glycan functions in their native surrounds". Firstly, this is not a sentence. Secondly, this was not done in this manuscript, in fact, it is not even suggested how this redox activity of selenium might be exploited. Please revise/clarify.

"[...] thus implying selenosugars remain intact under protein glycosylation processes". This sentence is in reference to Fig. 2a. These data do not support this at all (although other experiments do). Clarity is important here. These data indicate that the selenium atom is in these HeLa cells: that is it! Unless I'm mistaken, these are whole cells subjected to ICP-MS after acid digestion. Thus, Fig 2a. shows that the Se is on or in the cells, nothing more. Fig. 2b. does support that the label is in the protein fraction and Fig 2c. does this even more convincingly, but we cannot support the claim that this is in the glycoprotein fraction until the PNGaseF/sialidase tests are used to verify that the Se is associated with sialic acid.

Potential anticancer effects: this heading is pure wishful thinking. What was exactly done here was a measurement of the ROS generation in a range of immortalized cell lines. Comparing MCF7 to MCF10a simply tells me that these two cell lines are different from each other (which will likely be the case no matter how the ROS generation is induced). Things look even more ambiguous in Supporting Figure S12: cells vary widely in the ROS concentrations and very few appear to exhibit a concentration-dependence. Worse: ROS capacity, if it is due to the mere presence of the Se atom, should be equivalent for all compounds tested. If, on the other hand, the selenosugar must be incorporated into glycoconjugates to induce ROS, then some attempt needs to be made to normalize ROS levels to the amount of Se contained in the glycoconjugate fraction in each cell. Finally, there needs to be some statement indicating that the amount of ROS generation induced (and this is far from clear) is actually sufficient to kill or inhibit cancer cell growth. Is it? Can these data be correlated with the cell viability assays?

"A plethora of sialoglycoconjugates from Jurkat cells [...] were transferred to K562 cells....". There is in fact no evidence that sialoglycoconjugates were transferred, let alone "a plethora". These data show that Se is

transferred between the cells, but there are zero controls performed to indicate that the Se is in the form of sialic acid, let alone glycoside-bound sialic acid. If the Ac4ManNSe is inefficiently metabolized—and fig 3d makes it clear that even after 48 h, most of it still exists as “other forms” (i.e. not glycosides)—how can we be sure that the Jurkat cells are not simply excreting the ManNSe for subsequent re-uptake by the K562 cells? Can the K562 cells be neuraminidase treated before flow cytometry? (A loss of some Se signal here would verify that at least some Neu5Ac transfer occurred). Also, why are the controls showing such high Se concentrations here (Fig 4c)?

“We also demonstrated that participants in the indirect cellular communication were also O-glycosylated.....”. How? This wasn’t demonstrated at all. There are tools/procedures for discriminating between N- and O-glycans, none of which were employed in this manuscript. Please revise this sentence.

9. There are several instances where the authors seem unaware of existing literature. Some rewording is needed here. For example:

Describing alkyne-containing carbohydrates as “new-to-nature”. Azides may be new-to-nature, but alkyne-containing natural products do exist [<https://doi.org/10.1039/D0CB00190B>]. I would suggest using the term “biorthogonal” to describe these modified carbohydrates. In the following sentence describing “the Nobel-Prize-winning biorthogonal reaction” I would recommend referring explicitly to the more chemically-descriptive cycloaddition reactions used to detect the azide/alkyne-containing carbohydrates.

[...] which we termed “selenosugars” The term selenosugar has been in existence for sometime now see [<https://doi.org/10.1021/acsomega.1c02160> for example]. Also (this point is debatable but should be noted) “selenosugar” is usually in reference to a carbohydrate where one of the hydroxyl groups is replaced with the selenium atom, analogous to amino-sugars or thio-sugars.

10. Why was the lower abundance ^{78}Se for quantitation by ICP-MS?

11. Several instances where more precise scientific terminology should be used:

“Perceived” is used throughout the manuscript including the title. This is an ambiguous word in this context. I would suggest changing to “detected” or “quantitated” throughout.

“K562 cells were sorted and the nibbled selenoglycans [...]”. I would recommend “transferred” or “ingested”.

[...] and portrayed as an avant garde tool to visualize and quantify newly-synthesized sialoglycans in animals with elemental precision. Avant-garde is a term usually reserved for art/literature, indicating that the work produced by an artist is innovative/experimental while usually being unacceptable to the establishment at the time. I don’t think the authors mean to imply SeMOE will be viewed as unacceptable by *Nat. Com.* readers; if the intend to convey that the technique is innovative, then please use the word “innovative” rather than avant garde. Likewise, precision is an analytical term referring to the agreement between replicate measurements, usually expressed as the standard deviation. Please revise this sentence.

[...] interdisciplinary thought collision is vivified”: this is both unclear and, if I may guess the intended meaning, seriously overstates the conclusions of this research. Please revise.

“The core concept of SeMOE is minimalism”. In addition to access to a synthetic chemistry lab suitable for handling selenium-containing reagents, the entire SeMOE protocol uses instruments/tools/infrastructure that are not available in most university campuses. There is absolutely nothing minimalist about this research! Please revise.

Reviewer #3 (Remarks to the Author):

The submitted manuscript is an interesting exploration of the utility of monosaccharides tagged with selenoethers. The addition of selenium to the monosaccharides provides an elemental handle to allow convenient monitoring by elemental mass spectroscopy, and a unique isotopic signature for glycoproteomics applications. The manuscript combines many types of experiments showing the potential of these labelled monosaccharides in a variety of applications. Overall this is good work but many statements need to be clarified and many key technical details are missing.

The key innovation in the work is the ability to directly quantify the amount of selenium with atomic mass spectroscopy. The synthesis of the seleo-tagged sugars is well described and the identity and purity of the compounds by NMR is adequate.

In the description of the toxicity of these compounds the authors state that because selenoglycosides are produced as a selenium detoxification mechanism, the seleo-tagged sugars should not be toxic. This is not necessarily true, as they are very different chemical structures. Furthermore, the authors state "comparable (toxicity) with native monosaccharide precursor or azido sialic acid at the range of 0-5 mM". This is a gross simplification of the data in sup. Fig. 1 where the majority of the compounds have toxicity above 1mM and many are more toxic. This statement needs to be clarified.

In Figure 2 the authors characterize the uptake of the seleno-tagged monosaccharides in cell culture. The methods describing these experiments do not align with the data presented. In the method for Figure 2a, a protein concentration determination method is not described, yet the signal is normalized to protein content on the y-axis. Figure 2d, the authors change cell lines from HeLa, used in 1a-c to 293T, this makes the CyTOF signal difficult to interpret in terms of the degree of labelling. It is necessary to have a bulk measure of the incorporation into 293T cells to provide an estimate of signal in CyTOF. Clearly the CyTOF signal is very low, nearly unusable, due to the low sensitivity of the method for selenium and the authors should comment on this in the text. The CyTOF data should be presented as a histogram to more clearly show the relative event count numbers.

Figure 2f is central to all of the work in this manuscript. In the majority of the subsequent experiments the authors rely on the data in 2f to support the central tenant of the manuscript that the seleno-tagged sugars deposit selenium only into glycans, and not into other biochemical pathways. This is not explicitly stated and should be. One could imagine that there is degradation, or simple metabolism that could lead to Se accumulation via other mechanisms not related to glycan biosynthesis. Only if all of the Se accumulation is glycan related is the 'absolute quantitation' claimed in the introduction supported. As can be seen in the neuraminidase treated cells a large fraction of the signal is from sialylated glycans, however not all the signal, and it is surprisingly dependent on the monosaccharide. The Ac4Man derivative gives the most neuraminidase resistant material, and this is consistent with non-specific labelling known with peracylated sugars, and the data in figure S6, so this should be stated. The sialic acid derivatives also give significant neuraminidase resistant material. How the data was obtained for the neuraminidase treated lysate is not complete in the methods section. The method section does not explain how the glycans were separated from the protein after enzymatic treatment, thus it is unclear if this measurement also reflects the loss of the small molecule tagged Se pool which may or may not be glycan related. This experiment needs to be clarified and elaborated on in the text. It is central and needs to be rigorously analyzed and discussed.

Figure 2g is additional support for where the Se signal is localizing. It is difficult to know what to take from this data as the Ac4Man derivative shows significant toxicity at the concentrations used.

Overall from the data in figure 2 it appears that most of the Se accumulation is into cellular glycans, but not all. Any claims of 'quantitative detection' of MOE labelling need to be tempered to reflect these data.

I read the methods and results section for panels 3b-d multiple times and did not understand the

experiments. The description of the fractions needs to be revised.

In figure 4 the authors describe the transfer of material between cells whom are allowed to contact, or whom are separated by a membrane. The authors claim that the labeled material is sialoglycans, and this may be true, but it could also be any of the other intermediates, or potentially off pathway Se containing molecules. At the very least this limitation needs to be explicitly stated. Control experiments similar to those done in figure 2 would aid in the support of the authors hypothesis. It is odd that the Se labelling with the different monosaccharides in GC's is opposite to what is observed in the Hela cells in Figure 2.

In figure 5 the authors do biodistribution studies on the seleno-tagged sugars. Interestingly the Se signal in the organs decreases when the dosing was increased from 160 to 320 mg/kg in both Se-sugars studied. This is unexpected and should be commented on in the text, is there a rational for this observation? The laser ablation studies do not include a scale bar. Does the observed morphology of the signal agree with previous studies using MOE with sialic acid?

Figure 6 explores the effects of the tagged sugars on ROS in MCF7 vs MCF10A. A dramatic difference was observed but the reason for this difference was not investigated.

Overall this manuscript nicely shows the potential of using heavy element tagging of metabolic precursors for following MOE. The range of studies presented clearly demonstrate the value of direct quantification using atomic mass spectrometry. References to some of the other metabolites that have been tagged with heavy elements and used for CyTOF experiments is warranted.

Despite these benefits that authors need to clarify that other Se metabolites may be playing a role in these measurements and all of the y-axis measuring Se content should be modified from seleno-sialic acid to Se content.

Reviewer #4 (Remarks to the Author):

The manuscript "SeMOE allows for quantitative glycan perception and exhibits anti-cancer potentiality" is an interesting, well written, and (generally) meticulously performed study that provides a way to directly detect incorporation of a sugar analog used in metabolic oligosaccharide engineering (MOE) into cellular glycans. The main advance in this study is the direct detection of the non-natural glycan through detection of the inherent selenium atom(s); this avoids cumbersome second-step labeling methods required with "classic" MOE approaches. As an important bonus, the method is efficient, with metabolic incorporation (most likely) of the selenosugars more facile than the widely-used azido-modified analogs that are currently workhorse reagents in the field. The study goes on to show biological applications of the selenosugars, mostly the transfer from one cell type to another as occurs in trogocytosis (note that these "applications" basically verify / illustrate already-known biological knowledge but they do demonstrate that the method has future potential as a biological probe). Overall I believe that this study could be worked into a high impact publication with a few edits and improvements as described below

Major points

The N-glycoproteomics data shown in Figure 3 suggests that there is biased incorporation of the seleno-sialic acids into certain subsets of sialoglycans (e.g., to use a hypothetical example of three heavily sialylated proteins – let's say each with 20 sialic acids – one might have 0 or 1 natural sialic acids replaced with seleno-sialic acid, the second might have 3 or 4, and the third might have 10 to 12). Note that this a totally expected result, that seems to be reflected in this data. The reason I mention this is that the authors should avoid use of language that could be interpreted that each of these three hypothetical sialylated proteins have the same degree of incorporation, oh let's say 5

seleno-sialic acids each. For example, the statement "SeMOE developed in this work provides a simplified, yet versatile and convenient protocol for absolute quantification during glycan-mediated processes based on the feature of the non-metal element Se" could be interpreted that absolute quantification of the endogenous glycans involved in these processes is possible with this technique, In reality, what can be measured is the absolute quantification on non-natural analog incorporation, which can (indeed, most likely) is biased. To be more rigorous about providing (or disproving) this premise, a comparison of the seleno-sialic acid signals with endogenous sialic acid levels (e.g., perhaps through lectin staining) would be interesting.

A "selling point" for the use of seleno-sugars is the low background of selenium in cells and tissues; nevertheless (as the authors acknowledge by mentioning the existence of seleno-proteins) there is *some* level of background signal in human cells and tissues. It would be helpful if the author attempt to quantify the background ◊ for example, is the background signal in Figure 2c just "noise" or is it legitimately from endogenous selenoproteins? Just going from memory, I recall that a person has ~ 10¹⁸ selenoproteins in their body, which (if distributed equally, which is almost certainly not the case) works out to ~100,000 copies per cell. By comparison, going from data presented on Page 12 (1.6-4.5 femtomoles selenosugar per Jurkat cell) that works out according to my quick calculations to be ~109 copies of selenosugar per cell, or about four orders of magnitude over the background/"noise" from endogenous signal. If my quick calculations are correct the authors might want to more explicitly/quantitatively outline this rather outstanding signal to noise/background ratio. Also, "femtomoles sugar per cell" is, in my opinion, not that understandable by the average reader insofar as it comes across as a rather small number; again, as a "selling" point the authors might want to briefly (perhaps in the Discussion) provide context that this level is actually ~ one billion copies per cell, of which 0.1-3% is transferred during trogocytosis (and possible compare this to levels of conventional CD markers, which occur at much lower levels)

Experimentally, the data in Figure 6 concerning ROS production in meaningless w/out appropriate controls. For example, does atomic (in salt form?) selenium have the same effects at the equivalent doses? Contrary-wise, do MOE analogs w/out selenium (e.g., Ac4ManNAc, Ac4ManNAz, or perhaps thiol-modified ManNAc analogs) have the same impact by perturbing sialic acid biosynthesis.

Minor points

Page 1, Abstract: type in heading (Abatract)

Page 2, first line: "the most appealing" should be less hyperbolic (i.e., "an appealing" would be better)

Page 2, lines 4/5: the statement "Such compounds are converted into glycoenzyme substrates within cells" is often but not always correct; for example, the first generation ManNAc analogs developed by Reutter's group were used direct in MOE w/out any metabolic conversion (e.g., removal of O-hydrosyl acetyl protecting groups)

Page 4, Figure 1a: the meaning "of limitation" (in "attomolar detection of limitation") is unclear (would just "attomolar detection" be sufficient?)

Page 4, last paragraph: considering that two studies (plural) are referenced, should "A previous work" also be plural (e.g., something like "Previous studies")?

Page 5, second line: "lyzed" is more commonly "lysed"

Page 5, first paragraph: It'd be helpful to state here how proteins were extracted (e.g., whole cell fractions, membrane fractions, cytosolic fractions)

Page 5, first paragraph: compared to Figure 1a where attomolar sensitivity is mentioned, why is

nanomolar sensitivity mentioned here? (can the disparity be explained? [e.g., perhaps the authors mean "attomole" rather than "attomolar"?)

Page 11, Figure 3d: should "newly-synthezied" be "newly-synthesized"?

Page 12, near the bottom: "scavenged" might be a more appropriate word compared to "nibbled"

Page 16, approximate middle of the page: Please clarify what "are challenging to analyze using classical MOE strategy when blood-brain barrier (BBB) arises" means. Does this mean that the BBB blocks transit of the classic MOE monosaccharides such as Ac4ManNAz but not the selenium analogs? Or is the pitfall in the classic strategies the need for the imaging/labeling agents to cross the BBB? (or some combination of these two factors?)

Page 20, in the middle: the phrase "The interdisciplinary thought collision is vivified" is certainly a catchy phrase! On the other hand, the meaning of this flowery language is not completely clear to me (do the authors mean something like "The interdisciplinary nature of MOE is vivified..." or maybe "The interdisciplinary nature of classic MOE is further extended by..."?)

Page 22, end of references: In some type of weird formatting glitch, References 48-50 are separate from the rest of the references (appearing on Page 38) \diamond I suspect that all of the references should be presented with those on Page 38, i.e., they should appear after the methods, section).

Page 25: The synthesis of Ac4GalNSe is not described and/or referenced

Page 27: please provide the g force, not rpm, for the 12,000 rpm centrifugation step(s) \diamond the rpm is meaningless w/out knowing what centrifuge rotor was used (check for 1,500 and 2,000 rpm on Page 29 as well, and throughout)

We would like to acknowledge all 4 reviewers for your precious time and efforts working on this manuscript. We have carefully considered each comment, and addressed all points with additional data to improve the overall quality of the manuscript. Below are our ***point-to-point responses*** to the comments and concerns raised by the reviewers. Similar comments raised by different reviewers are ***annotated in blue*** for cross-reference if needed. The cited, newly added or revised figures mentioned in the responses are shown correspondingly for the ease of your references. The manuscript file is now provided with both an annotated version with changes ***marked in blue***, and a formatted manuscript.

Referee:1

Review Comment 1:

“The authors have done stellar work to develop SeMOE, the incorporation of selenium into metabolic labelling strategies to complement existing methods in chemical biology. The work is very comprehensive, from meticulous characterisation of the strategy, metabolic incorporation, proteomics, “imaging”, and application in measuring cell-to-cell transfer for glycans between cells.

In my view, the manuscript is very suitable to a broad readership as it will be very useful. I only have a few suggestions based on understandability and potential drawbacks.”

Response: We are grateful for the positive feedback and strong support from respected Reviewer 1.

Review Comment 2:

“P. 2 the list of bioorthogonal sugars is longer if bacterial glycans are included.”

Response: We thank Reviewer 1 for pointing this out. We notice that we have only discussed the eukaryotic monosaccharide precursors in metabolic oligosaccharide engineering. Rare sugar metabolic chemical probes such as bacillosamine, pseudaminic acid, 3-deoxy-d-manno-oct-2-ulosonic acid (Kdo), trehalose, *N*-acetyl-D-fucose (d-FucNAc), and *N*-acetyl muramic acid (MurNAc) and others have been reported, and constitute a substantial amount of metabolic monosaccharide chemical reporters. We therefore select and include several literature with regard to these bacteria-specific carbohydrate reporters.

Review Comment 3:

“ICP-MS should be introduced more carefully, including to discuss the workflow, advantages and drawbacks. It is very likely that readers (including myself) do not know the details of the technique.”

Response: We thank Reviewer 1 for these insightful suggestions for improving the quality of our manuscript. We therefore include a brief introduction of ICP-MS as well as other elemental analysis techniques in the Introduction section. Additionally, we include a paragraph with regard to the advantages and limitations of this study in the Discussion section.

Review Comment 4:

“The emphasis here is not on the GalNSe monosaccharides – however, it needs to be stated that for these sugars, it is not clear whether they are taken up by the mammalian GalNAc salvage pathway. Multiple papers have addressed metabolic compatibility in the

past, and I would urge the authors not to overlook this.”

Response: We acknowledge the reviewer for this critical comment. We believe it would be appropriate to mention the GalNAc salvage pathway not only for literature inclusion but also for scientific clarity. A brief discussion on this pathway, and how it would alter the metabolic compatibility can be found in the Discussion section that summarizes the limitations of this study.

Review Comment 5:

“Fig. 2b, being able to detect signal with low nanomolar monosaccharide concentrations is impressive but it is not clear what this is being compared to when stating that it is 5 orders of magnitude better compared to bioorthogonal sugars (p. 5). Furthermore, how does this fit to the authors using 50-200 μ M concentrations for the rest of the manuscript which is the same concentration range as for conventional MOE probes?”

Response: We apologize for the misunderstandings this sentence brings. The classical MOE strategy, in general, uses three “gold standard” characterization methodologies, *i.e.*, confocal fluorescent microscopy, flow cytometry, and anti-biotin Western blot and/or in-gel fluorescence imaging after click reaction with either complementary biotin-tag or fluorescently-labeled tag (*e.g.*, alkyne-biotin or AF488-alkyne).

Based on previously reported literature, typically, millimolar (mM) concentrations of unprotected sialic acid analogues (ManNAz, SiaNAz), or micromolar (μ M) concentrations of peracetylated, or *O*-acylated N-acetyl hexosamine analogues (Ac₄ManNAz, 1,6-Pr₂GalNAz, 1,6-Pr₂ManNAz) are needed in order to achieve a significant level of metabolic labeling (*Nat. Commun.*, **2019**, 10(1): 4065; *Chem. Euro. J.*, **2023**, 29(11): e202203054.; *ChemBioChem*, **2004**, 5(3): 371-374.) (**Response Letter Figure 1-3**). The 5 orders of magnitude being compared means that when we set the signal-to-noise ratio at 6.2 as the distinguishable “real signal” in contrast to “background”, the signal readout from MOE (10-50 μ M range for 1,6-Pr₂ManNAz) is 5 orders of magnitude higher than SeMOE (<1 nM), even though the signal readouts between MOE and SeMOE are not directly comparable.

We choose the 50-200 μ M concentration of SeMOE probes based on three reasons: (1) Though SeMOE probes exhibit low nanomolar detection limitation, micromolar probes are still needed to achieve labeling efficacy to the most extent, and provide a better comparison with conventional MOE; (2) The usage of bifunctional sugars (azidoseleno sugar probes) in this work requires adequate azide density after metabolic incorporation, to achieve successful pull-down experiments. To guarantee the azide level, we chose the micromolar concentration range for experiments; (3) The *in vitro* system herein minimizes the possible interferences from selenoprotein(s), but not in the *in vivo* experiment settings. We used the conventional level of MOE probe concentration (as calculated based on our *in vitro* experiment, and empirically referred from previous MOE experiments *in vivo*), to increase the contrast between SeMOE probes and native selenoprotein background. In other words,

the usage of SeMOE probes at nanomolar concentration would make it more difficult in the transition from *in vitro* to *in vivo* experiments.

Still, we thank the reviewer for noticing this description inconsistency. We therefore clarify and streamline the manuscript accordingly.

Supplementary Fig. 7 | The dose- and time- dependence of 1,3-Pr₂GalNAz labeling in HeLa cells. **a**, In-gel fluorescence scanning showing HeLa cells incubated with respective unnatural sugars at varied concentrations for 48 h, followed by reaction with alkyne-Cy5. **b**, In-gel fluorescence scanning showing HeLa cells incubated with 100 μM Pr₂GalNAz (**4**) for up to 48 h, followed by reaction with alkyne-Cy5. CBB-stained gel in **a** and **b** demonstrate equal loading. Representative results are shown from three independent experiments. Source data for figures **a** and **b** are provided as a Source Data file.

Response Letter Figure 1. Metabolic efficacy of 1,3-Pr₂GalNAz. Excerpted from *Nat. Commun.*, 2019, 10(1): 4065. <https://doi.org/10.1038/s41467-019-11942-y>

[redacted]

Response Letter Figure 2. Metabolic efficacy of 1,6-Pr₂ManNAz. Excerpted from *Chem. Euro. J.*, **2023**, 29(11): e202203054. <https://doi.org/10.1002/chem.202203054>

[redacted]

Response Letter Figure 3. Metabolic efficacy of SiaNAz. Excerpted from *ChemBioChem*, **2004**, 5(3): 371-374. <https://doi.org/10.1002/cbic.200300789>

Review Comment 6:

"It should be stated what the current drawbacks are. ICP-MS is common but not common"

enough to compete with current technology in imaging and MS. For MS efforts, I would expect some kind of statement to realistically state the drawbacks, i.e. dynamic range, signal, number of glycopeptides... it is really not a problem if SeMOE does not have the same sensitivity/capabilities yet as other approaches as it is a new technique, but I would appreciate if the authors stated where they see room for improvement.”

Similar comment was raised in Reviewer 2 Comment 2.

Response: We thank the Reviewer for pointing out the applicability limitation of ICP-MS. We have added the discussion about the limitations and improvements in the Discussion section. We also appreciate the candidness and encouragement of our work from this reviewer. For MS efforts, the current version of SeMOE in tandem with glycoproteomic analysis is readily applicable to intact *N*-glycoproteomics analysis, as well as the SESTAR++ algorithm based on Se isotopic signatures. However, enrichment of labelled proteins is still a must, and therefore bifunctional SeMOE probe with bioorthogonal chemical reporters for enrichment is used. Despite these limitations, we have been able to identify a considerable number of labeled glycopeptides, as exemplified in the main text of the manuscript. Of course, the pursuit of better efficacy and accuracy in glycoproteomics will always be the next step. Our team is working on developing the next-generation of SeMOE probes that a reactive selenium element is installed onto metabolic precursors, which we hope would quantify and enrich labelled glycoproteins at ease. In addition, update of algorithms that can distinguish selenium-isotope signals without the need of enrichment is currently under pursuit in our lab too. We believe that there is plenty of room for improvement, and these topics are included in the updated paper in the Discussion section.

Review Comment 7:

“In Fig. 4, I don’t follow what was APC labelled and what was PE labelled. The figure legends are not really informative, e.g. “Seleno-sialic acid transfer from RMI to RAW264.7 cells” and similar vague legends need to be expanded to show what was done. In the ICP-MS experiments, the authors need to bring the y axis into context, i.e. what are common values observed here, what is expected?”

Response: We feel sorry for the misunderstandings the reviewer encountered while reading the figure legend. The DiI-PE and DiD-APC used here are classical cell membrane staining trackers, which we used to label Jurkat and K562 cells, respectively. We have rewritten the figure legend and main text to provide and clarify the necessary information for figure interpretation.

With regard to the y-axis of the ICP-MS experiment, due to the scarce report of the absolute quantification of metabolic glycan precursors using ICP-MS, it would be difficult to provide the expected or common values in this scenario. However, we believe the comment that the respected reviewer gave is actually critical. Although the amount of sialic acid and fucose cannot be directly compared and depends on different cell lines, here is a case for

reference. A previous report (*iScience*, **2021**, 24, 102397.) used a stable europium (Eu) isotopic mass signal to label azido-modified fucose on cells to quantitate the fucosylation event. Based on this result, the absolute Eu intensity (which represents metabolically incorporated azido-fucose) from 3000 independent single cell runs is 5~100 x 10⁻¹⁸ mol/cell (**Response Letter Figure 4**). Even though a direct comparison would not be feasible between Eu and Se signals, the “unit incorporation amount” we observed in **Figure 4f** and **Supplementary Figure 17**, based on Se ICP-MS, is 100~2000 x 10⁻¹⁸ mol/cell in various cancer cell lines, which is comparable to previous literature, and suggests that SeMOE is able to compensate for the missing signals due to click reaction efficiency. In addition, we systematically evaluated the SeMOE incorporation efficacy in **Supplementary Figure 17**, which should provide perceptual knowledge about these values. A proper citation with the pertinent literature and the Supplementary Figure, as well as a brief discussion about the y-axis information, will now be included in the revised manuscript.

[redacted]

Response Letter Figure 4. Quantification of azido-fucose based on Europium ICP-MS. Excerpted from *iScience*, **2021**, 24, 102397. <https://doi.org/10.1016/j.isci.2021.102397>

Supplementary Fig. 17 Systematic evaluation of SeMOE in various cancer cell lines. The cells were treated with respective selenosugars at indicated concentrations for 48 h, and analyzed using ICP-MS. Error bars represent mean \pm SD of n= 3 independent biological replicates. Source data are provided as a Source Data file.

Review Comment 8:

“At some point, the authors switch to using the free monosaccharides which they need to use much higher concentrations of. This is especially true in the ROS experiments. I am not sure why this was done as an increase in concentration would also lead to an increase in ROS production. Did the authors also try with per-acetylated compounds?”

Similar comment was raised in Reviewer 2 Comment 14, and Reviewer 4 Comment 4

Response: We thank the reviewer for noticing this change. We systematically evaluated the impact of all SeMOE probes on ROS, including the per-acetylated compounds, in various cell lines (**Supplementary Figure 25**), and noticed that free monosaccharides generally trigger higher ROS responses. We admit that the data between free monosaccharide and its protected counterpart is not directly comparable in the original data presentation. To optimize and figure it out, we treated MCF-7 cells with 10 μM of vehicle, Na_2SeO_3 , and corresponding monosaccharides, to measure the ROS assay at the same level (**Supplementary Figure 26e**). The results indicated that an increase in ROS production by the free monosaccharide is an intrinsic property, rather than discrepancies in concentration increases.

Supplementary Fig. 25 ROS assay of various cell lines treated with respective SeMOE probes at indicated concentrations for 24 h. The ROS level was assayed using flow cytometry. Error bars represent mean \pm SD of n=3 independent biological replicates. Source data including exact *P* values are provided as a Source Data file.

Supplementary Fig. 26e ROS assay of MCF-7 cells treated with 10 μ M respective monosaccharides for 1 h. Error bars represent mean \pm SD of n=3 independent biological replicates. **P* < 0.05, ***P* < 0.01, ****P* < 0.001, ns, not significant (one-way ANOVA followed by post hoc Dunnett's test). Source data are provided as a Source Data file.

Review Comment 9:

“Some phrases or sentences are off e.g. “nibbled” or “The interdisciplinary thought is vivified”. In the SI, please take care to include volumes of buffers, numbers of cells, confluency, types of plates etc. used in all experiments so anyone can repeat the experiment. For instance, the statement “MCF-7 cells were seeded in 6-well plates and incubated in 1640 medium overnight. The cells were collected, treated with vehicle or selenosugars as indicated for 36 h.” needs much more information to be reproducible.”

Similar comment was raised in Reviewer 2 Comment 20.

Response: We appreciate the professionalism our reviewer has put in improving the quality of the manuscript. We have fixed these phrases accordingly. We also realize that some claims in the previous manuscript seemed inaccurate and overstated. We apologize for the misunderstandings that the original claim brings. Language in both the main text and the Supporting Information has also been grammatically adjusted to eradicate the misunderstanding of our research. We also streamline the Methods section by adding more detailed information such as buffer volume, cell number and confluency, for better understanding and experiment repeatability.

Review Comment 10:

“In my view, the potential anti-cancer application is a bit too preliminary to be included in the title of the paper.”

Similar comment was raised in Reviewer 2 Comment 14.

Response: We thank the Reviewer for this valuable suggestion. We changed the original manuscript topic into “Selenium-based metabolic oligosaccharide engineering strategy for quantitative glycan detection”, in order to clarify the methodological development nature of this work. Figures related to the potential anti-cancer application of SeMOE probes are now moved into **Supplementary Figures 25-28**.

Taken together, we wish to express our gratitude again to the respected Referee 1 for providing truly insightful suggestions.

Referee:2

Review Comment 1:

“In this manuscript “SeMOE allows for quantitative glycan perception and exhibits anti-cancer potentiality” the authors describe the synthesis and in vitro/vivo evaluation of selenium-containing sialic acid precursors, the presence of the selenium atom permitting quantitation by ICP-MS. The most useful innovation in this research, albeit one that is not well elaborated in the manuscript, is that the ICP-MS permits absolute glycan quantitation, the ICP MS source effectively eliminating the different response factors between different glycans and even different classes of glycoconjugates. However, there are some serious problems with the manuscript in its current form, both scientific and stylistic. These issues must be addressed before publication in Nat. Comm.”

Response: We are grateful for the overall positive feedback, invaluable expertise, and superb professionalism of the respected Referee 2. We acknowledge the reviewer for this insightful yet concise innovation summary of our method, which we have added in the Discussion section, to transparently discuss the advantages and limitations of this study, in the revised manuscript. We have thoroughly revised the whole manuscript, both scientifically by addressing useful comments in experiments and grammatically eradicating the misunderstandings, overstatements or inaccuracies in languages. As a relatively young PI at an early career stage, cautions should be made to ensure the validation of an argument in science and to avoid de validation based on exaggeration. We hope with these necessary quality improvement processes, the paper is now suitable for publication. We thank you for your encouragement as well as the spurs on our manuscript.

Review Comment 2:

“Aside from the ease of absolute quantitation (without external instrument calibration) there is nothing that the SeMOE approach brings to the glycoscience community that cannot be achieved by other, currently existing tools. A major (if not the major) impetus behind the entire biorthogonal-labelling field was that the method permitted the enrichment/isolation of tagged glycoconjugates for subsequent analysis and/or tracking in live cells. The only way that SeMOE permits these classic MOE experiments is by still retaining the azide functional group. Thus, the sole innovation of the SeMOE is quantitative.”

Similar comment was raised in Reviewer 1 Comment 6.

Response: We thank the reviewer for the thoughtful consideration, and hope to clarify several points. The reviewer correctly points out the advantage of absolute quantification that SeMOE probes bring, and understandably notes that some researchers may alternatively use currently existing tools, such as bioorthogonal reactions and DMB analysis, to indirectly achieve glycan quantification (*iScience*, **2021**, 24, 102397. <https://doi.org/10.1016/j.isci.2021.102397>; *Anal Chem.* **2017**, 89, 538–543. <https://doi.org/10.1021/acs.analchem.6b04141>; *ChemBioChem.* **2021**, 22, 1243–1251.

<https://doi.org/10.1002/cbic.202000715>). These works are properly acknowledged in the reference list. We have **zero intention** of belittling the classic MOE method (which is a groundbreaking work in current Chemical Biology), but rather believe that it is the classical MOE that provides the inspiration for this work, and this label-free SeMOE strategy can provide more information with regard to a glycan-participating event such as glycan mass transfer in trophocytosis and biodistribution of sialic acids in living mice, which provides ease of secondary labeling with azides.

We appreciate the candidness of our work from this reviewer. We agree with the reviewer that chemical handle in classical MOE (e.g., azide) provides the versatility and the opportunity for tagged glycoconjugates' enrichment/isolation. For MS efforts, the current version of SeMOE in tandem with glycoproteomic analysis is readily applicable to intact *N*-glycoproteomics analysis, as well as the SESTAR++ algorithm based on Se isotopic signatures. However, enrichment of labelled proteins is still a must, and therefore bifunctional SeMOE probe with bioorthogonal chemical reporters for enrichment is used without better choice. We agree with the reviewer that the pursuit of better efficacy and accuracy in glycoproteomics, as well as technical innovations will always be the next step. Our team is working on developing the next-generation of SeMOE probes that a reactive selenium element is installed onto metabolic precursors, which we hope would **simultaneously** achieve quantification and enrichment of labelled glycoproteins at ease. In addition, an update of algorithms that can distinguish selenium-isotope signals without the need of enrichment (unfortunately not applicable in the SESTAR++ setting) is currently under pursuit in our lab too. We believe that there is plenty of room for improvement, and these topics are included in the updated paper in the Discussion section. Positive comments from Reviewer 1 and 4 also gained us more confidence in this method, as excerpted below (quotation):

"I would expect some kind of statement to realistically state the drawbacks, i.e. dynamic range, signal, number of glycopeptides... it is really not a problem if SeMOE does not have the same sensitivity/capabilities yet as other approaches as it is a new technique, but I would appreciate if the authors stated where they see room for improvement."

"The main advance in this study is the direct detection of the non-natural glycan through detection of the inherent selenium atom(s); this avoids cumbersome second-step labeling methods required with "classic" MOE approaches. As an important bonus, the method is efficient, with metabolic incorporation (most likely) of the selenosugars more facile than the widely-used azido-modified analogs that are currently workhorse reagents in the field."

Review Comment 3:

"In other words, the authors do almost nothing to indicate why a researcher should adopt this tool: no urgent research need is presented in the introduction, and the conclusion is highly exaggerated. ["We believe that the SeMOE strategy can serve as the Occam's razor [...]". Occam's razor is a philosophical concept used by scientists to choose between (or

make) hypotheses; it is not a chemical biology research tool. If the authors believe their tool is simpler than current MOE strategies (hence their reference to Occam) they are mistaken! There is nothing simple about this approach, either biochemically or analytically: the SeMOE approach requires sophisticated and expensive analytical tools that are virtually unavailable to nearly all life science researchers in most institutions. Please, remove reference to Occam!].”

Response: We want to sincerely apologize for the misuse of the Occam razor metaphor. Glycans serve as key checkpoints in immunology and shapers of tumor microenvironment. ***How the exchange of biomolecules such as glycan in these events affects the biological consequences*** was still hampered by analytical challenges. The platform we report here bridges this gap by using SeMOE. Again, we have **zero intention** of belittling the classic MOE method. By “simplicity” of this study, what we actually want to express is that this label-free SeMOE strategy can provide more information with regard to glycan-participating events without secondary azide labeling, but rather than the instrument setting for ICP-MS detection is “simple”. Of course, such misunderstandings should be avoided throughout the paper. We believe the development and implementation of tools hold the potential for broad applicability in biology and medicine, as exemplified by the ROS assay (although the data is preliminary). Our trial is only the first step toward a better goal. However, we value these critical comments, and admit that we may have confused the readers/reviewers about the need of this paper, and the manuscript writing definitely needs improvement. Hence, we provide more information in the Introduction section for better scientific elucidation and reading experience. Language in both the main text, and the Supporting Information has also been grammatically adjusted to eradicate the misunderstanding of our research.

Review Comment 4:

“With respect to both absolute quantitation and low detection limit, radiolabeling approaches exist [<https://doi.org/10.1002/0471142727.mb1704s87>] which should be acknowledged. Additionally, the radiolabeling approach does not encounter issues about the tolerability of the unnatural Se-containing group appended to the mannosamine (see below). In sum, the manuscript in its current form gives no real reason for investing in SeMOE, other than that it can be done.”

Response: We thank the reviewer for this information input. This work, and related work we deem necessary, is now properly cited in the revised manuscript. The radiolabeling approach is no doubt a good strategy to achieve absolute quantification and low detection limit. Besides, as the reviewer mentioned, no chemical derivatization or modification of the monosaccharide precursor is needed when radiolabeling methods are applied. Still, isotopic labeling possesses limitations such as high cost and lab safety (radioactivity). Also, challenges in dynamic and live imaging using radiolabeled probes, although partially evaded using SPECT/CT or secondary ion mass spectroscopy (SIMS), still restrict the

applicability for absolute quantification. The merit of MOE, or in general, the chemical reporter strategy, from our point of view, is that by adding chemical handles onto the monosaccharide precursor for versatility, a sacrifice in undesired perturbation (though minimal, but still exists) is acceptable. For instance, a recent report on protein S-glycomodification (*J. Am. Chem. Soc.* **2020**, 142, 20, 9382. <https://doi.org/10.1021/jacs.0c02110>) demonstrated side reactions from MOE, which indicated that cautions should be made even with the classical MOE azidosugars, and the next step toward a “better” glycan detection strategy is always a pursuit of chemical glycobiologists. We believe the biocompatibility, the ICP-MS detection modality, the unique isotope signature, as well as the detection of limitation in SeMOE, although not perfect, will also provide insights into chemical glycobiology.

Review Comment 5:

“Se is a large atom. We know that subtle changes to the N-acetimido group of mannosamine influence the rate of incorporation of the corresponding sialic acid [https://www.nature.com/articles/s41598-022-26521-3]. The authors should compare by in gel fluorescence (like Fig 2j) 9AzSiaNSe and 9AzSia to verify. Likewise, the unnatural CMP-Neu5Ac analogues may, inadvertently, be inhibitors of sialyltransferases and/or the enzymes in the CMP-Neu5Ac pathway: a lectin blot or flow cytometry experiment should be performed to demonstrate that these SeMOE tools do not perturb the formation of sialosides. The utility of SeMOE, like MOE, is lost if these reagents influence the normal biosynthesis of glycoconjugates: I believe they do and it is erroneous to assume otherwise in the absence of evidence.”

Similar comment was raised in Reviewer 4 Comment 2.

Response: We thank the reviewer for raising these pertinent suggestions to improve the quality of this manuscript. The N-acyl group does influence the incorporation rate of the corresponding sialic acid (*Sci Rep*, **2022**, 12, 22129; *ChemBioChem*, **2004**, 5(3): 371., both papers are properly cited after revision), and it should be imperative to measure the metabolic efficiency of selenium-containing metabolic precursors in our system. To begin with, the DMB derivatization experiment confirms that SeMOE probes exhibit similar incorporation rates compared to classical MOE azidosugars, and no significant perturbation on native sialic acid levels is observed (**Fig. 2h**). These results show decent tolerance for SeMOE probes. Besides, we have performed in-gel fluorescence, confocal imaging and flow cytometry analysis on 9AzSiaNSe and 9AzSia, two probes with the discrepancy in N-acyl position (**Supplementary Fig.12a-e**). The results showed similar labeling efficiency of 9AzSiaNSe and 9AzSia, based on azide signal output. In addition, to test whether Se-containing analogues serve as inhibitors, we performed a lectin flow cytometry experiment, and found no significant fluctuation of α 2,6-linked (SNA lectin) and α 2,3-linked (with preference to Sia α 2-3Gal β 1-3(\pm Sia α 2-6)GalNAc, MAL-II lectin) sialic acid on the cell surface (**Supplementary Fig. 2**). These results collectively prove that SeMOE probes preserved as a metabolic precursor, rather than an inhibitor, in sialic acid

biosynthesis pathway.

Fig. 2h DMB-derived sialic acid analysis of glycoproteins from HeLa cells treated with vehicle, 2 mM ManNSe, 2 mM SiaNSe, 200 μ M 1,6-Pr₂ManNSe and 100 μ M Ac₄ManNSe for 48 h, respectively. Peak 1, 2 and 3 represent Neu5Gc, Neu5Ac and SiaNSe, respectively. The metabolic incorporation rate is calculated as the peak area ratio between peak 3 and the sum of peak 1, 2, and 3.

Supplementary Fig. 12 Metabolic labeling of HeLa cells by 9AzSiaNSe. a-b, Confocal fluorescence imaging (a) or in-gel fluorescence scanning (b) of HeLa cells treated with 9AzSiaNSe at indicated concentrations for 48 h, respectively. c-d, Confocal fluorescence imaging (c) or in-gel fluorescence scanning (d) of HeLa cells treated with 2 mM 9AzSiaNSe for indicated time. e, Flow cytometry analysis of 9AzSiaNSe-or 9AzSia-treated HeLa cells. f, Confocal fluorescence imaging of 9AzSiaNSe-labelled Hela cells treated with or without sialidase, followed by reaction with alkyne-Alexa 488. Scale bar: 10 μ m. g, Flow cytometry analysis of HeLa cells treated with 9AzSiaNSe and Sia at indicated concentrations for 48 h. Error bars represent mean \pm SD of n= 3 independent biological replicates. * P < 0.05, ** P < 0.01, *** P < 0.001, ns, not significant (One-way ANOVA, post hoc Dunnett's test). Source data including P values are provided as a Source Data file.

Supplementary Fig. 2 Flow cytometry analysis of sialic acid levels of SeMOE probe-treated cells. α2,6-linked and α2,3-linked sialic acids were recognized by *Sambucus nigra* (SNA) and *Maackia amurensis* leucoagglutinin II (MALII), respectively. a-b, Flow cytometry analysis of α2,6-linked sialic acids (a) and α2,3-linked sialic acids (b) of SeMOE probe-treated HeLa cells. c-d, Flow cytometry analysis of α2,6-linked sialic acids (c) and α2,3-linked sialic acids (d) of SeMOE probe-treated MCF-7 cells. Error bars represent mean ± SD of n=3 independent biological replicates. **P* < 0.05, ***P* < 0.01, ****P* < 0.001, ns, not significant (one-way ANOVA followed by post hoc Dunnett's test). Source data including exact *P* values are provided as a Source Data file.

Review Comment 6:

“The flow cytometry data (Fig 2d and Sup. Fig. S4) should be displayed as histograms as is conventional (additionally, the number of events in these figures should be clearly indicated). Presenting the data as scatter plots obscures the fact that in 1,6-Pr₂ManNSe-treated cells (Fig 2d) the vast majority of events display a Se signal essentially undistinguishable from the control. Have the authors an explanation for this? This phenomenon is also inconsistent with the linear relationship presented in Fig 2e (and Sup. Fig S5): some form of gating must have happened here, correct?”

Similar comment was raised in Reviewer 3 Comment 3.

Response: We thank the reviewer for pointing these out. We agree with the reviewer that the histogram is a better way to present flow cytometric data. The original CyTOF results were based on 293T cells and not ideal (Se signal essentially undistinguishable from the control). We attribute this result to the relative low incorporation rate of SeMOE precursors compared to other cell lines. We now redo the CyTOF experiment in HeLa cells and MCF-7 cells, respectively, and display the data as histograms. The Se signal is now essentially distinguishable from the control (**Fig. 2d, Supplementary Fig. 5**).

In the original manuscript, Fig. 2d shows the CyTOF analysis of SeMOE-treated 293T cells, while Fig. 2e shows the ICP-MS analysis of various amounts of HeLa cells treated with SeMOE. In view that both the analyzing tool and cell lines are varied, these two results are inconsistent and not comparable. We believe the new CyTOF experiment with HeLa and MCF-7 is now clear and concise for data presentation, and consistent with other experiment results. We also add a note to specify the gating strategy and analyze procedures (**Supplementary Note 3**) both in the Fig. 2d legend and the Methods section.

Fig.2d (left) and Supplementary Fig. 5 (right) CyTOF analysis of SeMOE probe-treated HeLa cells and MCF-7 cells. Cells were treated with vehicle, 2 mM ManNSe, 2 mM SiaNSe, 2 mM 9AzSiaNSe, 200 μM 1,6-Pr₂ManNSe, and 100 μM Ac₄ManNSe for 48 h, respectively, followed by CyTOF analysis. m/z at 78 was used for calculation of ⁷⁸Se. At least 30,000 events were gated and analyzed according to the CyTOF gating strategy (Supplementary Note 3).

Supplementary Note 3 CyTOF gating strategy

Review Comment 7:

“*ManNAc (and analogues) is converted to GlcNAc by most cells [https://doi.org/10.1074/jbc.M212127200]; GlcNAc-6-phosphate, meanwhile, is a substrate for a de-Nacylase while UDP-GlcNAc (and analogues) is epimerized to UDP-GalNAc. Thus, there is ample opportunity for the Se to end up in various glycoconjugates. Could this be why almost one third of the signal from Ac₄ManNSe is retained after neuraminidase treatment (Fig 2f)?*”

Similar comment was raised in Reviewer 3 Comment 4.

Response: We thank the reviewer for proposing this interesting hypothesis. We believe that this is also a good way to gain knowledge through communication. According to the biosynthesis and interconversion pathway of monosaccharides (Essentials of Glycobiology, 4th edition. <https://www.ncbi.nlm.nih.gov/books/NBK579932/>), GlcNAc is able to convert to ManNAc but not vice versa (**Response Letter Figure 5**). Yet paradigm shifts for this textbook knowledge based on new researches (*J Biol Chem*, **2003**, 278(10): 8035-8042. <https://doi.org/10.1074/jbc.M212127200>; *ACS Chem Biol*, **2020**, 15, 10, 2692–2701. <https://doi.org/10.1021/acscchembio.0c00453>) (**Response Letter Figure 6**). We assume the reviewer raised an interesting open question herein.

Combined with a series of validation experiments including enzymatic treatment, metabolic competition and DMB-derived sialic acid analysis we did in this work, we validate that the Se signals indeed represent sialo-glycoconjugates (at least to some extent) under physiological conditions. Currently, because we do not have enough evidence, it's too preliminary to say whether this hypothesis the respected reviewer raised is true or not. Several other possible explanations for why signal from SeMOE probes is retained after neuraminidase treatment is proposed here: (1) Side reaction between Ac₄ManNSe and cysteines on cell-surface proteins (protein S-glyco-modification, *J. Am. Chem. Soc.* **2020**,

142, 20, 9382.); (2) Sialidase specificity toward cell-surface selenium-containing sialoglycans may be low. (3) Monosaccharide cross-talk (the selenium-containing sugar resides on cell-surface glycoconjugates, but not in the form of sialic acid). We believe a thorough investigation based on HPAEC-PAD or selenium speciation would give the answer. We performed selenium speciation on selenocysteine (SeCys₂), Se-methyl-L-selenocysteine (MeSeCys), selenomethionine (SeMet), selenate [SeO₄²⁻], and selenite [SeO₃²⁻], five compounds associated with selenoprotein conversion. Major organs administered with or without SeMOE probes displayed near-identical selenium speciation, suggesting that injection of seleno-sialoglycans will not significantly affect the selenoprotein constituents in mice (**Supplementary Fig. 23**). In addition, we do not observe significant changes in major selenoprotein expression level *in vitro*, as evidenced by RT-qPCR analysis (**Supplementary Fig. 3**).

[redacted]

Response Letter Figure 5 Biosynthesis and interconversion of monosaccharides, excerpted from Essentials of Glycobiology, 4th edition. <https://www.ncbi.nlm.nih.gov/books/NBK579932/>

[redacted]

Response Letter Figure 6 Dynamic conversion of non-natural monosaccharides within the cellular environment and the non-natural metabolite utilization pathways observed in cells. In vivo incorporation of Ac₄ManNAz as SiaNAz, Ac₄GalNAz as GalNAz and GlcNAz (epimerization of UDP-GalNAz by GALE enzyme), and Ac₄GlcNAz as GlcNAz and SiaNAz (facile conversion of GlcNAc to ManNAz). *ACS Chem Biol*, **2020**, 15, 10, 2692–2701. <https://doi.org/10.1021/acscchembio.0c00453>

Supplementary Fig. 3 Selenoprotein mRNA levels of SeMOE-treated cells. HeLa cells were treated with PBS, 2 mM ManNSe, 2 mM SiaNSe, 2 mM 9AzSiaNSe, 200 μ M 1,6-Pr₂ManNSe, 200 μ M 1,6-Pr₂GalNSe, 100 μ M Ac₄ManNSe or 100 μ M Ac₄GalNSe for 48 h, respectively, followed by RT-qPCR analysis. n=4. Source data are provided as a Source Data file.

Supplementary Fig. 23 Selenoamino acid levels of SeMOE-treated mice. a-e, Male BALB/c mice (8 to 10 weeks old) were once-daily, intraperitoneally injected with Ac₄ManNSe (160 mg Ac₄ManNSe/kg/day), while control mice received the 70% DMSO alone. On day 5, mice were euthanized. The heart, liver, spleen, lung and kidney of mice were collected, digested by proteinase K and trypsin, followed by selenoamino acid analysis via ICP-MS. f, Comparison of the selenoamino acid peak integral between the Control group and SeMOE group. Error bars represent mean \pm SD of n=3 independent biological replicates. **P* < 0.05, ***P* < 0.01, ****P* < 0.001, ns, not significant (two-tailed Student's t-test). Source data are provided as a Source Data file.

Review Comment 8:

“Statistics: Some statement of how stats were performed should be included in the methods section. Secondly, throughout the manuscript two-way ANOVAs are performed, eg. Fig 2f comparing untreated to neuraminidase cell surface and neuraminidase cell lysates. This should be recalculated with a one-way ANOVA followed by an appropriate post hoc test (like Tukey). A two-way test could be used to compare the different SeMOE reagents and neuraminidase treatment. Alternatively, if multiple comparisons are made to a single control (as is frequently the case in this manuscript) then a Dunnett’s test should be performed. In any case, the two-way test is inappropriate in all cases I could find in this paper.”

Response: We are very grateful for this meticulousness and professional suggestion from the reviewer. We apologize for the confusing statement of statistics in the submitted manuscript. We have checked and reanalyzed all data, and added statements of statistics. We deem it necessary and correct now (updated in the Methods section and figure legends): “Experiments in this study were performed at least in triplicate. Each dot represents an individual sample. Statistical analysis was performed using GraphPad Prism and Origin. Error bars in all statistical data represent mean \pm standard deviation (SD). Comparison between the two groups was analyzed by Student’s two-tailed t-test when not otherwise specified. Differences among multiple groups were analyzed by One-Way ANOVA test followed by post hoc Dunnett’s test.”

Review Comment 9:

“Figure 3d attempts to present a mass balance for Se by using a pie graph, a format that does not permit an indication of the standard deviation of each measurement. More importantly, though, the experimental section indicates that each class of glycoconjugate appears to have been acquired separately, that is a protein fraction was collected from a separate set of samples as the lipid and RNA (and no details are provided about how the “other forms” of Se were deduced. If I understand the protocol correctly, then the parts-of-a-whole pie graph are misleading, because no attempt at making a mass balance was performed. To properly do this, separate fractions from the same group of cells should be measured and then normalized.”

Response: We thank the sharp yet helpful comment from the reviewer. We have replaced the pie graph with a column graph, and displayed the standard deviation in the updated **Fig. 3d**. In fact, we indeed separated fractions from the same group of cells, and the measurement by ICP-MS for Se levels are normalized by cell numbers. In brief, MCF-7 cells were treated with or without 200 μ M 1,6-Pr₂ManNSe for varied time (6 h, 12 h, 24 h, 36 h, 48 h). After co-incubation, 1.5×10^7 cells were trypsinized, washed three times with PBS, counted by the automated cell counter (Invitrogen), and evenly divided into three equal aliquots for protein, lipid and RNA extraction, respectively. The samples of protein, lipid and RNA isolated from the same group were analyzed by ICP-MS, and Se levels were normalized by the corresponding cell numbers (Please refer to the Methods section).

Fig. 3d Newly-synthesized seleno-sialic acids in different glycoconjugates from MCF-7 cells treated with 200 μ M 1,6-Pr₂ManNSe for indicated time.

Review Comment 10:

“Comparison of the SeMOE tool to SILAC is made with some support provided by the glycoproteomics work. [Note: no mention of the mass spectrometer used for this was provided]. Substitution rates of up to 23% are reported (Fig 3f). As I understand this figure, the number of peptide spectra containing normal Neu5Ac are compared with those containing Se-Neu5Ac. But this isn’t how SILAC is done at all! To get a real substitution rate, the same glyco-peptides need to be compared head-to-head.....it does not appear that this was even attempted. In the present form, these data tell me more about the non-detection rate of the HPLC-MS that they do about the incorporation rate. Please pick several glycopeptides in common across all data sets and compare normal and Se-Neu5Ac quantities in each. You will very likely find that the incorporation rate is glycopeptide-dependent, which will be a reflection of protein half-life. In the present form, these data are not convincing and incompletely analyzed.”

Response: We thank the respected reviewer for insightful comments and would hope to clarify several points. The reviewer may get confused with the experiment settings we used in the glycoproteomic analysis herein. We did not apply the SILAC technique, a good one based on isotopic signature too in this experiment, but used SESTAR++ (*ACS Cent Sci.* **2018**,22,4(8):960; *Methods Enzymol.* **2022**, 662:241), a selenium-encoded isotopic signature targeted profiling which utilizes the distinct natural isotopic distribution of selenium to assist detection of trace selenium-containing signals from shotgun proteomic data, and can work seamlessly with chemical-proteomic profiling strategies to enhance identification of post-translational modifications (please refer to **Response Letter Figure 7** for the workflow).

For **Fig. 3f**, the substitution rate reported here is the overall mean percentage of Ac₄ManNSe-originated seleno-sialoglycopeptides divided by the total sialoglycopeptides. That is, the substitution rate = spectrum numbers of seleno-sialoglycopeptides / spectrum numbers of natural sialoglycopeptides x100%. We apologize for the confusion in the data presentation. A more detailed method description for the algorithm we use in SESTAR is now included in the revised paper.

As requested by the respected reviewer, we have randomly picked ten glycopeptides across all data sets (acquired via pGlyco3, an intact-glycopeptide analysis algorithm), compared their seleno-Sia incorporation rates individually, and indeed found that the incorporation rate is glyco-peptide-dependent, which is, as perfectly explained by the reviewer, a reflection of protein half-life. We have added the statement of this finding in the manuscript (Please also refer to **Supplementary Fig.13**).

[redacted]

Response Letter Figure 7 Workflow of SESTAR, excerpted *ACS Cent Sci.* **2018,22,4(8):960.**

Number	Peptide	Glycan composition	Protein	Gene
#1	GYVJQSEAGSHTIQR		Q95604	HLA-C
#2	GYVJQSEAGSHTIQR		Q95604	HLA-C
#3	EAGJHTSGAGLVQJK		P20645	M6PR
#4	VTGLJCTTNHPINPK		P08648	ITGA5
#5	AFJSTLPTMAQMEK		P16070	CD44
#6	AFJSTLPTMAQMEK		P16070	CD44
#7	GYVJQSEAGSHTIQR		Q95604	HLA-C
#8	VTGLJCTTNHPINPK		P08648	ITGA5
#9	KDFEDLYTPVJGSIVIVR		P02786	TFRC
#10	KEJSSEICSNNGECVCGQCVCVR		P05556	ITGB1

Supplementary Fig. 13 Examples of glycopeptides in pGlyco3 analysis. a, peptide sequences and glycan compositions of ten random glycopeptides in common across all data sets in pGlyco3 analysis. b-c, Spectra numbers of seleno-Sia and natural sia in glycopeptide #1 to #8. The percentages in blue font represent seleno-Sia incorporation rates of different glycopeptides. For glycopeptide #9 and #10, no seleno-sia were found (shown in the source data). Source data are provided in the Source Data file.

Review Comment 11:

“Fig. 3h demonstrates the presence of a Se isotope window of 8 m/z, yet the experimental

protocols suggest that the HCD-MS/MS isolation window was 2 m/z. How is this possible? If precursor ions were selected across a 2 m/z range, then at most only two Se atoms will be present.”

Response: We thank the reviewer for pointing this out. m/z represents the mass-to-charge, and we set 2 m/z as the precursor isolation window. Considering that targeted precursors usually carrying multiple charges, the actually isolation window was actually wider than 2 Da. These precursors were dissociated in MS2, and the fragment ions usually carrying single charge. Therefore, a Se isotope window of 8 m/z in a MS2 spectrum is possible.

Review Comment 12:

“There are many over-stated or unsupported sentences in the manuscript. Here are several: (1) “And the unique property of selenium, such as redox response, can be exploited instead to interrogate glycan functions in their native surrounds”. Firstly, this is not a sentence. Secondly, this was not done in this manuscript, in fact, it is not even suggested how this redox activity of selenium might be exploited. Please revise/clarify.”

Response: We thank the tremendous efforts the reviewer has put in for improving our manuscript. We have deleted this sentence.

Review Comment 13:

“(2) “[...] thus implying selenosugars remain intact under protein glycosylation processes”. This sentence is in reference to Fig. 2a. These data do not support this at all (although other experiments do). Clarity is important here. These data indicate that the selenium atom is in these HeLa cells: that is it! Unless I’m mistaken, these are whole cells subjected to ICP-MS after acid digestion. Thus, Fig 2a. shows that the Se is on or in the cells, nothing more. Fig. 2b. does support that the label is in the protein fraction and Fig 2c. does this even more convincingly, but we cannot support the claim that this is in the glycoprotein fraction until the PNGaseF/sialidase tests are used to verify that the Se is associated with sialic acid.”

Response: We have rephrased these sentences in a clear, concise and accurate way. We agree with the reviewer that cautions should be made to ensure the validation of an argument in science, and to avoid devalidation based on exaggeration.

Review Comment 14:

“(3) Potential anticancer effects: this heading is pure wishful thinking. What was exactly done here was a measurement of the ROS generation in a range of immortalized cell lines. Comparing MCF7 to MCF10a simply tells me that these two cell lines are different from each other (which will likely be the case no matter how the ROS generation is induced). Things look even more ambiguous in Supporting Figure S12: cells vary widely in the ROS

concentrations and very few appear to exhibit a concentration-dependence. Worse: ROS capacity, if it is due to the mere presence of the Se atom, should be equivalent for all compounds tested. If, on the other hand, the selenosugar must be incorporated into glycoconjugates to induce ROS, then some attempt needs to be made to normalize ROS levels to the amount of Se contained in the glycoconjugate fraction in each cell. Finally, there needs to be some statement indicating that the amount of ROS generation induced (and this is far from clear) is actually sufficient to kill or inhibit cancer cell growth. Is it? Can these data be correlated with the cell viability assays?”

Similar comment was raised in Reviewer 1 Comment 8 and Reviewer 4 Comment 4.

Response: We agree with the reviewer that the claim of anti-cancer effect is still preliminary in the current experiment settings. We systematically evaluated the labeling efficiency of all SeMOE probes in various cell lines (**Supplementary Fig. 25**). SeMOE probes can indeed promote ROS generation, and noticed that free monosaccharides in general trigger higher ROS responses, though in a manner with no apparent concentration dependence (1 μ M-1 mM). We admit that the data between free monosaccharide and its protected counterpart is not directly comparable in the original data presentation. To test whether the inducement of ROS of the selenium-containing monosaccharide precursors depends on their unique carbohydrate forms rather than the Se atoms alone, we treated MCF-7 cells with 10 μ M of vehicle, Na₂SeO₃, and corresponding monosaccharides, to measure the ROS assay at the same level (**Supplementary Fig. 26e**). The results indicated that the increase in ROS production by the free monosaccharide is an intrinsic property, rather than discrepancies in concentration increases and Se atom/ion/metabolite alone.

After a thorough discussion with all the authors, we unanimously decide to change the original manuscript topic into “Selenium-based metabolic oligosaccharide engineering strategy for quantitative glycan detection”, in order to clarify the methodological development nature of this work. Figures related to the potential anti-cancer application of SeMOE probes are now moved into **Supplementary Figs. 25-28**.

Supplementary Fig. 26e ROS assay of MCF-7 cells treated with 10 μ M respective monosaccharides for 1 h. Error bars represent mean \pm SD of n= 3 independent biological replicates. * $P < 0.05$, ** $P < 0.01$, *** $P < 0.001$, ns, not significant (one-way ANOVA followed by post hoc Dunnett's test). Source data are provided as a Source Data file.

Review Comment 15:

“(4) “A plethora of sialoglycoconjugates from Jurkat cells [...] were transferred to K562 cells....”. There is in fact no evidence that sialoglycoconjugates were transferred, let alone “a plethora”. These data show that Se is transferred between the cells, but there are zero controls performed to indicate that the Se is in the form of sialic acid, let alone glycoside-bound sialic acid. If the Ac4ManNSe is inefficiently metabolized—and fig 3d makes it clear that even after 48 h, most of it still exists as “other forms” (i.e. not glycosides)—how can we be sure that the Jurkat cells are not simply excreting the ManNSe for subsequent re-uptake by the K562 cells? Can the K562 cells be neuraminidase treated before flow cytometry? (A loss of some Se signal here would verify that at least some Neu5Ac transfer occurred). Also, why are the controls showing such high Se concentrations here (Fig 4c)?”

Similar comment was raised in Reviewer 3 Comment 8.

Response: We appreciate this valuable suggestion from the reviewer, and performed a sialidase treatment experiment in Jurkat-K562 trogocytosis (Please refer to **Supplementary Fig. 16a**). Before co-incubation, Se-labelled Jurkat cells were treated with sialidase, and then co-incubated with K562 cells with sialidase. After co-incubation, K562 cells were sorted, followed by ICP-MS analysis. The results showed a significant decrease in Se levels, indicating that at least some Neu5Ac transfer occurred, in the sialoside form.

Moreover, we thank the reviewer for reminding us of the high Se level of control in **Fig.4c, d**. We have revisited and repeated the experiment, corrected the graph according to the raw data.

Supplementary Fig. 16a, Seleno-sialic acid transfer from SeMOE probe-labelled Jurkat cells to K562 cells during trogocytosis with or without sialidase treatment. Error bars represent mean \pm SD of $n=3$ independent biological replicates. * $P < 0.05$, ** $P < 0.01$, *** $P < 0.001$, ns, not significant (two-tailed Student's t-test). Source data including P values are provided as a Source Data file.

Fig. 4c-d Seleno-sialic acid transfer from 1,6-Pr₂ManNSe (c) or Ac₄ManNSe-treated (d) Jurkat cells to K562 cells.

Review Comment 16:

“(5) “We also demonstrated that participants in the indirect cellular communication were also O-glycosylated...”. How? This wasn’t demonstrated at all. There are tools/procedures for discriminating between N- and O-glycans, none of which were employed in this manuscript. Please revise this sentence.”

Response: We actually treated cells with 1,6-Pr₂GalNSe, which should be incorporated

into mucin-type O-linked glycans (**Supplementary Fig. 19**). However, we agree with the reviewer that neither β -elimination of O-glycanase (endo- α -N-acetylgalactosaminidase) were used to verify the O-glycosidic linkage due to inaccessibility or lack of necessary protocols. Even with the influence of the GalNAc salvage pathway (**Review Comment 4**) and side reactions, we believe at least some (if not major) 1,6-Pr₂GalNSe can exist in the form of mucin-type O-linked glycans in the cell. This sentence, as well as the discussion with regard to GalNAc salvage pathway, is now added as follows: “We also noticed selenoglycan transfer from 1,6-Pr₂GalNSe-treated cancer cells to RAW 264.7 cells.”

Review Comment 17:

“There are several instances where the authors seem unaware of existing literature. Some rewording is needed here. For example: Describing alkyne-containing carbohydrates as “new-to-nature”. Azides may be new-to-nature, but alkyne containing natural products do exist [<https://doi.org/10.1039/D0CB00190B>]. I would suggest using the term “biorthogonal” to describe these modified carbohydrates. In the following sentence describing “the Novel Prize-winning biorthogonal reaction” I would recommend referring explicitly to the more chemically descriptive cycloaddition reactions used to detect the azide/alkyne-containing carbohydrates.”

Response: We thank the reviewer for the meticulousness with glycoscience and revised these words accordingly.

Review Comment 18:

“[...] which we termed “selenosugars”” The term selenosugar has been in existence for sometime now see <https://doi.org/10.1021/acsomega.1c02160> for example]. Also (this point is debatable but should be noted) “selenosugar” is usually in reference to a carbohydrate where one of the hydroxyl groups is replaced with the selenium atom, analogous to amino-sugars or thio-sugars.”

Response: We thank the reviewer for this professional suggestion, and we have replaced “selenosugar” with “SeMOE probe”, “unnatural selenium-containing monosaccharide” or “seleno-sialoglycan” for clarification.

Review Comment 19:

“Why was the lower abundance ⁷⁸Se for quantitation by ICP-MS?”

Response: We thank the reviewer for this insightful comment. The isotopic envelope pattern of selenium is guaranteed by its six stable isotopes with distinctive distribution (⁷⁴Se, 0.89%; ⁷⁶Se, 9.37%; ⁷⁷Se, 7.63%; ⁷⁸Se, 23.77%; ⁸⁰Se, 49.61%; ⁸²Se, 8.73%). Despite its high natural abundance, the isotope ⁸⁰Se is not amenable to detection via ICP-MS due to the presence of Ar₂, which shares the same *m/z* value in this analytical technique. The

natural abundances of ^{74}Se and ^{77}Se are too low to be reliably detected. Additionally, the detection of ^{76}Se is hindered by interference from polyatomic ions $^{40}\text{Ar}^{36}\text{Ar}$. Both ^{82}Se and ^{78}Se encounter interference from other factors during detection. However, ^{78}Se exhibits a relatively higher natural abundance and decent response in ICP-MS, and the interference from polyatomic ions can be effectively addressed through a high-resolution ICP-MS or the implementation of a collision gas, typically a mixture of 95% helium and 5% hydrogen), a technique termed the collision cell technique (CCT). Consequently, ^{78}Se is generally considered the most suitable isotope for detection and quantification via ICP-MS.

Review Comment 20:

“Several instances where more precise scientific terminology should be used:

(1) *“Perceived” is used throughout the manuscript including the title. This is an ambiguous word in this context. I would suggest changing to “detected” or “quantitated” throughout.*

(2) *“K562 cells were sorted and the nibbled selenoglycans [...]”. I would recommend “transferred” or “ingested”.*

(3) *“[...] and portrayed as an avant garde tool to visualize and quantify newly-synthesized sialoglycans in animals with elemental precision”. Avant-garde is a term usually reserved for art/literature, indicating that the work produced by an artist is innovative/experimental while usually being unacceptable to the establishment at the time. I don’t think the authors mean to imply SeMOE will be viewed as unacceptable by Nat. Com. readers; if the intend to convey that the technique is innovative, then please use the word “innovative” rather than avant garde. Likewise, precision is an analytical term referring to the agreement between replicate measurements, usually expressed as the standard deviation. Please revise this sentence.*

(4) *“[...] interdisciplinary thought collision is vivified”: this is both unclear and, if I may guess the intended meaning, seriously overstates the conclusions of this research. Please revise.*

(5) *“The core concept of SeMOE is minimalism”. In addition to access to a synthetic chemistry lab suitable for handling selenium-containing reagents, the entire SeMOE protocol uses instruments/tools/infrastructure that are not available in most university campuses. There is absolutely nothing minimalist about this research! Please revise.”*

Response: We acknowledge the reviewer for the meticulousness with science. We appreciate the professionalism our reviewer has put in improving the quality of the manuscript. We have fixed these phrases accordingly. We also realize that some claims in the previous version of the manuscript seemed inaccurate and overstated. We apologize for the misunderstandings that the original claim brings. Language in both the main text, and the Supporting Information has also been grammatically adjusted to eradicate the misunderstanding of our research. We also streamline the Methods section by adding more

detailed information for better understanding and experiment repeatability.

Taken together, we wish to express our gratitude again to the respected Referee 2 for providing truly insightful suggestions.

Referee:3

Review Comment 1:

“The submitted manuscript is an interesting exploration of the utility of monosaccharides tagged with selenoethers. The addition of selenium to the monosaccharides provides an elemental handle to allow convenient monitoring by elemental mass spectroscopy, and a unique isotopic signature for glycoproteomics applications. The manuscript combines many types of experiments showing the potential of these labelled monosaccharides in a variety of applications. Overall this is good work but many statements need to be clarified and many key technical details are missing.”

“The key innovation in the work is the ability to directly quantify the amount of selenium with atomic mass spectroscopy. The synthesis of the seleo-tagged sugars is well described and the identity and purity of the compounds by NMR is adequate.”

Response: We are grateful for the positive feedback and strong support from the respected reviewer. We acknowledge the reviewer for this concise yet insightful summary of the interesting points, key innovation and the amount of data in our work.

Review Comment 2:

“In the description of the toxicity of these compounds the authors state that because selenoglycosides are produced as a selenium detoxification mechanism, the seleo-tagged sugars should not be toxic. This is not necessarily true, as they are very different chemical structures. Furthermore, the authors state “comparable (toxicity) with native monosaccharide precursor or azido sialic acid at the range of 0-5 mM”. This is a gross simplification of the data in sup. Fig. 1 where the majority of the compounds have toxicity above 1mM and many are more toxic. This statement needs to be clarified.”

Response: We thank the reviewer for pointing out that toxicity may vary based on differences in chemical structures, and a direct correlation upon chemical structure similarity seems doubtful. We apologize for the misunderstanding in writing, and rephrased this sentence to clarify.

Review Comment 3:

“In Figure 2 the authors characterize the uptake of the seleno-tagged monosaccharides in cell culture. The methods describing these experiments do not align with the data presented. In the method for Figure 2a, a protein concentration determination method is not described, yet the signal is normalized to protein content on the y-axis. Figure 2d, the authors change cell lines from HeLa, used in 1a-c to 293T, this makes the CyTOF signal difficult to interpret in terms of the degree of labelling. It is necessary to have a bulk measure of the incorporation into 293T cells to provide an estimate of signal in CyTOF.”

Clearly the CyTOF signal is very low, nearly unusable, due to the low sensitivity of the method for selenium and the authors should comment on this in the text. The CyTOF data should be presented as a histogram to more clearly show the relative event count numbers.

Similar comment was raised in Reviewer 2 Comment 6.

Response: We thank the reviewer for your effort in improving the quality of our manuscript. We now add the protein concentration determination and normalization in the figure legend of Figure 2a, and streamline the Methods section by adding more detailed information for better understanding and experiment repeatability.

With regard to the CyTOF experiment (Figure 2d), the instrument setting was provided by one of the collaborators listed, who only had the 293T cell at hand by then. We agree with reviewer 2 that a histogram is a better way to present flow cytometric data. The original CyTOF results were not ideal (Se signal is essentially undistinguishable from the control). We attribute this result to the relative low incorporation rate of SeMOE precursors compared to other cell lines. Of course, the relative low signal from CyTOF may also be due to the low sensitivity of the method for selenium (compared to other Lanthanide elements). We now redo the CyTOF experiment in HeLa cells and MCF-7 cells, respectively, and display the data in the form of histograms. The Se signal is now essentially distinguishable from the control (**Fig. 2d, Supplementary Fig. 6**). We believe the new CyTOF experiment with HeLa and MCF-7 is now clear and in alignment with Figure 2a-2c for data presentation. We also add a note to specify the gating strategy and analyze procedures (**Supplementary Note 3**) both in the Figure 2d legend and in the Methods section.

Fig.2d (left) and Supplementary Fig. 6(right) CyTOF analysis of SeMOE probe-treated HeLa cells and MCF-7 cells. Cells were treated with vehicle, 2 mM ManNSe, 2 mM SiaNSe, 2 mM 9AzSiaNSe, 200 μ M 1,6-Pr₂ManNSe, and 100 μ M Ac₄ManNSe for 48 h, respectively, followed by CyTOF analysis. m/z at 78 was used for calculation of ⁷⁸Se. At least 30,000 events were gated and analyzed according to the CyTOF gating strategy (Supplementary Note 3).

Supplementary Note 3

Review Comment 4:

“Figure 2f is central to all of the work in this manuscript. In the majority of the subsequent experiments the authors rely on the data in 2f to support the central tenant of the manuscript that the seleno-tagged sugars deposit selenium only into glycans, and not into other biochemical pathways. This is not explicitly stated and should be. One could imagine that there is degradation, or simple metabolism that could lead to Se accumulation via other mechanisms not related to glycan biosynthesis. Only if all of the Se accumulation is glycan related is the ‘absolute quantitation’ claimed in the introduction supported. As can be seen in the neuraminidase treated cells a large fraction of the signal is from sialyated glycans, however not all the signal, and it is surprisingly dependent on the monosaccharide. The Ac4Man derivative gives the most neuraminidase resistant material, and this is consistent with non-specific labelling known with peracylated sugars, and the data in figure S6, so this should be stated. The sialic acid derivatives also give significant neuraminidase resistant material. How the data was obtained for the neuraminidase treated lysate is not complete in the methods section. The method section does not explain how the glycans were separated from the protein after enzymatic treatment, thus it is unclear if this measurement also reflects the loss of the small molecule tagged Se pool which may or may not be glycan related. This experiment needs to be clarified and elaborated on in the text. It is central and needs to be rigorously analyzed and discussed.”

Similar comment was raised in Reviewer 2 Comment 7.

Response: We are very grateful for these meticulousness and professional suggestions from the reviewer. We apologize for the confusing statement both in the main text and the Methods section in the previous manuscript. The cellular fate of unnatural sugars, in our humble opinions, not only in the context of selenium-containing precursors, but also in classical MOE probes such as azido sialic acid, is the key factor that governs the successful signal readout transfer (azide/selenium truly represents glycans). We agree

with the reviewer that in the context of “absolute quantification” this is especially significant, and it is the main reason why we conducted a series of experiments in Figure 3 to interpret it. We now explicitly state this sentence in the revised paper, per the reviewer’s suggestion. In addition, we add a paragraph in the Discussion section to transparently discuss the advantages and limitations of this study.

For the neuraminidase treatment experiment (**Fig. 2f**), we can only hypothesize that the remnant signal is actually from: (1) A side reaction between Ac₄ManNSe and cysteines on cell-surface proteins (protein S-glyco-modification, *J. Am. Chem. Soc.* **2020**, 142, 20, 9382.), as the reviewer mentioned in the comment; (2) Sialidase specificity toward cell-surface selenium-containing sialoglycans may be low. (3) Monosaccharide cross-talk (the selenium-containing sugar resides on cell-surface glycoconjugates, but not in the form of sialic acid). We believe a thorough investigation based on HPAEC-PAD or selenium speciation would give the answer. We performed selenium speciation on selenocysteine (SeCys₂), Se-methyl-L-selenocysteine (MeSeCys), selenomethionine (SeMet), selenate [SeO₄²⁻], and selenite [SeO₃²⁻], five compounds associated with selenoprotein conversion. Major organs administered with or without SeMOE probes displayed near near-identical selenium speciation, suggesting that injection of seleno-sialoglycans will not significantly affect the selenoprotein constituents in mice (**Supplementary Fig. 23**). In addition, we do not observe significant changes in major selenoprotein expression level *in vitro*, as evidenced by RT-qPCR analysis (**Supplementary Fig. 3**).

Supplementary Fig. 3 Selenoprotein mRNA levels of SeMOE-treated cells. HeLa cells were treated with PBS, 2 mM ManNSe, 2 mM SiaNSe, 2 mM 9AzSiaNSe, 200 μM 1,6-Pr₂ManNSe, 200 μM 1,6-Pr₂GalNSe, 100 μM Ac₄ManNSe or 100 μM Ac₄GalNSe for 48 h, respectively, followed by RT-qPCR analysis. n=4. Source data are provided as a Source Data file.

Supplementary Fig. 23 Selenoamino acid levels of SeMOE-treated mice. a-e, Male BALB/c mice (8 to 10 weeks old) were once-daily, intraperitoneally injected with Ac₄ManNSe (160 mg Ac₄ManNSe/kg/day), while control mice received the 70% DMSO alone. On day 5, mice were euthanized. The heart, liver, spleen, lung and kidney of mice were collected, digested by proteinase K and trypsin, followed by selenoamino acid analysis via ICP-MS. f, Comparison of the selenoamino acid peak integral between the Control group and SeMOE group. Error bars represent mean \pm SD of $n=3$ independent biological replicates. * $P < 0.05$, ** $P < 0.01$, *** $P < 0.001$, ns, not significant (two-tailed Student's t-test). Source data are provided as a Source Data file.

We also explicitly clarify and elaborate the Method section to provide more detailed information on the enzyme treatment experiment in the revised manuscript. In brief, after enzymatic treatment, cell proteins were precipitated by adding into a MeOH/CHCl₃ mixture (aqueous phase/MeOH/CHCl₃=4:4:1, v/v/v). The precipitated proteins were collected by centrifugation (20,000 g, 10 min, 4°C), and washed three times with cold methanol. Above enzyme-treated live cells and protein samples were digested in nitric acid and hydrogen peroxide (2:1, v/v) at room temperature overnight, and analyzed by ICP-MS.

Review Comment 5:

“Figure 2g is additional support for where the Se signal is localizing. It is difficult to know what to take from this data as the Ac₄Man derivative shows significant toxicity at the concentrations used.”

Response: We thank the reviewer for this candid reminder. Peracetylated ManNAc derivatives indeed show significant toxicity when their concentrations are greater than 200 μM, according to the CCK-8 results (**Supplementary Fig. 1**). To make the argumentation in a more convincing way, we have deleted the data of 200 μM Ac₄ManNAc in **Fig. 2g and Supplementary Fig. 10**. Both Ac₄ManNAc and Sia are convertible to sialic acid by the Roseman-Bertozzi biosynthesis pathway, and they competitively inhibit the incorporation of SeMOE probes, implying that SeMOE probes also participate the sialic acid biosynthesis pathway.

Review Comment 6:

“Overall from the data in figure 2 it appears that most of the Se accumulation is into cellular glycans, but not all. Any claims of ‘quantitative detection’ of MOE labelling need to be tempered to reflect these data.”

Response: We agree with the reviewer that cautions should be made to ensure the validation of an argument in science, and to avoid devalidation based on exaggeration. Due to the fact that not all of Se accumulated in the cell is in the form of glycans, we would specifically adjust the “quantitative detection” in a more “normalized” way. We thank the reviewer for the meticulousness and rigorousness of this system.

Review Comment 7:

“I read the methods and results section for panels 3b-d multiple times and did not understand the experiments. The description of the fractions needs to be revised.”

Similar comment was raised in Reviewer 4 Comment 11.

Response: We apologize for any confusion that has been caused during your reading

experience. We have revised the Methods section and the Results section with regard to these data. In brief, for analysis of total selenoglycans in subcellular fractions, we first isolated subcellular fractions in which Se exists in the form of covalently bound to proteins and free small molecules (**Fig. 2c**). Next, for analysis of selenoglycans covalently bound to proteins, proteins in the isolated subcellular fractions were precipitated, in which Se exists only in the form of covalently bound to proteins (**Fig. 2b**). In addition, we have replaced the pie graph with a column graph, as requested by another reviewer, to better display the standard deviation in Fig. 3d. The corresponding revise on Methods is also done for clarification.

Review Comment 8:

“In figure 4 the authors describe the transfer of material between cells whom are allowed to contact, or whom are separated by a membrane. The authors claim that the labeled material is sialoglycans, and this may be true, but it could also be any of the other intermediates, or potentially off pathway Se containing molecules. At the very least this limitation needs to be explicitly stated. Control experiments similar to those done in figure 2 would aid in the support of the authors hypothesis.”

Similar comment was raised in Reviewer 2 Comment 15.

Response: We greatly appreciate this insightful suggestion from the reviewer, and explicitly mention the limitations in the main text. For experiment, we additionally performed the sialidase treatment in Jurkat-K562 trogocytosis (Please refer to **Supplementary Fig. 16a**). Before co-incubation, Se-labelled Jurkat cells were treated with sialidase, and then co-incubated with K562 cells with sialidase. After co-incubation, K562 cells were sorted, followed by ICP-MS. The results showed a significant decrease in Se levels, indicating that Se is transferred to K562 cells in the sialoside form.

Supplementary Fig. 16a, Seleno-sialic acid transfer from SeMOE probe-labelled Jurkat cells to K562 cells during trogocytosis with or without sialidase treatment. Error bars

represent mean \pm SD of $n = 3$ independent biological replicates. * $P < 0.05$, ** $P < 0.01$, *** $P < 0.001$, ns, not significant (two-tailed Student's t-test). Source data including P values are provided as a Source Data file.

Review Comment 9:

“It is odd that the Se labelling with the different monosaccharides in GC’s is opposite to what is observed in the Hela cells in Figure 2”

Response: We thank the reviewer for raising this interesting open question. We indeed find that the Se labelling with different monosaccharides in GCs is inconsistent with usual cell lines (e. g. HeLa cell). We have therefore repeated the confocal imaging experiment (**Response Letter Figure 8**), and found two interesting phenomena: (1) the labelling effect of 9AzSiaNSe is better than that of 1,6-Pr₂ManNAz, though not directly comparable due to concentration variation; (2) the labelled sialic acids of GCs are mostly distributed inside the cells, instead of residing on the cell surface (Note: GCs were not permeabilized during the sample preparation and labeling procedures). We did not pry on the mechanism in depth, but we believe it has something to do with the nature of primary cell types rather than a cell line. A small discussion of these phenomena is included in the revised manuscript.

Response Letter Figure 8 Confocal fluorescence imaging of 9AzSiaNSe- or 1,6-Pr₂ManNAz-treated GCs. Mouse GCs were treated with vehicle (PBS), 9AzSiaNSe and 1,6-Pr₂ManNAz at indicated concentrations for 48 h, respectively. The GCs were reacted with alkyne-AZDye 647 via click chemistry. The nucleus was stained by DAPI. Scale bar: 20 μ m.

Review Comment 10:

"In figure 5 the authors do biodistribution studies on the seleno-tagged sugars.

(1) Interestingly the Se signal in the organs decreases when the dosing was increased from 160 to 320 mg/kg in both Se-sugars studied. This is unexpected and should be commented on in the text, is there a rationale for this observation?"

Response: We adapted the original dosage empirically from the Ac₄ManNAz experiment *in vivo* (*Nature*, **2004**, 430, 873-877), and added several other concentrations for comparison and possible toxicity evaluation on SeMOE probes *in vivo*. The results in Figure 5b are consistent with the safety dose range (**Supplementary Fig. 22**). 320 mg/kg Ac₄ManNSe-treated mice lost significant weight, while the mice treated with 80 mg/kg or 160 mg/kg Ac₄ManNSe maintained a relatively stable body weight. We have added a sentence to explicitly describe the observed phenomenon. In all, the Se labelling efficiency was closely related to dosage as well as the health status (possible toxicity) of the treated mice.

Supplementary Fig. 22 Body weight of SeMOE-treated BALB/c mice. Error bars represent mean \pm SD of n=3 independent biological replicates. Source data are provided as a Source Data file.

Review Comment 11:

"(2) The laser ablation studies do not include a scale bar."

Response: We thank the reviewer for this kind reminder. We have revised the scale bar to make it clearer.

Review Comment 12:

"(3) Does the observed morphology of the signal agree with previous studies using MOE with sialic acid?"

Response: This is a very good question. Such comparison would definitely help with the understanding of SeMOE and (or) MOE spatial biodistribution. In a previous study, liver, spleen, kidney, intestine, heart and lung of Ac₄ManNAz-treated mouse showed obvious

labelling signal (*PNAS*, **2016**, 113(19): 5173-5178), which is consistent with our results in this manuscript (**Fig. 5b, c**). However, it was failed to find labelling signal in the brain (**Response Letter Figure 9**). It is speculated that BBB prevent the entrance of peracetylated ManNAz into brain. So, we assume that the BBB blocks transit of the classic MOE monosaccharides such as Ac₄ManNAz but not the selenium analogs would be a better way to explain this phenomenon because the signal observed in Figure 5c is distinguishable, although additional mechanistic experiments are needed, which we believe is beyond the scope of this paper.

[redacted]

Response Letter Figure 9 *Ex vivo* imaging of major murine organs. Excerpted from *PNAS*, **2016**, 113(19): 5173-5178. <https://doi.org/10.1073/pnas.1516524113>

Review Comment 13:

“Figure 6 explores the effects of the tagged sugars on ROS in MCF7 vs MCF10A. A dramatic difference was observed but the reason for this difference was not investigated.”

Response: We thank the reviewer for this insightful comment. We speculate that the effect of SeMOE probes on cellular ROS generation is related to the redox properties of the Se element. Selenium is potentially a redox modulator, and selenium-containing compounds may have antioxidant and prooxidative properties, depending on the compound structure, experimental condition and their concentrations. Generally speaking, selenium-containing compounds have anticancer potentiality when they act as prooxidants. Many studies have used selenium-containing compounds or selenium-containing nano-drugs to target tumor cells and increase the ROS level, thereby assisting in the killing of tumor cells (*Dalton Transactions*, **2016**, 45(46): 18465.; *Chemical Reviews*, **2019**, 119(8): 4881.; *Nanoscale*, **2020**, 12(3): 1389.). However, in view that the results of SeMOE biological impact are a bit

too preliminary and lack mechanistic elucidation, we decide to change the original manuscript topic to “Selenium-based metabolic oligosaccharide engineering strategy for quantitative glycan detection” to clarify the methodological development nature of this work. Based on the comments from the respected reviewers, we integrated and moved all figures regarding the potential anti-cancer application to **Supplementary Figures 25-28**.

Review Comment 14:

“Overall this manuscript nicely shows the potential of using heavy element tagging of metabolic precursors for following MOE. The range of studies presented clearly demonstrate the value of direct quantification using atomic mass spectrometry. References to some of the other metabolites that have been tagged with heavy elements and used for CyTOF experiments is warranted. Despite these benefits that authors need to clarify that other Se metabolites may be playing a role in these measurements and all of the y-axis measuring Se content should be modified from seleno-sialic acid to Se content.”

Response: We are grateful for the extremely supportive and inspiring comments from the respected reviewer. References about other metabolites in the CyTOF experiment are added in the revised paper. Also, although speciation analysis for selenoproteins and RT-qPCR experiment validated the minimal perturbation of SeMOE probes on native selenium-containing molecules, we explicitly clarified that other Se metabolites may be playing a role in these measurements in the Discussion section. In addition, we have revised the y-axis for a clearer demonstration of signal output.

Taken together, we wish to express our gratitude again to the respected Referee 3 for providing truly insightful suggestions.

Referee:4

Review Comment 1:

“The manuscript “SeMOE allows for quantitative glycan perception and exhibits anti-cancer potentiality” is an interesting, well written, and (generally) meticulously performed study that provides a way to directly detect incorporation of a sugar analog used in metabolic oligosaccharide engineering (MOE) into cellular glycans. The main advance in this study is the direct detection of the non-natural glycan through detection of the inherent selenium atom(s); this avoids cumbersome second-step labeling methods required with “classic” MOE approaches. As an important bonus, the method is efficient, with metabolic incorporation (most likely) of the selenosugars more facile than the widely-used azido-modified analogs that are currently workhorse reagents in the field. The study goes on to show biological applications of the selenosugars, mostly the transfer from one cell type to another as occurs in trogocytosis (note that these “applications” basically verify/illustrate already-known biological knowledge but they do demonstrate that the method has future potential as a biological probe). Overall I believe that this study could be worked into a high impact publication with a few edits and improvements as described below.”

Response: We are grateful for the positive feedback and strong support from the respected reviewer. We acknowledge the reviewer for this concise yet insightful summary of the interesting points and key innovations in our work.

Review Comment 2:

“The N-glycoproteomics data shown in Figure 3 suggests that there is biased incorporation of the seleno-sialic acids into certain subsets of sialoglycans (e.g., to use a hypothetical example of three heavily sialylated proteins – let’s say each with 20 sialic acids – one might have 0 or 1 natural sialic acids replaced with seleno-sialic acid, the second might have 3 or 4, and the third might have 10 to 12). Note that this a totally expected result, that seems to be reflected in this data. The reason I mention this is that the authors should avoid use of language that could be interpreted that each of these three hypothetical sialylated proteins have the same degree of incorporation, oh let’s say 5 seleno-sialic acids each. For example, the statement “SeMOE developed in this work provides a simplified, yet versatile and convenient protocol for absolute quantification during glycan-mediated processes based on the feature of the non-metal element Se” could be interpreted that absolute quantification of the endogenous glycans involved in these processes is possible with this technique, In reality, what can be measured is the absolute quantification on non-natural analog incorporation, which can (indeed, most likely) is biased. To be more rigorous about providing (or disproving) this premise, a comparison of the seleno-sialic acid signals with endogenous sialic acid levels (e.g., perhaps through lectin staining) would be interesting.”

Similar comment was raised in Reviewer 2 Comment 5.

Response: We thank the reviewer for this interesting and lucid example. We agree with the reviewer that newly-synthesized, selenium-containing glycans (also other selenium component transformed from the SeMOE, to some extent) is the real part that exhibit absolute quantification ability. We therefore rephrased the sentence in the paragraph to more specifically express this.

As suggested by the reviewer, we performed a lectin flow cytometry experiment, and found no significant fluctuation of α 2,6-linked (SNA lectin) and α 2,3-linked (with preference to Sia α 2-3Gal β 1-3(\pm Sia α 2-6)GalNAc, MAL-II lectin) sialic acid on the cell surface (**Supplementary Fig. 2**). These results collectively prove that SeMOE probes preserved as a metabolic precursor, and do not perturb the formation of sialosides.

Supplementary Fig. 2 Flow cytometry analysis of sialic acid levels of SeMOE probe-treated cells. α 2,6-linked and α 2,3-linked sialic acids were recognized by *Sambucus nigra* (SNA) and *Maackia amurensis* leucoagglutinin II (MALII), respectively. a-b, Flow cytometry analysis of α 2,6-linked sialic acids (a) and α 2,3-linked sialic acids (b) of SeMOE probe-treated HeLa cells. c-d, Flow cytometry analysis of α 2,6-linked sialic acids (c) and α 2,3-

linked sialic acids (d) of SeMOE probe- treated MCF-7 cells. Error bars represent mean \pm SD of n= 3 independent biological replicates. * $P < 0.05$, ** $P < 0.01$, *** $P < 0.001$, ns, not significant (one-way ANOVA followed by post hoc Dunnett's test). Source data including exact P values are provided as a Source Data file.

Review Comment 3:

*“A “selling point” for the use of seleno-sugars is the low background of selenium in cells and tissues; nevertheless (as the authors acknowledge by mentioning the existence of seleno-proteins) there is *some* level of background signal in human cells and tissues. It would be helpful if the author attempt to quantify the background. for example, is the background signal in Figure 2c just “noise” or is it legitimately from endogenous selenoproteins?”*

Just going from memory, I recall that a person has $\sim 10^{18}$ selenoproteins in their body, which (if distributed equally, which is almost certainly not the case) works out to $\sim 100,000$ copies per cell. By comparison, going from data presented on Page 12 (1.6-4.5 femtomoles selenosugar per Jurkat cell) that works out according to my quick calculations to be $\sim 10^9$ copies of selenosugar per cell, or about four orders of magnitude over the “background/noise” from endogenous signal. If my quick calculations are correct the authors might want to more explicitly/quantitatively outline this rather outstanding signal to noise/background ratio.

Also, “femtomoles sugar per cell” is, in my opinion, not that understandable by the average reader insofar as it comes across as a rather small number; again, as a “selling” point the authors might want to briefly (perhaps in the Discussion) provide context that this level is actually \sim one billion copies per cell, of which 0.1-3% is transferred during trogocytosis (and possible compare this to levels of conventional CD markers, which occur at much lower levels)”

Response: We agree with the reviewer that native selenium-containing molecules, including selenoproteins, would bring in “some” level of background signal in human cells and tissues. Because Fig. 2c is actually acquired in in vitro, we can only speculate that these signals should be originated from the “noise” because the DMEM medium does have an ingredient without the addition of selenium. Measurement of standard recovery rate (adding known amount of selenium in the medium and seeing whether an increase in Se signal can be observed) should be a good design in solving the question the respected reviewer has asked. We believe this quantitative signal-to-noise contrast matters more *in vivo* since selenoproteins, though not abundant, do exist in living animals with biological functions. We agree with the reviewer that a quantitative calculation would be worth a thousand words in an “absolute quantification”. As suggested by the respected reviewer, we did a rigorous literature research and calculation with regard to the selenoprotein and CD antigen density (**Supplementary Note 4**). We want to again express our gratitude for the nice suggestion the reviewer has raised. We now revised the statement of “femtomoles

sugar per cell” to express the relative numbers, and hopefully it “sells” in a really good way (at least in a more perceptual way).

Review Comment 4:

“Experimentally, the data in Figure 6 concerning ROS production in meaningless w/out appropriate controls. For example, does atomic (in salt form?) selenium have the same effects at the equivalent doses? Contrary-wise, do MOE analogs w/out selenium (e.g., Ac₄ManNAc, Ac₄ManNAz, or perhaps thiol-modified ManNAc analogs) have the same impact by perturbing sialic acid biosynthesis.”

Similar comment was raised in Reviewer 1 Comment 8 and Reviewer 2 Comment 14.

Response: We thank the reviewer for noticing this flaw. We systematically evaluated the labeling efficiency of all SeMOE probes, including the per-acetylated compounds, in various cell lines (**Supplementary Figure 25**), and noticed that free monosaccharides generally trigger higher ROS responses. We admit that the data between free monosaccharide and its protected counterpart is not directly comparable in the original data presentation. To optimize and figure it out, we treated MCF-7 cells with 10 μ M of vehicle, Na₂SeO₃, and corresponding monosaccharides, to measure the ROS assay at the same level (**Supplementary Figure 26e**). The results indicated that an increase in ROS production by the free monosaccharide is an intrinsic property, rather than discrepancies in concentration increases. With these controls, we believe the experimental design should be more rigorous.

Supplementary Fig. 26e ROS assay of MCF-7 cells treated with 10 μ M respective monosaccharides for 1 h. Error bars represent mean \pm SD of n=3 independent biological replicates. * P < 0.05, ** P < 0.01, *** P < 0.001, ns, not significant (one-way ANOVA followed by post hoc Dunnett’s test). Source data are provided as a Source Data file.

Review Comment 5:

“Minor points 1. Page 1, Abstract: typo in heading (Abatract)”

Response: The typo is now fixed.

Review Comment 6:

“2. Page 2, first line: “the most appealing” should be less hyperbolic (i.e., “an appealing” would be better”

Response: Always happy to know English grammar better. The sentence is fixed as suggested.

Review Comment 7:

“3. Page 2, lines 4/5: the statement “Such compounds are converted into glycoenzyme substrates within cells” is often but not always correct; for example, the first generation ManNAc analogs developed by Reutter’s group were used direct in MOE w/out any metabolic conversion (e.g., removal of O-hydrosyl acetyl protecting groups)”

Response: Also, always happy to learn more. The sentence is fixed accordingly.

Review Comment 8:

“4. Page 4, Figure 1a: the meaning “of limitation” (in “attomolar detection of limitation”) is unclear (would just “attomolar detection” be sufficient?)”

Response: We thank the reviewer for pointing this out. We have revised the phrase to “attomolar detection”.

Review Comment 9:

“5. Page 4, last paragraph: considering that two studies (plural) are referenced, should “A previous work” also be plural (e.g., something like “Previous studies”)?”

Response: We have fixed the sentence as the respected reviewer suggested.

Review Comment 10:

“6. Page 5, second line: “lyzed” is more commonly “lysed”.”

Response: We thank the reviewer for pointing this out and have fixed the spelling.

Review Comment 11:

“7. Page 5, first paragraph: It’d be helpful to state here how proteins were extracted (e.g., whole cell fractions, membrane fractions, cytosolic fractions)”

Similar comment was raised in Reviewer 3 Comment 7.

Response: We have revised the description in the corresponding sections to clarify.

Review Comment 12:

“8. Page 5, first paragraph: compared to Figure 1a where attomolar sensitivity is mentioned, why is nanomolar sensitivity mentioned here? (can the disparity be explained? [e.g., perhaps the authors mean “attomole” rather than “attomolar”])”

Response: We apologize for the typo and revised the sentence.

Review Comment 13:

“9. Page 11, Figure 3d: should “newly-synthezied” be “newly-synthesized?””

Response: The typo is now fixed.

Review Comment 14:

“10. Page 12, near the bottom: “scavenged” might be a more appropriate word compared to “nibbled”.”

Response: We have changed the word as suggested.

Review Comment 15:

“10. Page 16, approximate middle of the page: Please clarify what “are challenging to analyze using classical MOE strategy when blood-brain barrier (BBB) arises” means. Does this mean that the BBB blocks transit of the classic MOE monosaccharides such as Ac₄ManNAz but not the selenium analogs? Or is the pitfall in the classic strategies the need for the imaging/labeling agents to cross the BBB? (or some combination of these two factors?)”

Response: An earlier work by the Carolyn Bertozzi lab (*Nature*, **2004**, 430(7002): 873-877) used Ac₄ManNAz metabolic labelling and Western blotting to characterize the sialylated glycoproteins in mouse brain, but failed to find them in brain (**Response Letter Figure 10**). It is speculated that BBB prevent the entrance of peracetylated ManNAz into brain. So, we assume that the BBB blocks transit of the classic MOE monosaccharides such as Ac₄ManNAz but not the selenium analogs would be a better way to explain this

phenomenon because the signal observed in Figure 5c is distinguishable, although additional mechanical experiments are needed, which we believe is beyond the scope of this paper.

[redacted]

Response Letter Figure 10 Western blot analysis of tissue lysates from Es1e /Es1e mice administered Ac₄ManNAz (+) or vehicle alone (-). Excerpted from *Nature*, **2004**, 430(7002): 873-877. <https://doi.org/10.1038/nature02791>

Review Comment 16:

“11. Page 20, in the middle: the phrase “The interdisciplinary thought collision is vivified” is certainly a catchy phrase! On the other hand, the meaning of this flowery language is not completely clear to me (do the authors mean something like “The interdisciplinary nature of MOE is vivified...” or maybe “The interdisciplinary nature of classic MOE is further extended by...”?)”

Response: We now change the sentence into a simpler and more scientific one.

Review Comment 17:

“12. Page 22, end of references: In some type of weird formatting glitch, References 48-50 are separate from the rest of the references (appearing on Page 38) I suspect that all of the references should be presented with those on Page 38, i.e., they should appear after the methods, section).”

Response: We thank the reviewer for the kind reminder. We have updated the reference list in a formal way.

Review Comment 18:

“13. Page 25: The synthesis of Ac₄GalNSe is not described and/or referenced”

Response: The synthesis of **8b** (Ac₄GalNSe) was obtained from **7b** (GalNSe) with the same procedure for synthesis of **8a** (Ac₄ManNSe) as a white solid. Please refer to

Supplementary information file for details (Supplementary Note 1, pages 46-47).

Review Comment 19:

“14. Page 27: please provide the g force, not rpm, for the 12,000 rpm centrifugation step(s) the rpm is meaningless w/out knowing what centrifuge rotor was used (check for 1,500 and 2,000 rpm on Page 29 as well, and throughout)”

Response: We thank the reviewer for pointing this out and have revised all units of centrifugal speed to g force.

Taken together, we wish to express our gratitude again to the respected Referee 4 for providing truly insightful suggestions.

In all, we wish to express our gratitude again to all the respected reviewers for the precious time, invaluable expertise, and superb professionalism you have put in improving the quality of our paper. We hope the manuscript is now suitable for the publication in ***Nature Communications***.

REVIEWER COMMENTS

Reviewer #1 (Remarks to the Author):

The authors have addressed most comments. Some issues still need further attention:

- GalNAc salvage pathway: In contrast to sialic acid analogs, GalNAc (and GlcNAc) analogs are far more restricted in their conversion into UDP-sugars. This has been shown repeatedly (DOIs 10.1002/cbic.201700020, 10.1021/acscchembio.1c00034), and from this literature evidence, it seems unlikely that GalNSE would enter the GalNAc salvage pathway. If the authors wish to keep their results e.g. in Supporting Fig. 19 in the manuscript, in my opinion they need to convincingly show that the signal originates from glycoproteins (measurement of UDP-GalNSE biosynthesis is the strongest indicator) and not, for instance, free GalNSE.
- Comparison of labeling techniques: This is in response to my earlier comment that the authors state a difference in detection of 5 orders of magnitude. I suggest the authors to re-phrase this since the difference in detection limit is caused by instrumentation (ICP-MS is more sensitive) rather than the sugars themselves. In turn, imaging has other benefits that, in many cases, outweigh the benefits of ICP-MS.

Reviewer #2 (Remarks to the Author):

I believe that most of my earlier comments/concerns have been adequately addressed. I also appreciate the extra experiment completed by the authors as well as the removal of unsupported (or exaggerated) conclusions from the text. I believe that this manuscript can be published but would request one minor addition to the supplementary notes. Specifically, the CyTOF gating is somewhat confusing to me. Specifically, what is the point of the Eu-by-Eu (selecting 97% of all events) and Ir-by-Ir gates? Why not just gate on a population from the Pt-by-Ir plot?

Reviewer #3 (Remarks to the Author):

In the revision of this manuscript the authors have addressed some of the concerns raised by the reviewers. However, there remains claims about the 'quantitative' measurement of glycans that are unfounded. The data supports 60-80% of the selenium being present in the modified sialic acid analogues. 60-80% is not quantitative, it is a good level of labelling but in any quantification the remaining 20-40% of the signal is of unknown origin. This is also true of all of MOE type experiments, which are also not quantitative.

The authors have included a selenium speciation experiment in this revision. It is unclear why this is added. Is there evidence that dosing with organoselenium species significantly perturbs this equilibrium?

Another reviewer has brought up the relevant point that SeMOE is not as good as radiolabelled derivative. I generally agree, but the authors should consider the other potential application of SeMOE analogs if they were synthesized as Se-isotopically enriched derivatives. These could be distinguished by ICP-MS and used in temporal studies. I suggest the authors read the recent review 'Reagents for Mass Cytometry' in Chemical Reviews with focus on what has been done in hypoxia measurements to see what is possible with isotopologous probes.

Overall, while improved, the manuscript needs to adjust any claims of quantitative glycan measurement.

Reviewer #4 (Remarks to the Author):

The authors have adequately met the concerns I raised in the initial review process

We would like to acknowledge all 4 reviewers for your precious time and efforts working on this manuscript. We have carefully considered each comment, and addressed all points to improve the overall quality of the manuscript. Below are our ***point-to-point responses*** to the comments and concerns raised by the reviewers. The cited, newly added or revised figures mentioned in the responses are shown correspondingly for the ease of your references. The manuscript file is now provided with both an annotated version with changes ***marked in red***, and a formatted manuscript.

Referee:1

Review Comment 1:

*“The authors have addressed most comments. Some issues still need further attention:
- GalNAc salvage pathway: In contrast to sialic acid analogs, GalNAc (and GlcNAc) analogs are far more restricted in their conversion into UDP-sugars. This has been shown repeatedly (DOIs 10.1002/cbic.201700020, 10.1021/acscchembio.1c00034), and from this literature evidence, it seems unlikely that GalNSe would enter the GalNAc salvage pathway. If the authors wish to keep their results e.g. in Supporting Fig. 19 in the manuscript, in my opinion they need to convincingly show that the signal originates from glycoproteins (measurement of UDP-GalNSe biosynthesis is the strongest indicator) and not, for instance, free GalNSe.”*

Response: We appreciate the reviewer’s positive support and expert feedback regarding the GalNAc salvage pathway. We first included the concentration-dependent detection of selenium levels in 1,6-Pr₂GalNSe-treated cells (**Supplementary Fig. 5a**), to demonstrate that 1,6-Pr₂GalNSe do serve as a SeMOE probe to be incorporated into cellular glycoproteins. Secondly, we treated cells with either 200 μM 1,6-Pr₂GalNSe or vehicle (PBS), conducted a β-elimination reaction on total proteins for O-GalNAc cleavage, and then measured the selenium contents in these proteins (**Supplementary Fig. 12a**). As expected, a signal decrease for selenium signals is observed, which should prove that these selenium signals originate from O-GalNAc glycoproteins. We speculate that the residual selenium signal after β-elimination treatment is originated from the possible UDP-GalNSe conversion toward UDP-GlcNSe, via the GNE:UDP-GlcNAc-2-epimerase/ManNAc kinase, which we have already mentioned in the manuscript (Discussion section, paragraph 3). Besides, we have measured the subcellular distribution of selenium signals by partitioning the proteins in 1,6-Pr₂GalNSe-treated cells from nucleus, cytosol and membrane (**Supplementary Fig. 14**). The observation that 57% of signals exhibited on total membrane partially implicated/suggested that the incorporation of 1,6-Pr₂GalNSe precursor into the O-GalNAc glycoprotein on cell membrane, while the remaining signals from cytosol and nucleus (~43% in total) are glycosylated, Se-containing proteins, presumably in the form of O-GlcNSe, corresponding to the O-GlcNAcylation modification. This also vaguely indicated that the interconversion between UDP-GalNSe and UDP-GlcNSe exists, in accordance with the distribution percentages in **Supplementary Fig. 12a**. In addition, we performed a UPLC-TSQ-MS/MS to validate that the existence of UDP-GalNSe, as kindly suggested by the respected reviewer. Luckily, we found a concentration-dependent UDP-GalNSe level increase when 1,6-Pr₂GalNSe treatment elevated (**Supplementary Fig. 12b**). We believe that these robust data would prove that 1) why we finally choose the sialylated SeMOE probes in the following experiment because the biosynthetic pathway is more transparent, and 2) we have convincingly shown that the Se signal do originate from glycoproteins, and UDP-GalNSe do serve as the metabolic precursor in the GalNAc salvage pathway for selenium-containing unnatural monosaccharides. Collectively, we have added several sentences with regard to the above-mentioned discussion in the manuscript for clarification.

Supplementary Fig. 5a Se levels of glycoproteins from SeMOE-treated HeLa cells. a, Se levels of glycoproteins from HeLa cells treated with GalNSe, 1,6-Pr₂GalNSe or Ac₄GalNSe at varied concentrations for 48 h. Error bars represent mean \pm SD of n=3 independent biological replicates. Source data are provided as a Source Data file.

Supplementary Fig. 12 SeMOE of 1,6-Pr₂GalNSe via GalNAc salvage pathway. a, MCF-7 cells were treated with vehicle (PBS) or 200 µM 1,6-Pr₂GalNSe for 48 h, lysed for glycoprotein extraction, followed by β-elimination treatment and ICP-MS analysis. b, MCF-7 cells were treated with vehicle (PBS) or 1, 6-Pr₂GalNSe for 48 h. UPLC-TSQ-MS/MS was performed to determine the cellular level of UDP-GalNSe. NF, Not Found. Error bars represent mean ± SD of n= 3 independent biological replicates. **P* < 0.05, ***P* < 0.01, ****P* < 0.001, ns, not significant (two-tailed Student's t-test). Source data including exact *P* values are provided as a Source Data file.

Supplementary Fig. 14 Se level of different subcellular proteins in MCF-7 cells. MCF-7 cells were treated with vehicle (PBS) or 200 µM 1,6-Pr₂GalNSe for 48 h. Proteins in different subcellular regions were isolated and precipitated, followed by ICP-MS analysis. Error bars represent mean ± SD of n= 3 independent biological replicates. Source data are provided as a Source Data file.

Review Comment 2:

“Comparison of labeling techniques: This is in response to my earlier comment that the authors state a difference in detection of 5 orders of magnitude. I suggest the authors to re-phrase this since the difference in detection limit is caused by instrumentation (ICP-MS is more sensitive) rather than the sugars themselves. In turn, imaging has other benefits that, in many cases, outweigh the benefits of ICP-MS.”

Response: We thank the reviewer for the rigorous suggestion. We agree with the reviewer that a rough comparison with different instrumentation is vague and inaccurate. We have rephrased and deleted the comparison in this sentence.

We wish to express our gratitude again to the respected reviewer for your precious time, invaluable expertise, and superb professionalism you have put in improving the quality of our paper.

Referee:2

Review Comment 1:

"I believe that most of my earlier comments/concerns have been adequately addressed. I also appreciate the extra experiment completed by the authors as well as the removal of unsupported (or exaggerated) conclusions from the text. I believe that this manuscript can be published but would request one minor addition to the supplementary notes."

Response: We acknowledge the respected reviewer for the support of this manuscript.

Review Comment 2:

"Specifically, the CyTOF gating is somewhat confusing to me. Specifically, what is the point of the Eu-by-Eu (selecting 97% of all events) and Ir-by-Ir gates? Why not just gate on a population from the Pt-by-Ir plot?"

Response: We apologize for any confusion caused by the statement in **Supplementary Note 3**. To address this issue, we have included more detailed descriptions in the revised **Supplementary Information** file with regard to the gating strategy.

The CyTOF gating strategy in this paper is based on pertinent literatures (<https://doi.org/10.1002/cyto.a.23034>; https://doi.org/10.1007/978-1-4939-7346-0_3; <https://doi.org/10.1038/nprot.2017.155>). And the types and working principles of elemental mass tags are shown in the **Response Letter Figure 1**. Specifically, prior to the CyTOF instrument, the samples were mixed with element calibration beads containing Eu signals to standardize/calibrate the mass cytometry data. When gating cells, we initially used $^{151}\text{Eu}/^{153}\text{Eu}$ to eliminate the Eu-containing beads and correctly select the cell population (96.7%). Subsequently, $^{191}\text{Ir}/^{193}\text{Ir}$ was employed to identify DNA-positive single cells. This was achieved by using Cell-ID™ Intercalator-Ir, a cationic nucleic acid intercalator containing naturally-abundant iridium (^{191}Ir and ^{193}Ir). This intercalator can be embedded in DNA, and serve as an indicator for cells containing DNA. Following DNA-positive selection, cisplatin (^{195}Pt) negative events were gated as live cells. Cisplatin is a staining reagent that covalently binds to cellular proteins, preferentially with compromised cell membranes (*i.e.*, dead cells). Consequently, gating solely based on the Pt-by-Ir plot may yield less accurate results, and we used the above-mentioned protocols instead.

Supplementary Note 3 CyTOF gating strategy.

[redacted]

Response Letter Figure 1 Elemental mass tags are typically classified as protein-detection mass tags or cell-identification mass tags according to their utilities in CyTOF analysis. Excerpted from *Nature protocols*, **2018**, 13(10): 2121-2148. <https://doi.org/10.1038/s41596-018-0016-7>

We wish to express our gratitude again to the respected reviewer for your precious time, invaluable expertise, and superb professionalism you have put in improving the quality of our paper.

Referee:3

Review Comment 1:

“In the revision of this manuscript the authors have addressed some of the concerns raised by the reviewers. However, there remains claims about the ‘quantitative’ measurement of glycans that are unfounded. The data supports 60-80% of the selenium being present in the modified sialic acid analogues. 60-80% is not quantitative, it is a good level of labelling but in any quantification the remaining 20-40% of the signal is of unknown origin. This is also true of all of MOE type experiments, which are also not quantitative.”

Response: We appreciate the reviewer’s kind reminder, and would want to clarify several points. Firstly, we agree with the respected reviewer that a percentage of selenium presented in a living system would not be rigorously considered as “absolute quantitative” or “quantitative”. The reviewer also correctly pointed out that all of the MOE type experiment are not “quantitative”, due to the uncertainty of the cellular fate of metabolic precursors. However, in the pertinent literatures using MOE, a not “quantitative” strategy with regard to glycan measurement (*ChemBioChem*. **2021**, 22, 1243–1251. <https://doi.org/10.1002/cbic.202000715>; *Anal Chem*. **2017**, 89, 538–543. <https://doi.org/10.1021/acs.analchem.6b04141>; *iScience*. **2021**, 24, 102397. <https://doi.org/10.1016/j.isci.2021.102397>; *Anal Chem*. **2015**, 87, 11460–11467. <https://doi.org/10.1021/acs.analchem.5b03135>; *Anal Chem*. **2022**, 94, 13745–13752. <https://doi.org/10.1021/acs.analchem.2c01961>, *Chem Sci*. **2022**, 13, 9701-9705. <https://doi.org/10.1039/D2SC03881A>), the authors unanimously used the term “quantification/quantitative” either in their article title or throughout the main text. We do not mean that the respected reviewer is totally wrong in his/her critical comment on our work, but would argue that different people may have different views on to what extent a definition (quantitative herein) would convey. The sialylation level revealed by the approach developed herein just accounts for the “partial” sialylation (*60-80% of the selenium being present in the modified sialic acid analogues*) that are newly-synthesized/metabolically incorporated via cellular biosynthetic pathway. So far, the real “absolute quantification” can only be determined via the cell-lysed standard monosaccharide derivatization with 1,2-diamino-4,5-methylenedioxybenzene (DMB) coupled with LC-fluorescence detection or LC-MS analysis, but this method also suffer from other limitations.

A rough (and probably not **really** accurate) estimation for the “quantity” of sialoglycans, in compatibility with the SeMOE strategy herein, is to “normalize” the correlation between actual incorporated sialoglycans and Se signal. In simple words, if 60-80% of the selenium being present in the modified sialic acid analogues is a known fact for this strategy, the easiest way to quantitate is to multiply the detected Se signal with 0.6 or 0.8 as an adjustment factor (adjustment constant). However, this requires tedious and onerous detections for the correct calculation of this “adjustment constant”.

By definition, the word “quantitative” means “relating to, measuring, or measured by

the quantity (the amount or number of something measurable) of something rather than its quality.” If we understand correctly, we believe that the respected reviewer would point out that the usage of “absolute quantification” throughout the manuscript is overexaggerating, which after reminiscence, we would admit the word “absolute” is truly inappropriate under this scenario. Therefore, we hope to find a “balancing point” that not only highlights the “quantitative” (rather than “absolute quantitative”) characteristics of this method, but also avoids exaggeration and confuses the readers. We have rephrased most of the sentences with regard to the absolute quantitative description and would hope this revised version would be more suitable for the respected reviewers and broader readerships.

Review Comment 2:

“The authors have included a selenium speciation experiment in this revision. It is unclear why this is added. Is there evidence that dosing with organoselenium species significantly perturbs this equilibrium?”

Response: We apologize for the unclear description of speciation experiment and have revised the relevant sentence in the manuscript to clarify. Selenium-containing compounds are typically metabolized into selenoamino acids *in vivo* (*J. Am. Chem. Soc.* **2023**, 145, 22, 12193–12205. <https://doi.org/10.1021/jacs.3c02179>). We want to figure out whether administration of the SeMOE probe would perturb the selenoamino acid levels (*i.e.*, transformed into other known selenium-containing biological entities rather than in the form of selenium-containing sugar/glycan). Therefore, a selenoamino acid speciation experiment was conducted. The results of this experiment demonstrated that major organs of mice, whether administered with or without the SeMOE probe, exhibited similar levels of selenoamino acids. This suggests that the administration of the SeMOE probe does not significantly perturb the selenoamino acid levels, and that the Se signal is predominantly derived from seleno-glycan rather than selenoamino acids (“background selenium level” if a sialoglycan quantification experiment is needed). The only exception was observed in the liver, with a signal increase for SeCys₂, MeSeCys and SeO₃²⁻, probably because liver is mostly involved in the digestive product metabolism (Please refer to **Supplementary Fig. 25**).

Supplementary Fig. 25 Selenoamino acid levels of SeMOE-treated mice. a-e, Male BALB/c mice (8 to 10 weeks old) were once-daily, intraperitoneally injected with Ac₄ManNSe (160 mg Ac₄ManNSe/kg/day), while control mice received the 70% DMSO alone. On day 5, mice were euthanized. The heart, liver, spleen, lung and kidney of mice were collected, digested by proteinase K and trypsin, followed by selenoamino acid analysis via ICP-MS. f, Comparison of the selenoamino acid peak integral between the Control group and SeMOE group. Error bars represent mean \pm SD of n=3 independent biological replicates. **P* < 0.05, ***P* < 0.01, ****P* < 0.001, ns, not significant (two-tailed Student's t-test). Source data are provided as a Source Data file.

Review Comment 3:

“Another reviewer has brought up the relevant point that SeMOE is not as good as radiolabelled derivataive. I generally agree, but the authors should consider the other potential application of SeMOE analogs if they were synthesized as Se-isotopically enriched derivatives. These could be distinguished by ICP-MS and used in temporal studies. I suggest the authors read the recent review ‘Reagents for Mass Cytometry’ in Chemical Reviews with focus on what has been done in hypoxia measurements to see what is possible with isotopologous probes.”

Response: We appreciate the insightful suggestions provided by the reviewer regarding Se-isotopically enriched derivatives. We have carefully reviewed the recent review titled 'Reagents for Mass Cytometry' in **Chemical Reviews**, with a specific focus on hypoxia measurements, and have included it in our reference list.

We admit that the current version of SeMOE probes has limitations in conducting temporal studies, and believe that Se-isotopically enriched derivatives have the potential to overcome this limitation. In fact, the initial design for Se application is exactly as the respected reviewer designed (Great minds think alike?). However, due to the high cost of Selenium standard and complex synthetic endeavors (For instance, 50 mg of ^{80}Se and ^{77}Se standards, the starting amount for the conversion from Se to SeMOE probes, prices €226.5-527.50, <https://en.institut-seltene-erden.de/unsere-service-2/metall-preise/preise-fuer-stabile-isotope/>), this application is economically not feasible. We therefore navigated our experiment toward the selenium-encoded isotopic signature targeted profiling (SESTAR algorithm) for glycoproteomic analysis.

Element	isotope	Dry designation	Clarity	Price in mg EURO
Selenium	Se 74	Se	77,7100	700,25
	Se 76	Se	97,0500	27,70
	Se 77	Se	94,3800	31,32
	Se 78	Se	98,8000	10,55
	Se 80	Se	99,4500	4,53
	Se 82	Se	97,1900	33,98

Furthermore, our laboratory is actively working on the development of the next generation of SeMOE probes, which will be based on Se-click reactions. We are confident that these advancements will address the limitations of the current probe, and enhance its performance in time temporal studies.

Review Comment 4:

“Overall, while improved, the manuscript needs to adjust any claims of quantitative glycan measurement.”

Response: As discussed in **Review Comment 1**, we have rephrased most of the sentences with regard to the absolute quantitative description and would hope this revised version would be more suitable for the respected reviewers and broader readerships.

We wish to express our gratitude again to the respected reviewer for your precious time, invaluable expertise, and superb professionalism you have put in improving the quality of our paper.

Referee:4

“The authors have adequately met the concerns I raised in the initial review process.”

Response: We appreciate the reviewer for the support of this manuscript.

In all, we wish to express our gratitude again to all the respected reviewers for the precious time, invaluable expertise, and superb professionalism you have put in improving the quality of our paper. We hope the manuscript is now suitable for the publication in ***Nature Communications***.